# FUSED PARTIAL GROMOV-WASSERSTEIN FOR STRUCTURED OBJECTS

## ABSTRACT

Structured data, such as graphs, plays a vital role in machine learning due to its capacity to capture complex relationships and interactions. The Fused Gromov-Wasserstein (FGW) distance has recently garnered significant interest as it enables comparison of structured data by jointly considering feature similarity and geometric structure. However, as a variant of optimal transport (OT), classical FGW assumes equal mass constraints on the compared data. In this work, we relax this constraint and propose the Fused Partial Gromov-Wasserstein (FPGW) framework, which extends FGW to accommodate unbalanced data. Theoretically, we establish the relationship between FPGW and FGW and prove the metric properties of FPGW. We develop Frank-Wolfe and Sinkhorn solvers for the proposed FPGW framework. Finally, we evaluate the FPGW distance through graph matching, classification, and clustering experiments, demonstrating its robust performance. The code for reproducing all numerical results is available in the anonymous repository at `https://anonymous.4open.science/r/fused-pgw-041B`.

## 1 INTRODUCTION

Analyzing structured data, which combines feature-based and relational information, represents a longstanding challenge in machine learning, data science, and statistics. Graphs constitute a classical type of structured data, where nodes with attributes model data features while edges describe structural relationships. Examples of such data structures are abundant, including molecular graphs for drug discovery Ruddigkeit et al. (2012), functional and structural brain networks Bassett & Sporns (2017), and social network graphs Hamilton et al. (2017). Beyond graphs, structured data encompasses diverse domains, including sequences Graves et al. (2006), hierarchical structures such as trees Billé (2008), and pixel-based data such as images Wang et al. (2004).

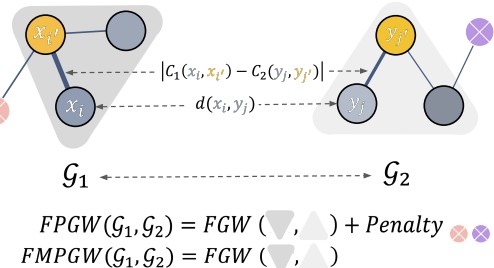

$$FPGW(\mathcal{G}_1, \mathcal{G}_2) = FGW\ (\ \triangledown,\ \triangle\ ) + Penalty$$
$$FMPGW(\mathcal{G}_1, \mathcal{G}_2) = FGW\ (\ \triangledown,\ \triangle\ )$$

Figure 1: An intuitive understanding of the fused-PGW problem in (9). The distance between node features is modeled by the metric $d(\cdot, \cdot)$. The structural information of each graph is represented by its shortest path distance matrices $C_1(\cdot, \cdot)$ and $C_2(\cdot, \cdot)$. The orange and purple nodes correspond to the destroyed and created mass, which contribute to the mass-penalty term. The remaining nodes represent the transported mass.

The Optimal Transport (OT) distance Villani (2009) and its extensions, including unbalanced Optimal Transport Chizat et al. (2018b); Figalli (2010), linear Optimal Transport Wang et al. (2013); Cai et al. (2022); Bai et al. (2023a); Martin et al. (2023), sliced optimal transport Kolouri et al. (2019); Bonneel et al. (2015); Bai et al. (2023b), and expected optimal transport Rowland et al. (2019), have become widely adopted in machine learning tasks due to their ability to measure similarity between unstructured datasets, such as point clouds, images, or data in Euclidean domains. Building on classical OT, the Gromov-Wasserstein problem and its unbalanced extensions Mémoli (2011; 2009); Séjourné et al. (2021);

Chapel et al. (2020); Bai et al. (2023c); Kong et al. (2024) have been proposed to capture the inherent structure of data.

While classical OT can incorporate data features to measure similarity, Gromov-Wasserstein formulations capture structural information. Inspired by these works, the fused Gromov-Wasserstein Titouan et al. (2019b); Vayer et al. (2020) has emerged as a powerful tool for analyzing attributed graphs. This method can be viewed as a "linear combination" of classical optimal transport (OT) and the Gromov-Wasserstein distance. Despite its successful applications in graph data analysis, the fused GW formulation, like classical OT, requires an equal mass constraint. To address this limitation, Fused Unbalanced Gromov Wasserstein (FUGW) Thual et al. (2022); Halmos et al. (2025) and related Sinkhorn solvers have been proposed and applied in brain image analysis. However, FUGW relies exclusively on the Sinkhorn solver, and its metric properties remain unclear. To address these limitations and bridge the theoretical gap, we introduce the fused-partial Gromov-Wasserstein formulations in this paper:

- We introduce the fused partial Gromov-Wasserstein formulations (11), which enable comparison of structured objects with unequal total mass. Theoretically, we demonstrate that FPGW admits a (semi-)metric.
- We develop Frank-Wolfe and Sinkhorn algorithms to solve the FPGW problem. Additionally, we present the FPGW barycenter and related computational solvers.
- We evaluate FPGW in graph matching, clustering, and classification experiments, demonstrating that FPGW-based methods exhibit superior robustness.

## 2 BACKGROUND: GROMOV-WASSERSTEIN (GW) PROBLEMS

Graph-structured data can be represented as a metric measure space (mm-space) consisting of a set $X$ endowed with a metric structure—that is, a notion of distance $d_X$ between its elements—and equipped with a Borel measure $\mu$. Following Mémoli (2011, Ch. 5), we assume that $X$ is compact and that $\mathrm{supp}(\mu) = X$. Given two probability mm-spaces $\mathbb{X} = (X, d_X, \mu)$, $\mathbb{Y} = (Y, d_Y, \nu)$, with $\mu \in \mathcal{P}(X)$ and $\nu \in \mathcal{P}(Y)$, and a non-negative lower semi-continuous cost function $L : \mathbb{R}^2 \to \mathbb{R}_+$ (e.g., the Euclidean distance or the KL-loss), the Gromov-Wasserstein (GW) matching problem is defined as:

$$GW(\mathbb{X}, \mathbb{Y}) := \inf_{\gamma \in \Gamma(\mu, \nu)} \gamma^{\otimes 2}(L(d_X^r(\cdot, \cdot), d_Y^r(\cdot, \cdot))), \tag{1}$$

where $r \geq 1$ and

$$\Gamma(\mu, \nu) := \{\gamma \in \mathcal{P}(X \times Y) : \gamma_1 = \mu, \gamma_2 = \nu\}. \tag{2}$$

For brevity, we employ the notation $\gamma^{\otimes 2}$ for the product measure $d\gamma^{\otimes 2}((x, y), (x', y')) = d\gamma(x, y) d\gamma(x', y')$. If $L(a, b) = |a - b|^q$, for $1 \leq q < \infty$, we denote $GW(\cdot, \cdot)$ by $d_{GW}^q(\cdot, \cdot)$. In this case, the expression (1) defines an equivalence relation $\sim$ among probability mm-spaces, i.e., $\mathbb{X} \sim \mathbb{Y}$ if and only if $d_{GW}(\mathbb{X}, \mathbb{Y}) = 0$[1]. A minimizer of the GW problem (1) always exists, and thus, we can replace inf by min. Moreover, similar to OT, the above GW problem defines a distance for probability mm-spaces after taking the quotient under $\sim$. For details, we refer to Mémoli (2011); Bai et al. (2024, Ch. 5 and 10).

Classical GW requires an equal mass assumption, i.e., $|\mu| = |\nu|$, which limits its applicability to many machine learning tasks, such as positive unsupervised learning Chapel et al. (2020); Séjourné et al. (2023). To address this issue, the above formulation has been recently extended to the unbalanced setting Chapel et al. (2020); Séjourné et al. (2023); Bai et al. (2024; 2023a). In particular, two equivalent extensions of the Gromov-Wasserstein problem, named **Partial Gromov-Wasserstein problem** and **Mass-constrained Partial Gromov-Wasserstein problem**, have been proposed:

$$PGW(\mathbb{X}, \mathbb{Y}) = \inf_{\gamma \in \Gamma_{\leq}(\mu, \nu)} \gamma^{\otimes 2}(L(d_X^r, d_Y^r)) + \lambda(|\mu^{\otimes 2} - \gamma_1^{\otimes 2}| + |\nu^{\otimes 2} - \gamma_2^{\otimes 2}|) \tag{3}$$

$$MPGW(\mathbb{X}, \mathbb{Y}) = \inf_{\gamma \in \Gamma_{\leq}^{\rho}(\mu, \nu)} \gamma^{\otimes 2}(L(d_X^r, d_Y^r)). \tag{4}$$

---

[1]Moreover, given two probability mm-spaces $\mathbb{X}$ and $\mathbb{Y}$, $d_{GW}(\mathbb{X}, \mathbb{Y}) = 0$ if and only if there exists a bijective isometry $\phi : X \to Y$ such that $\phi_\# \mu = \nu$. In particular, the GW distance is invariant under rigid transformations (translations and rotations) of a given probability mm-space.

where

$$\Gamma_{\leq}(\mu, \nu) := \{\gamma \in \mathcal{M}_+(X \times Y) : \gamma_1 \leq \mu, \gamma_2 \leq \nu\}, \tag{5}$$

$$\Gamma_{\leq}^{\rho}(\mu, \nu) := \{\gamma \in \Gamma_{\leq}(\mu, \nu) : |\gamma| \geq \rho\}, \rho \in [0, \min(|\mu|, |\nu|)], \tag{6}$$

and the notation $\gamma_1 \leq \mu$ denotes that for each Borel set $B \subset X$, $\gamma_1(B) \leq \mu(B)$.

Both GW and its unbalanced extension can measure similarity of structured data by utilizing their internal structure distances $d_X$ and $d_Y$. This formulation naturally enables similarity measurement between graph data $(X, E_X)$ and $(Y, E_Y)$, as we can define $d_X$ (and $d_Y$) using the (weights of) edges $E_X$ (and $E_Y$). However, when nodes $X$ and $Y$ contain features (e.g., in attributed graphs), classical GW/PGW cannot incorporate feature information when defining distances between node pairs $x \in X$ and $y \in Y$. To address this limitation, the **Fused Gromov-Wasserstein distance** has been proposed.

## 2.1 FUSED GROMOV-WASSERSTEIN PROBLEM

Given two $mm-$spaces $\mathbb{X}, \mathbb{Y}, \omega_1, \omega_2 \geq 0$ with $\omega_1 + \omega_2 = 1$, a cost function $C : X \times Y \to \mathbb{R}_+$, the fused Gromov-Wasserstein problem is defined as:

$$FGW(\mathbb{X}, \mathbb{Y}) := \inf_{\gamma \in \Gamma(\mu, \nu)} \omega_1 \gamma(C) + \omega_2 \gamma^{\otimes 2}(L(d_X^r, d_Y^r)). \tag{7}$$

Similar to the original GW problem, the above problem admits a minimizer. In addition, when $C(x, y) = \|x - y\|^q$, and $L(\cdot_1, \cdot_2) = |\cdot_1 - \cdot_2|^q$, it defines a semi-metric Titouan et al. (2019a).

## 2.2 FUSED UNBALANCED GROMOV-WASSERSTEIN PROBLEM.

The above formulation relies on the equal mass assumption, i.e., $|\mu| = |\nu|$. By relaxing this constraint, Thual et al. (2022) proposed the following fused-UGW problem:

$$FUGW_\lambda(\mathbb{X}, \mathbb{Y}) := \inf_{\gamma \in \mathcal{M}_+(X \times Y)} \omega_1 \gamma(C) + \omega_2 \gamma^{\otimes 2}(L(d_X^r, d_Y^r)) + \lambda(D_{\phi_1}(\gamma_1^{\otimes 2} \| \mu^{\otimes 2}) + D_{\phi_2}(\gamma_2^{\otimes 2} \| \nu^{\otimes 2})), \tag{8}$$

where $\mathcal{M}_+(X \times Y)$ denotes the set of all positive Radon measures defined on $X \times Y$, and $D_{\phi_i}, i \in [1:2]$ are the f-divergence terms. In Thual et al. (2022), $D_{\phi_i}, i \in [1:2]$ are set as KL divergences. The authors employ entropic regularization and develop a corresponding Sinkhorn solver.

# 3 FUSED PARTIAL GROMOV-WASSERSTEIN PROBLEM

Inspired by these previous works, we set the f-divergence terms $D_{\phi_i}$ to be the total variation and propose the following "fused partial Gromov-Wasserstein problem" and its corresponding mass-constrained version:

$$FPGW_\lambda(\mathbb{X}, \mathbb{Y}) = \inf_{\gamma \in \mathcal{M}_+(X \times Y)} \omega_1 \gamma(C) + \omega_2 \gamma^{\otimes 2}(L(d_X^r, d_Y^r)) + \lambda(|\mu^{\otimes 2} - \gamma_1^{\otimes 2}|_{TV} + |\nu^{\otimes 2} - \gamma_2^{\otimes 2}|_{TV}), \tag{9}$$

$$FMPGW_\rho(\mathbb{X}, \mathbb{Y}) = \inf_{\gamma \in \Gamma_{\leq}^{\rho}(\mu, \nu)} \omega_1 \gamma(C) + \omega_2 \gamma^{\otimes 2}(L(d_X^r, d_Y^r)). \tag{10}$$

where $\lambda \geq 0$.

**Remark 3.1.** *The discrete version of formulation (10) has been discussed in Liu et al. (2023a), and a special case of (10) was introduced in Pan et al. (2024). In September 2025, both the Frank–Wolfe and Sinkhorn solvers for this formulation were implemented in PythonOT Flamary et al. (2021). Therefore, we **do not** claim this formulation or its associated solvers as contributions of this paper. We include it only as a byproduct of our main developments, in order to complete the theoretical foundation and to provide a consistent comparison with related methods.*

**Remark 3.2.** *In theory, the FPGW can be treated as "Language form" of FMPGW, however, their equivalent relation is unclear in general due to the non-convexity issue. We refer Proposition K.1 in the appendix. In practice, we observe the proposed Frank Wolfe and Sinkhorn solver FPGW (see the next section) is significantly faster than FMPGW. We refer the Appendix K.2 for details.*

**Theorem 3.3.** *We have the following:*

*(1) When $C(x, y) \geq 0, \forall x, y, L(r_1, r_2) \geq 0, \forall r_1, r_2 \in \mathbb{R}$, the problem (9) can be further simplified as*

$$FPGW_\lambda(\mathbb{X}, \mathbb{Y}) = \inf_{\gamma \in \Gamma_{\leq}(\mu, \nu)} \omega_1 \gamma(C) + \omega_2 \gamma^{\otimes 2}(L(d_X^r, d_Y^r) - 2\lambda) + \lambda(|\mu|^2 + |\nu|^2) \quad (11)$$

*(2) The problems (9), (11) and (10) admit a minimizer $\gamma$.*

*(3) When $C(x, y) = |x - y|^q, L(\cdot_1, \cdot_2) = |\cdot_1 - \cdot_2|^q, \omega_2, \lambda > 0$, the above formulation (11) admits a semi-metric. Furthermore, when $q = 1$, (11) defines a metric.*

### 3.1 ALGORITHMS FOR DISCRETE FPGW

In the discrete case, let $\mu = \sum_{i=1}^n p_i \delta_{x_i}$ and $\nu = \sum_{i=1}^m q_j \delta_{y_j}$. Let $C^X = [d_X(x_i, x_{i'})]_{i,i'} \in \mathbb{R}^{n \times n}$ and $C^Y = [d_Y(y_j, y_{j'})]_{j,j'}$. The pairs $(C^X, \mu)$ and $(C^Y, \nu)$ represent the mm-spaces $\mathbb{X} = (X, d_X, \mu)$ and $\mathbb{Y} = (Y, d_Y, \nu)$, respectively. We discuss only the Frank-Wolfe and Sinkhorn solvers for FPGW 11 in the main text. The solvers for FMPGW 10 are presented in the appendix.

### 3.2 FRANK-WOLFE ALGORITHM

The above FPGW problem (11) becomes the following:

$$FPGW_\lambda(\mathbb{X}, \mathbb{Y}) = \min_{\gamma \in \gamma_{\leq}^\rho(p,q)} \underbrace{\omega_1 \langle C, \gamma \rangle + \omega_2 \langle (M - 2\lambda) \circ \gamma, \gamma \rangle}_{\mathcal{L}_{C, M-2\lambda}} + \underbrace{\lambda(|\mu|^2 + |\nu|^2) \rangle}_{\text{constant}} \quad (12)$$

where $C = [d(x_i, y_j)]_{i \in [1:n], j \in [1:m]}, M = [|d_X(x_i, x_{i'}) - d_Y(y_j, y_{j'})|^2]_{i,i' \in [1:n], j,j' \in [1:m]}$ are defined in the previous subsection, $M - 2\lambda$ denote the elementwise subtraction, and the constant term will be ignored in the remainder of the paper.

Similarly to the Fused Gromov-Wasserstein problem, we propose the following Frank-Wolfe algorithm as a solver: The above problem will be solved iteratively. In every iteration, say $k$, we will adapt the following steps:

**Step 1. Gradient computation.**

Suppose $\gamma^{(k-1)}$ is the transportation plan in the previous iteration; it is straightforward to verify:

$$\nabla \mathcal{L}_{C, M-2\lambda}(\gamma) = \omega_1 C + \omega_2 (M + M^\top - 4\lambda) \circ \gamma.$$

Next, we aim to find the optimal $\gamma \in \Gamma_{\leq}(\mu, \nu)$ for the following partial OT problem:

$$\gamma^{(k)\prime} := \arg \min_{\gamma \in \Gamma_{\leq}(\mu, \nu)} \langle \nabla \mathcal{L}_{C, M-2\lambda}(\gamma^{(k-1)}), \gamma^k \rangle. \quad (13)$$

**Step 2. linear search algorithm.** In this step, we aim to find the optimal step size $\alpha^* \in [0, 1]$. In particular,

$$\alpha^* := \arg \min_{\alpha \in [0,1]} \mathcal{L}_{C, M-2\lambda}((1 - \alpha)\gamma^{(k-1)} + \alpha\gamma^{(k)\prime}).$$

The term $\alpha^*$ is given by the following:

$$\alpha^* = \begin{cases} 1 & \text{if } a \leq 0, a + b \leq 0, \\ 0 & \text{if } a \leq 0, a + b > 0 \\ \text{clip}(\frac{-b}{2a}, [0, 1]), & \text{if } a > 0 \end{cases}, \begin{cases} a &= \omega_2 \langle (M - 2\lambda) \circ \delta\gamma, \delta\gamma \rangle \\ b &= \langle \omega_2 (M + M^\top - 4\lambda) \circ \gamma^{(k-1)} + \omega_2 C, \delta\gamma \rangle \\ \delta\gamma &= \gamma^{(k)\prime} - \gamma^{(k-1)} \end{cases}$$
$$(14)$$

In algorithm (1), the computational complexity can be written as $\mathcal{O}(\text{C} \cdot \text{L})$, where $\text{C}$ is the complexity of each iteration and $\mathcal{L}$ is the number of iterations that the algorithms converge.

When a linear programming solver Bonneel et al. (2011) for partial OT is adopted, $\mathcal{C} = (n+m)nm$. If we adapt Sinknorn algorithm Cuturi & Doucet (2014), $\mathcal{C} = \mathcal{O}(\frac{1}{\epsilon} \ln(n + m)nm)$ where $\epsilon$ is weight of Sinkhorn term. The number of iterations $\mathcal{L}$ refers to the convergence analysis of the FW algorithm. We refer to the Appendix (H) for details.

---

**Algorithm 1:** Frank-Wolfe Algorithm for FPGW

---

**Input:** $C \in \mathbb{R}^{n \times m}, C^X \in \mathbb{R}^{n \times n}, C^Y \in \mathbb{R}^{m \times m}, p \in \mathbb{R}_+^n, q \in \mathbb{R}_+^m, \omega_2 \in [0,1], \lambda \geq 0.$

**Output:** $\gamma^{(final)}$

**for** $k = 1, 2, \ldots$ **do**

    $G^{(k)} \leftarrow \omega_1 C + \omega_2 (M + M^\top - 4\lambda) \circ \gamma^{(k)}$ // Compute gradient

    $\gamma^{(k)\prime} \leftarrow \arg\min_{\gamma \in \Gamma_\leq(\text{p,q})} \langle G^{(k)}, \gamma \rangle_F$ // Solve the POT problem.

    Compute $\alpha^{(k)} \in [0, 1]$ via (14) // Line Search

    $\gamma^{(k+1)} \leftarrow (1 - \alpha^{(k)})\gamma^{(k)} + \alpha^{(k)}\gamma^{(k)\prime}$ // Update $\gamma$

    if convergence, break

**end for**

$\gamma^{(final)} \leftarrow \gamma^{(k)}$

---

### 3.3 Sinkhorn Algorithm

Another popular solver for the Gromov Wasserstein problem is the Sinkhorn algorithm Séjourné et al. (2023). In the Fused-PGW setting, the problem is defined as:

$$EFPGW(\mathbb{X}, \mathbb{Y}) := \min_{\gamma \in \Gamma_\leq(\mu, \nu)} \mathcal{L}(\gamma) + \epsilon H(\gamma^{\otimes 2} \| (\mu \otimes \nu)^{\otimes 2}) \qquad (15)$$

$$H(A \| B) = \int \frac{dA}{dB} dA, \text{ is the relative entropy, for any positive radon measures } A, B$$

$$\mathcal{L}(\gamma) := \mathcal{L}_{C, M-2\lambda} = \omega_1 \langle c, \gamma \rangle + \omega_2 \langle L(d_X^r, d_Y^r), \gamma^{\otimes 2} \rangle + \lambda(|\mu|^2 + |\nu|^2 - 2|\gamma|^2).$$

Problem (15) can be further relaxed as:

$$\min_{\gamma, \pi \in \Gamma_\leq(\mu, \nu)} \mathcal{F}(\gamma, \pi) + \epsilon H(\pi \otimes \gamma \| (\mu \otimes \nu)^{\otimes 2}) \qquad (16)$$

$$\mathcal{F}(\mu, \nu) := \omega_1 \langle d(x, y), \frac{\gamma + \pi}{2} \rangle + \omega_2 \langle L(d_X^r, d_Y^r), \gamma \otimes \pi \rangle + \lambda(|\mu|^2 + |\nu|^2 - 2|\gamma||\pi|)$$

It is clear $\mathcal{F}(\gamma, \gamma) = \mathcal{F}(\gamma)$. Thus, (16) $\leq$ (15), and we denote (16) as $LB - FPGW_\lambda(\mathbb{X}, \mathbb{Y})$ (lower bound of Fused Partial Gromov Wasserstein). **Essentially, the Sinkhorn algorithm aims to solve the $LB - FPGW$ problem.**

We first introduce the following fundamental proposition. Note, a similar version can be found in (Séjourné et al., 2021, Proposition 4):

**Proposition 3.4.** *Given a fixed $\pi \in \Gamma_\leq(\mu, \nu)$, considering the problem:*

$$\min_{\gamma \in \mathcal{M}_+(X \times Y)} \mathcal{F}(\pi, \gamma) + \epsilon H(\pi \otimes \gamma \| (\mu \otimes \nu)^{\otimes 2}).$$

*It is equivalent to solving the following entropic optimal partial transport problem:*

$$\min_{\gamma \in \Gamma_\leq(\mu, \nu)} \int_{X \times Y} c_\pi(x, y) d\gamma + \lambda|\pi|(|\mu| + |\nu| - 2|\gamma|) + \epsilon|\pi|H(\gamma \| \mu \otimes \nu), \qquad (17)$$

*where*

$$c_\pi(x, y) = \frac{1}{2}\omega_1 d(x, y) + \omega_2 [L(d_X^r, d_Y^r) \circ \pi](x, y) + \epsilon H(\pi \| \mu \otimes \nu).$$

Given these fundamental results, we can present the Sinkhorn algorithm 2.

## 4 Numerical Applications

### 4.1 Toy example: Graph clustering

Given a set of unlabeled graphs $\{G_1, \ldots, G_K\}$, we compare the performance of FGW and FPGW in the graph clustering task.

**Algorithm 2:** Sinkhorn Algorithm for FPGW

**Input:** $C \in \mathbb{R}^{n \times m}, C^X \in \mathbb{R}^{n \times n}, C^Y \in \mathbb{R}^{m \times m}, p \in \mathbb{R}_+^n, q \in \mathbb{R}_+^m, \omega_2 \in [0,1], \lambda \geq 0.$
**Output:** $\gamma$
**for** $k = 1, 2, \ldots$ **do**
    $\pi \leftarrow \gamma$
    Solve the Sinkhorn partial OT problem (17):

$$\gamma \leftarrow \min_{\gamma \in \Gamma_{\leq}(\mu,\nu)} \int_{X \times Y} c_\pi(x,y)d\gamma + \lambda |\pi|(|\mu| + |\nu| - 2|\gamma|) + \epsilon |\pi| H(\gamma \parallel \mu \otimes \nu)$$

    Fix $\gamma$ and solve the similar Sinkhorn partial OT problem (17):

$$\pi \leftarrow \min_{\pi \in \Gamma_{\leq}(\mu,\nu)} \int_{X \times Y} c_\gamma(x,y)d\gamma + \lambda |\gamma|(|\mu| + |\nu| - 2|\pi|) + \epsilon |\pi| H(\pi \parallel \mu \otimes \nu)$$

    Rescale $\gamma \leftarrow \sqrt{|\pi|/|\gamma|}\gamma$
    Break if $\pi \approx \gamma$
**end for**

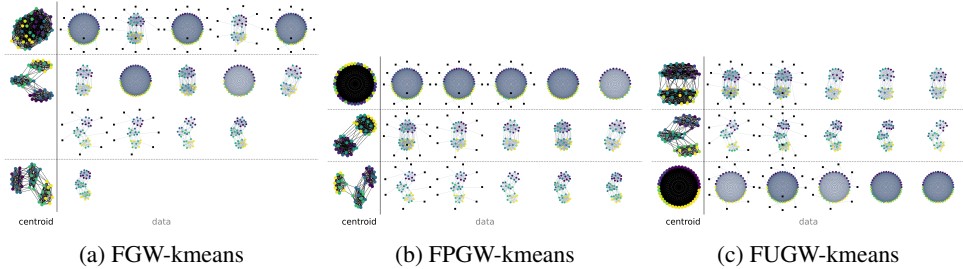

|  |  |  |
|---|---|---|
| (a) FGW-kmeans | (b) FPGW-kmeans | (c) FUGW-kmeans |

Figure 2: We present the clustering results of the FGW, FPGW, and FUGW k-means methods. In the first column, we visualize the centroids obtained using three methods. The centroids are represented by their features and structure (distance matrix). The edges of these graphs are either reconstructed or approximated based on the returned distance matrix. Additionally, the color of each node corresponds to the feature it represents.
The clustering results are shown in the remaining columns. For each graph, the color of regular nodes represents their features, while all outlier nodes are depicted as black squares.

**Dataset setup.** We use the dataset generated by Titouan et al. (2019b); Vayer et al. (2020). Each graph follows a simple Stochastic Block Model, with groups defined by the number of communities within each graph and the distribution of their labels. The dataset comprises three distinct graph types, each containing five graphs. Each graph contains 30 or 40 nodes, with node features randomly selected from the compact set $[-2, 2]$.

Additionally, we randomly generate outliers and add them to the graphs. Specifically, we select 50Similar to the setting of graph classification, for each graph $G$, we suppose $N_G$ is the number of nodes that are not outliers. We assume $N_G$ is known in this experiment. We refer to figure 2 for the visualization of these graphs.

**Baseline: FGW/FUGW k-means.** We consider the FGW/FUGW k-means method introduced in Section 4.3 of Vayer et al. (2020); Titouan et al. (2019b) as the baseline method. Specifically, given graphs $\{G_1, \ldots, G_K\}$ and the number of clusters $K' \leq K$, the method iteratively alternates between the following two steps:

– **Step 1.** For each $i \in [1 : K]$ and $j \in [1 : K']$, compute the distance between each graph $G_i$ and centroid $G'_j$. Assign each graph $G_i$ to the closest centroid based on the computed distances.

– **Step 2.** Using the updated assignments, recompute each centroid $G'_{j'}$, where $j' \in [1 : K']$, as the center of the graphs assigned to it.

In the FGW k-means method, the Fused-GW distance is used to define the distance in Step 1, while the Fused-GW barycenter is used to compute the "center" in Step 2.

**Our method: FPGW k-means.** Inspired by the FGW k-means method, we introduce the FPGW k-means method. In summary, we use the FMPGW discrepancy to measure the distance in Step 1, and introduce the **FPGW barycenter** to define the "center" in Step 2.

First, we convert the graphs to mm-spaces using the formulation introduced in Section L. Specifically, given a graph $G = (V = \{v_1, \ldots, v_N\}, E)$, we define $\mathbb{X} = \left( V, C, \sum_{i=1}^{N} \frac{1}{N_G} \delta_{v_i} \right)$, where $C = [c(v_i, v_{i'})]_{i,i' \in [1:N]} \in \mathbb{R}^{N \times N}$, and $c(v_i, v_{i'})$ is the "shortest path distance" determined by $E$. Here, $N_G$ is the number of nodes that are not outliers, and its value is known based on the experiment setup. The feature distance between nodes $v_i$ and $v_j$ is defined as the Euclidean distance. We then use the Fused-PGW discrepancy (10) to measure the distance between each pair of graphs and centroids in Step 1 of the k-means method.

For Step 2, suppose the set of graphs $\boldsymbol{G}^k \subset \{G_1, \ldots, G_K\}$ is assigned to cluster $k'$ for each $k' \in [1 : K']$. We define the "center" using the following **Fused-PGW barycenter**:

$$\min_{G} \sum_{G_k \in \boldsymbol{G}^k} \frac{1}{|\boldsymbol{G}^k|} \text{FPGW}_{\lambda_k}(G, G_k), \tag{18}$$

where $\lambda_k$ is sufficiently large for all $k$. The solution to this problem provides the updated centroid for cluster $k'$. Details of the formal formulation and solver are provided in Appendix J.

We iteratively repeat the above two steps until all centroids/assignments converge.

**Other parameter setting.** In this experiment, we set $\omega_2 = 0.5$ (see the result of $\omega_2 = 0.999$ in Appendix P) for the FGW, FPGW, and FUGW methods. In addition, we set the number of clustering $K' = 3$ for both methods. For each centroid, we initialize it as a random connected graph with $40$ nodes.

**Performance analysis.** Figure 2 presents the clustering results of FGW K-means, FPGW K-means, and FUGW K-means. The FGW method's performance is significantly degraded by the presence of outlier nodes in half of the graphs. In contrast, FPGW and FUGW demonstrate superior robustness, with clustering results closely aligning with the ground truth. This robustness stems from the partial matching property of FPGW and FUGW. Regarding the wall-clock time, FGW requires 1017.0 seconds, FPGW requires 22.6 seconds, while FUGW requires 1800.0 seconds.

The centroid visualizations for FGW/FPGW/FUGW methods show that FPGW and FUGW effectively exclude most outlier information, while FGW incorporates it into the centroids.

## 4.2  Graph Matching

**Dataset setup**. We evaluate our method on seven widely used graph datasets with continuous node attributes: *Synthetic* Feragen et al. (2013), *Enzymes, Protein, AIDS* Borgwardt & Kriegel (2005), *Cuneiform* Kriege et al. (2016), *COX2, and BZR* Sutherland et al. (2003). Each dataset contains hundreds or thousands of connected graphs. For each graph, we use its adjacency matrix as structural information and node attributes as features. To create partial matching tasks, we use breadth-first search (BFS) to randomly extract subgraphs containing 50% of the original nodes and their corresponding edges.[2] We also evaluate performance on the *Douban dataset* Zhang & Tong (2019), which provides a large online graph (3,906 nodes) and a smaller offline subgraph (1,118 nodes), using user locations as node features.

**Baselines.** We compare sink-FPGW against competitive methods, including balanced GW methods, *SpecGW* Chowdhury & Needham (2021), *eBPG* Solomon et al. (2016), *BPG* Xu et al. (2019), *BAPG* Li et al. (2023), *srGW* Vincent-Cuaz et al. (2022), and unbalanced GW methods: *UGW* Séjourné et al. (2021), *PGW* Chapel et al. (2020); Bai et al. (2024), *RGW* Kong et al. (2024), *FUGW* Thual et al. (2022) and FMPGW-Frank-Wolfe.

**Settings of GW methods and our method.** In all these methods, we first convert graphs into mm-spaces (see appendix L). In all these methods, we default the probability mass function of the query

---

[2] The value 50% is unknown to all methods in the experiment.

graph and original graph as $\mu = \sum_{i=1}^{m} p_i \delta_{v_i}, \nu_{i=1}^{n} q_j \delta_{v_i}$, with $p_i = 1/m, q_j = 1/n$, where $m$ and $n$ denote the numbers of the source and target nodes. For the computation of the cost matrix C in (11), we use the Euclidean distance among the continuous node attributes. See Appendix M for the detailed parameter settings for all the methods.

**Evaluation metrics and performance analysis.** Accuracy is defined as the fraction of ground-truth correspondences correctly recovered in the predicted set,

$$\text{Acc} = \frac{|S_{\text{gt}} \cap S_{\text{pred}}|}{|S_{\text{gt}}|} \times 100\%, \tag{19}$$

following Kong et al. (2024). Table 1 shows that our proposed Sinkhorn-FPGW achieves the highest accuracy on most datasets while maintaining strong computational efficiency. Because the experiment setting requires partial matching, all balanced methods (such as SpecGW and eBPG) exhibit low accuracy. PGW and UGW achieve relatively high accuracy but are unable to incorporate node-feature information. FGW is also limited by its balanced formulation.

Although FPGW-FW and FMPGW-FW can in principle produce partial matchings, their accuracy is severely impacted in practice by the highly non-convex optimization landscape induced by using 0–1 adjacency matrices as costs, which causes the Frank–Wolfe iterations to frequently converge to poor local minima. We emphasize that the FW-based variants are included here primarily for completeness of comparison.

FUGW and our Sinkhorn-FPGW obtain the best accuracy, with our method being significantly faster. This advantage arises from Sinkhorn-FPGW's ability to jointly utilize node features (linear term) and structural information (quadratic term), in contrast to methods that rely solely on structure. This design not only improves accuracy but also provides substantial computational benefits, making Sinkhorn-FPGW consistently faster than existing alternatives.

### 4.3 Geometry Matching

**Dataset setup**. We evaluate our method on a partial geometry matching task using the 'Victoria' model from the *TOSCA* Rodolà et al. (2015) 3D mesh dataset. This high-resolution mesh comprises 10,000 nodes, segmented into two parts: an upper body (4,691 nodes) and a lower body (5,436 nodes).

**Our methods.** For mesh objects, we encode them as mm-spaces using the approach described in Section L and then apply Algorithm 2 (sink-FPGW) to solve optimization problem (11) for the geometry matching task. Specifically, we define the source and target mass distribution functions as $p_i = q_j = \frac{1}{\max(n,m)}$ for all methods, where $m$ and $n$ denote the number of nodes in the source and target graphs. For both complete and partial geometries, we construct node features using the Euclidean distances from each node to a single randomly selected anchor node. The mesh structure is encoded via the adjacency matrix.

**Baselines.** We benchmark our proposed sink-FPGW method against two competitive baselines, RGW and FUGW. To ensure a fair comparison, RGW is initialized with a transport plan derived from node features, and the regularization hyperparameters for both FUGW and sink-FPGW are optimized via line search. We refer M for the detailed parameter settings for all the methods.

**Evaluation metric and performance analysis.**

Following the RGW evaluation protocol, sink-FPGW achieves matching accuracies of $99.47\%$ and $99.77\%$ on the two partial matching tasks ($43.47\%$ and $48.05\%$ in RGW, and $98.99\%$ and $56.09\%$ in FUGW), where $\text{Acc} = \frac{|S_{\text{gt}} \cap S_{\text{pred}}|}{|S_{\text{gt}}|} \times 100\%$. We visualize the one-hot transport plans as real shapes (Figure 3 (b)-(d)) and heatmaps (Figure 3 (e)-(h)), confirming that sink-FPGW yields the most accurate matches. The one-hot plans are obtained by taking the argmax match for each source node from the transport plans.

### 5 Summary.

In this paper, we propose a novel formulation called "fused-partial Gromov-Wasserstein" (fused-PGW) for comparing structured objects. Theoretically, we demonstrate the metric properties of

Table 1: Comparison of methods on multiple datasets. Acc = accuracy, Time = runtime. For the first seven datasets (excluding Douban), Time corresponds to the total graph set matching time; for Douban, it reflects the runtime of a single matching instance.

| Method | Synthetic | | Enzymes | | Cuneiform | | COX2 | |
| --- | --- | --- | --- | --- | --- | --- | --- | --- |
| | Acc | Time | Acc | Time | Acc | Time | Acc | Time |
| SpecGW | $1.03^{\pm 0.014}$ | 4.10 | $9.98^{\pm 0.11}$ | 3.13 | $7.15^{\pm 0.12}$ | 1.06 | $5.27^{\pm 0.06}$ | 2.82 |
| eBPG | $2.00^{\pm 0.012}$ | 121.84 | $12.68^{\pm 0.12}$ | 9950.78 | $5.99^{\pm 0.09}$ | 4310.76 | $8.99^{\pm 0.06}$ | 5760.38 |
| BPG | $4.00^{\pm 0.01}$ | 21.12 | $28.18^{\pm 0.20}$ | 90.71 | $5.81^{\pm 0.07}$ | 12.13 | $23.32^{\pm 0.13}$ | 48.88 |
| BAPG | $88.03^{\pm 0.05}$ | 45.85 | $62.07^{\pm 0.24}$ | 14.34 | $72.05^{\pm 0.17}$ | 2.02 | $21.23^{\pm 0.10}$ | 15.68 |
| srGW | $1.03^{\pm 0.01}$ | 55.18 | $15.22^{\pm 0.24}$ | 64.24 | $19.94^{\pm 0.09}$ | 7.24 | $1.67^{\pm 0.02}$ | 33.12 |
| srFGW | $9.20^{\pm 0.06}$ | 0.74 | $9.76^{\pm 0.12}$ | 0.45 | $24.06^{\pm 0.15}$ | 0.14 | $5.88^{\pm 0.07}$ | 0.29 |
| PGW | $1.00^{\pm 0.013}$ | 26.30 | $9.74^{\pm 0.13}$ | 17.26 | $9.94^{\pm 0.09}$ | 14.73 | $9.41^{\pm 0.11}$ | 11.69 |
| UGW | $2.00^{\pm 0.00}$ | 12.84 | $17.83^{\pm 0.26}$ | 732.21 | $4.17^{\pm 0.08}$ | 375.36 | $3.93^{\pm 0.05}$ | 65.03 |
| RGW | $37.24^{\pm 0.23}$ | 424.68 | $77.20^{\pm 0.28}$ | 137.70 | $85.33^{\pm 0.21}$ | 25.87 | $37.05^{\pm 0.15}$ | 188.99 |
| FGW | $46.68^{\pm 0.06}$ | 5.05 | $62.92^{\pm 0.24}$ | 9.26 | $86.78^{\pm 0.08}$ | 1.20 | $73.28^{\pm 0.22}$ | 5.32 |
| FUGW | $\mathbf{99.89}^{\pm 0.01}$ | 70.76 | $90.91^{\pm 0.19}$ | 335.71 | $96.88^{\pm 0.04}$ | 109.74 | $90.73^{\pm 0.23}$ | 166.92 |
| FW-FMPGW (POT) | $9.20^{\pm 0.06}$ | 26.82 | $9.81^{\pm 0.13}$ | 50.11 | $23.98^{\pm 0.15}$ | 48.79 | $5.88^{\pm 0.07}$ | 58.11 |
| FW-FPGW (ours) | $9.20^{\pm 0.06}$ | 7.90 | $9.86^{\pm 0.13}$ | 6.8 | $23.98^{\pm 0.15}$ | 0.74 | $5.88^{\pm 0.07}$ | 2.15 |
| sink-FPGW (ours) | $99.70^{\pm 0.01}$ | 4.90 | $\mathbf{93.47}^{\pm 0.18}$ | 1.81 | $\mathbf{99.96}^{\pm 0.01}$ | 0.32 | $\mathbf{92.62}^{\pm 0.23}$ | 1.38 |

| Method | BZR | | Protein | | AIDS | | Douban | |
| --- | --- | --- | --- | --- | --- | --- | --- | --- |
| | Acc | Time | Acc | Time | Acc | Time | Acc | Time |
| SpecGW | $9.51^{\pm 0.07}$ | 2.08 | $12.24^{\pm 0.15}$ | 9.29 | $23.89^{\pm 0.19}$ | 7.36 | 0.00 | 65.11 |
| eBPG | $14.21^{\pm 0.10}$ | 5540.20 | $14.88^{\pm 0.16}$ | 15936.79 | $25.98^{\pm 0.20}$ | 26182.34 | 0.09 | 13.24 |
| BPG | $25.67^{\pm 0.18}$ | 48.78 | $30.07^{\pm 0.19}$ | 134.55 | $30.49^{\pm 0.22}$ | 106.86 | 58.59 | 391.00 |
| BAPG | $34.54^{\pm 0.15}$ | 11.11 | $25.52^{\pm 0.22}$ | 66.90 | $50.08^{\pm 0.25}$ | 15.42 | 53.94 | 2194.70 |
| srGW | $3.20^{\pm 0.04}$ | 26.55 | $18.96^{\pm 0.26}$ | 149.17 | $23.76^{\pm 0.24}$ | 150.31 | 4.38 | 1584.68 |
| srFGW | $5.82^{\pm 0.07}$ | 0.26 | $11.13^{\pm 0.16}$ | 1.62 | $14.98^{\pm 0.19}$ | 1.10 | 0.09 | 7.92 |
| PGW | $6.80^{\pm 0.08}$ | 9.59 | $13.22^{\pm 0.18}$ | 97.74 | $22.48^{\pm 0.21}$ | 22.89 | 2.06 | 1363.59 |
| UGW | $6.76^{\pm 0.07}$ | 152.98 | $15.57^{\pm 0.22}$ | 1280.80 | $21.32^{\pm 0.20}$ | 1152.91 | 0.09 | 702.53 |
| RGW | $39.54^{\pm 0.19}$ | 120.97 | $38.26^{\pm 0.30}$ | 511.67 | $57.05^{\pm 0.34}$ | 340.94 | 51.88 | 17784.09 |
| FGW | $74.10^{\pm 0.27}$ | 8.09 | $63.95^{\pm 0.24}$ | 29.29 | $84.31^{\pm 0.21}$ | 6.76 | 24.60 | 367.15 |
| FUGW | $89.21^{\pm 0.23}$ | 133.96 | $74.87^{\pm 0.23}$ | 733.48 | $97.12^{\pm 0.10}$ | 617.36 | 66.37 | 1674.82 |
| FW-FPGW (POT) | $5.82^{\pm 0.07}$ | 53.49 | $11.14^{\pm 0.16}$ | 205.80 | $14.94^{\pm 0.19}$ | 2280.60 | 6.44 | 19.44 |
| FW-FPGW (ours) | $5.82^{\pm 0.07}$ | 1.65 | $11.09^{\pm 0.16}$ | 41.53 | $14.94^{\pm 0.19}$ | 6.42 | 22.54 | 3478.20 |
| sink-FPGW (ours) | $\mathbf{93.51}^{\pm 0.24}$ | 0.97 | $\mathbf{96.21}^{\pm 0.15}$ | 7.17 | $\mathbf{98.89}^{\pm 0.09}$ | 2.33 | $\mathbf{66.99}$ | 40.3306.64 |

fused-PGW, and computationally, we develop corresponding Frank-Wolfe and Sinkhorn solvers along with barycenter algorithms. Finally, we evaluate fused-PGW on graph matching and clustering experiments, showing that it achieves superior robustness due to its partial matching property.

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

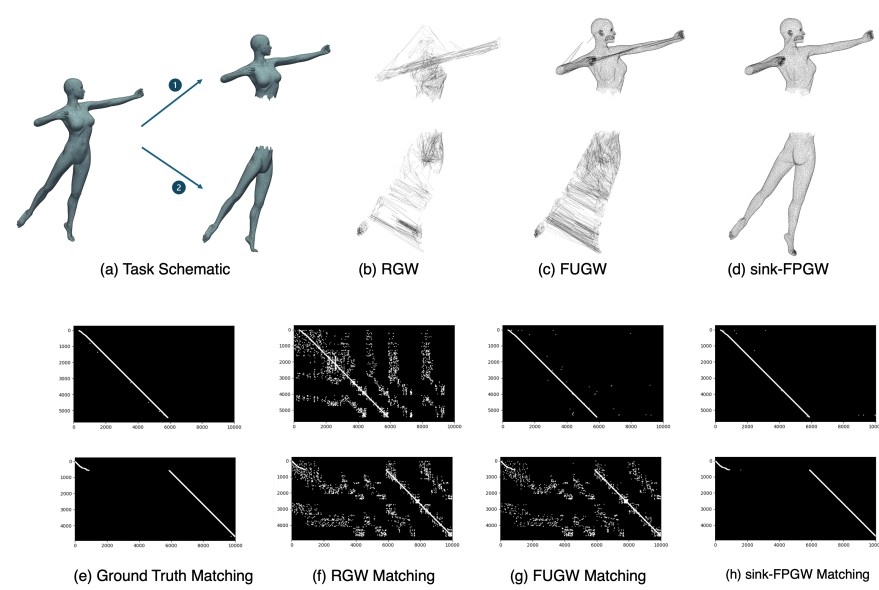

(a) Task Schematic      (b) RGW      (c) FUGW      (d) sink-FPGW

(e) Ground Truth Matching    (f) RGW Matching    (g) FUGW Matching    (h) sink-FPGW Matching

Figure 3: Qualitative results for the partial geometry matching experiment. (a): Schematic illustrating the two whole-to-partial matching tasks; (b)-(d): Visualizations of the partial mesh reconstructed from the whole source mesh using the transport plans computed by RGW, FUGW, and sink-FPGW.; (e)-(h) Corresponding heatmap visualizations for the ground truth, RGW, FUGW, and sink-FPGW, the top row corresponding to the first partial matching task (upper body) and the bottom row to the second (lower body). The x- and y-axes represent the vertex indices of the whole and partial meshes, respectively.

Danielle S Bassett and Olaf Sporns. Network neuroscience. *Nature neuroscience*, 20(3):353–364, 2017.

Florian Beier, Robert Beinert, and Gabriele Steidl. On a linear gromov–wasserstein distance. *IEEE Transactions on Image Processing*, 31:7292–7305, 2022.

Jean-David Benamou, Guillaume Carlier, Marco Cuturi, Luca Nenna, and Gabriel Peyré. Iterative bregman projections for regularized transportation problems. *SIAM Journal on Scientific Computing*, 37(2):A1111–A1138, 2015.

Raphaël Billé. Integrated coastal zone management: four entrenched illusions. *SAPI EN. S. Surveys and Perspectives Integrating Environment and Society*, (1.2), 2008.

Nicolas Bonneel, Michiel Van De Panne, Sylvain Paris, and Wolfgang Heidrich. Displacement interpolation using lagrangian mass transport. In *Proceedings of the 2011 SIGGRAPH Asia conference*, pp. 1–12, 2011.

Nicolas Bonneel, Julien Rabin, Gabriel Peyré, and Hanspeter Pfister. Sliced and Radon Wasserstein barycenters of measures. *Journal of Mathematical Imaging and Vision*, 51(1):22–45, 2015.

Karsten M Borgwardt and Hans-Peter Kriegel. Shortest-path kernels on graphs. In *Fifth IEEE international conference on data mining (ICDM'05)*, pp. 8–pp. IEEE, 2005.

Luis A Caffarelli and Robert J McCann. Free boundaries in optimal transport and monge-ampere obstacle problems. *Annals of mathematics*, pp. 673–730, 2010.

Tianji Cai, Junyi Cheng, Bernhard Schmitzer, and Matthew Thorpe. The linearized hellinger–kantorovich distance. *SIAM Journal on Imaging Sciences*, 15(1):45–83, 2022.

Laetitia Chapel, Mokhtar Z Alaya, and Gilles Gasso. Partial optimal tranport with applications on positive-unlabeled learning. *Advances in Neural Information Processing Systems*, 33:2903–2913, 2020.

Lenaic Chizat, Gabriel Peyré, Bernhard Schmitzer, and François-Xavier Vialard. Scaling algorithms for unbalanced optimal transport problems. *Mathematics of Computation*, 87(314):2563–2609, 2018a.

Lenaic Chizat, Gabriel Peyré, Bernhard Schmitzer, and François-Xavier Vialard. Unbalanced optimal transport: Dynamic and Kantorovich formulations. *Journal of Functional Analysis*, 274(11): 3090–3123, 2018b.

Samir Chowdhury and Tom Needham. Generalized spectral clustering via gromov-wasserstein learning. In *International Conference on Artificial Intelligence and Statistics*, pp. 712–720. PMLR, 2021.

Marco Cuturi and Arnaud Doucet. Fast computation of wasserstein barycenters. In *International conference on machine learning*, pp. 685–693. PMLR, 2014.

Giovanni Da San Martino, Nicolo Navarin, and Alessandro Sperduti. A tree-based kernel for graphs. In *Proceedings of the 2012 SIAM International Conference on Data Mining*, pp. 975–986. SIAM, 2012.

Asim Kumar Debnath, Rosa L Lopez de Compadre, Gargi Debnath, Alan J Shusterman, and Corwin Hansch. Structure-activity relationship of mutagenic aromatic and heteroaromatic nitro compounds. correlation with molecular orbital energies and hydrophobicity. *Journal of medicinal chemistry*, 34(2):786–797, 1991.

Aasa Feragen, Niklas Kasenburg, Jens Petersen, Marleen de Bruijne, and Karsten Borgwardt. Scalable kernels for graphs with continuous attributes. *Advances in neural information processing systems*, 26, 2013.

Alessio Figalli. The optimal partial transport problem. *Archive for rational mechanics and analysis*, 195(2):533–560, 2010.

Alessio Figalli and Nicola Gigli. A new transportation distance between non-negative measures, with applications to gradients flows with dirichlet boundary conditions. *Journal de mathématiques pures et appliquées*, 94(2):107–130, 2010.

Rémi Flamary, Nicolas Courty, Alexandre Gramfort, Mokhtar Z. Alaya, Aurélie Boisbunon, Stanislas Chambon, Laetitia Chapel, Adrien Corenflos, Kilian Fatras, Nemo Fournier, Léo Gautheron, Nathalie T.H. Gayraud, Hicham Janati, Alain Rakotomamonjy, Ievgen Redko, Antoine Rolet, Antony Schutz, Vivien Seguy, Danica J. Sutherland, Romain Tavenard, Alexander Tong, and Titouan Vayer. Pot: Python optimal transport. *Journal of Machine Learning Research*, 22(78): 1–8, 2021. URL `http://jmlr.org/papers/v22/20-451.html`.

Marguerite Frank, Philip Wolfe, et al. An algorithm for quadratic programming. *Naval research logistics quarterly*, 3(1-2):95–110, 1956.

Alex Graves, Santiago Fernández, Faustino Gomez, and Jürgen Schmidhuber. Connectionist temporal classification: labelling unsegmented sequence data with recurrent neural networks. In *Proceedings of the 23rd international conference on Machine learning*, pp. 369–376, 2006.

Peter Halmos, Xinhao Liu, Julian Gold, Feng Chen, Li Ding, and Benjamin J Raphael. Dest-ot: Alignment of spatiotemporal transcriptomics data. *Cell Systems*, 16(2), 2025.

Will Hamilton, Zhitao Ying, and Jure Leskovec. Inductive representation learning on large graphs. *Advances in neural information processing systems*, 30, 2017.

Vinay Jethava, Anders Martinsson, Chiranjib Bhattacharyya, and Devdatt Dubhashi. Lovász $\vartheta$ function, svms and finding dense subgraphs. *The Journal of Machine Learning Research*, 14(1): 3495–3536, 2013.

Soheil Kolouri, Kimia Nadjahi, Umut Simsekli, Roland Badeau, and Gustavo Rohde. Generalized sliced Wasserstein distances. *Advances in Neural Information Processing Systems*, 32, 2019.

Lemin Kong, Jiajin Li, Jianheng Tang, and Anthony Man-Cho So. Outlier-robust gromov-wasserstein for graph data. *Advances in Neural Information Processing Systems*, 36, 2024.

Nils M Kriege, Pierre-Louis Giscard, and Richard Wilson. On valid optimal assignment kernels and applications to graph classification. *Advances in neural information processing systems*, 29, 2016.

Nils M Kriege, Matthias Fey, Denis Fisseler, Petra Mutzel, and Frank Weichert. Recognizing cuneiform signs using graph based methods. In *International Workshop on Cost-Sensitive Learning*, pp. 31–44. PMLR, 2018.

Simon Lacoste-Julien. Convergence rate of frank-wolfe for non-convex objectives. *arXiv preprint arXiv:1607.00345*, 2016.

Jiajin Li, Jianheng Tang, Lemin Kong, Huikang Liu, Jia Li, Anthony Man-Cho So, and Jose H. Blanchet. A convergent single-loop algorithm for relaxation of gromov-wasserstein in graph data. In *International Conference on Learning Representations (ICLR)*, 2023. URL https://arxiv.org/abs/2303.06595. arXiv:2303.06595.

Xinhao Liu, Ron Zeira, and Benjamin J Raphael. Partial alignment of multislice spatially resolved transcriptomics data. *Genome Research*, 33(7):1124–1132, 2023a.

Xinran Liu, Yikun Bai, Huy Tran, Zhanqi Zhu, Matthew Thorpe, and Soheil Kolouri. Ptlp: Partial transport $l^p$ distances. In *NeurIPS 2023 Workshop Optimal Transport and Machine Learning*, 2023b.

László Lovász. On the shannon capacity of a graph. *IEEE Transactions on Information theory*, 25 (1):1–7, 1979.

Rocio Diaz Martin, Ivan Medri, Yikun Bai, Xinran Liu, Kangbai Yan, Gustavo K Rohde, and Soheil Kolouri. Lcot: Linear circular optimal transport. *arXiv preprint arXiv:2310.06002*, 2023.

Facundo Mémoli. Spectral gromov-wasserstein distances for shape matching. In *2009 IEEE 12th International Conference on Computer Vision Workshops, ICCV Workshops*, pp. 256–263. IEEE, 2009.

Facundo Mémoli. Gromov–wasserstein distances and the metric approach to object matching. *Foundations of computational mathematics*, 11:417–487, 2011.

Marion Neumann, Roman Garnett, Christian Bauckhage, and Kristian Kersting. Propagation kernels: efficient graph kernels from propagated information. *Machine learning*, 102:209–245, 2016.

Wen-Xin Pan, Isabel Haasler, and Pascal Frossard. Subgraph matching via partial optimal transport. In *2024 IEEE International Symposium on Information Theory (ISIT)*, pp. 3456–3461. IEEE, 2024.

Gabriel Peyré, Marco Cuturi, and Justin Solomon. Gromov-wasserstein averaging of kernel and distance matrices. In *International conference on machine learning*, pp. 2664–2672. PMLR, 2016.

Benedetto Piccoli and Francesco Rossi. Generalized wasserstein distance and its application to transport equations with source. *Archive for Rational Mechanics and Analysis*, 211:335–358, 2014.

Emanuele Rodolà, Luca Cosmo, Michael M. Bronstein, Andrea Torsello, and Daniel Cremers. Partial functional correspondence. *CoRR*, abs/1506.05274, 2015. URL http://arxiv.org/abs/1506.05274.

Ryan Rossi and Nesreen Ahmed. The network data repository with interactive graph analytics and visualization. In *Proceedings of the AAAI conference on artificial intelligence*, volume 29, 2015.

Mark Rowland, Jiri Hron, Yunhao Tang, Krzysztof Choromanski, Tamas Sarlos, and Adrian Weller. Orthogonal estimation of wasserstein distances. In *The 22nd International Conference on Artificial Intelligence and Statistics*, pp. 186–195. PMLR, 2019.

Lars Ruddigkeit, Ruud Van Deursen, Lorenz C Blum, and Jean-Louis Reymond. Enumeration of 166 billion organic small molecules in the chemical universe database gdb-17. *Journal of chemical information and modeling*, 52(11):2864–2875, 2012.

Filippo Santambrogio. Optimal transport for applied mathematicians. *Birkäuser, NY*, 55(58-63):94, 2015.

Thibault Séjourné, François-Xavier Vialard, and Gabriel Peyré. The unbalanced gromov wasserstein distance: Conic formulation and relaxation. *Advances in Neural Information Processing Systems*, 34:8766–8779, 2021.

Thibault Séjourné, Clément Bonet, Kilian Fatras, Kimia Nadjahi, and Nicolas Courty. Unbalanced optimal transport meets sliced-wasserstein. *arXiv preprint arXiv:2306.07176*, 2023.

Nino Shervashidze, SVN Vishwanathan, Tobias Petri, Kurt Mehlhorn, and Karsten Borgwardt. Efficient graphlet kernels for large graph comparison. In *Artificial intelligence and statistics*, pp. 488–495. PMLR, 2009.

Justin Solomon, Gabriel Peyré, Vladimir Kim, and Suvrit Sra. Entropic metric alignment for correspondence problems. *ACM Transactions on Graphics (Proc. SIGGRAPH 2016)*, 35(4):72:1–72:13, 2016. doi: 10.1145/2897824.2925903. URL http://hal.archives-ouvertes.fr/hal-01305808.

Mahito Sugiyama and Karsten Borgwardt. Halting in random walk kernels. *Advances in neural information processing systems*, 28, 2015.

Jeffrey J Sutherland, Lee A O'brien, and Donald F Weaver. Spline-fitting with a genetic algorithm: A method for developing classification structure- activity relationships. *Journal of chemical information and computer sciences*, 43(6):1906–1915, 2003.

Alexis Thual, Quang Huy Tran, Tatiana Zemskova, Nicolas Courty, Rémi Flamary, Stanislas Dehaene, and Bertrand Thirion. Aligning individual brains with fused unbalanced gromov wasserstein. *Advances in Neural Information Processing Systems*, 35:21792–21804, 2022.

Vayer Titouan, Nicolas Courty, Romain Tavenard, and Rémi Flamary. Optimal transport for structured data with application on graphs. In *International Conference on Machine Learning*, pp. 6275–6284. PMLR, 2019a.

Vayer Titouan, Nicolas Courty, Romain Tavenard, Chapel Laetitia, and Rémi Flamary. Optimal transport for structured data with application on graphs. In Kamalika Chaudhuri and Ruslan Salakhutdinov (eds.), *Proceedings of the 36th International Conference on Machine Learning*, volume 97 of *Proceedings of Machine Learning Research*, pp. 6275–6284, Long Beach, California, USA, 09–15 Jun 2019b. PMLR. URL http://proceedings.mlr.press/v97/titouan19a.html.

Titouan Vayer, Laetitia Chapel, Rémi Flamary, Romain Tavenard, and Nicolas Courty. Fused gromov-wasserstein distance for structured objects. *Algorithms*, 13(9):212, 2020.

Cedric Villani. *Optimal transport: old and new*. Springer, 2009. URL https://www.cedricvillani.org/sites/dev/files/old_images/2012/08/preprint-1.pdf.

Cédric Villani. *Topics in optimal transportation*, volume 58. American Mathematical Soc., 2021.

Cédric Vincent-Cuaz, Rémi Flamary, Marco Corneli, Titouan Vayer, and Nicolas Courty. Semi-relaxed gromov–wasserstein divergence with applications on graphs. In *International Conference on Learning Representations (ICLR)*, 2022. URL https://arxiv.org/abs/2110.02753. arXiv:2110.02753.

S Vichy N Vishwanathan, Nicol N Schraudolph, Risi Kondor, and Karsten M Borgwardt. Graph kernels. *The Journal of Machine Learning Research*, 11:1201–1242, 2010.

L Wang, WP Sousa, and P Gong. Integration of object-based and pixel-based classification for mapping mangroves with ikonos imagery. *International journal of remote sensing*, 25(24):5655–5668, 2004.

Wei Wang, Dejan Slepčev, Saurav Basu, John A Ozolek, and Gustavo K Rohde. A linear optimal transportation framework for quantifying and visualizing variations in sets of images. *International journal of computer vision*, 101(2):254–269, 2013.

Hongteng Xu, Dixin Luo, Hongyuan Zha, and Lawrence Carin. Gromov-wasserstein learning for graph matching and node embedding. In *Proceedings of the International Conference on Machine Learning (ICML)*, volume 97, pp. 6332–6341, 2019. doi: 10.5555/3327144.3327224. URL https://proceedings.mlr.press/v97/xu19b.html.

Si Zhang and Hanghang Tong. Attributed network alignment: Problem definitions and fast solutions. *IEEE Transactions on Knowledge and Data Engineering*, 31(9):1680–1692, 2019. doi: 10.1109/TKDE.2018.2866440.

# A   NOTATION AND ABBREVIATIONS.

## Abbreviations

- $OT$: Optimal Transport problem; see (20).
- $POT$: Partial Optimal Transport problem; see (22).
- $GW$: Gromov–Wasserstein problem; see (1).
- $PGW$: Partial Gromov–Wasserstein problem; see (3).
- $MPGW$: Mass-Constrained Partial Gromov–Wasserstein; see (4).
- FW: Frank–Wolfe algorithm Frank et al. (1956).
- $EFUGW(\mathbb{X}, \mathbb{Y})$: entropic fused unbalanced GW objective (cf. (87)).
- $EFPGW(\mathbb{X}, \mathbb{Y})$: entropic fused *partial* GW objective (cf. (89)).
- $LB\text{-}FPGW_\lambda(\mathbb{X}, \mathbb{Y})$: relaxed lower-bound formulation (cf. (90)).
- $EFMPGW(\mathbb{X}, \mathbb{Y})$: entropic fused mass-constrained PGW (cf. (93)); $LB\text{-}EFMPGW$ its relaxation.
- $c(x, y)$: feature cost; $d_X, d_Y$: intra-space distances; $|d_X - d_Y|^2$: structural discrepancy.

## Sets, Spaces, and Indices

- $\mathbb{R}^d$: Euclidean space; $\mathbb{R}_+$: nonnegative reals.
- $X, Y \subset \mathbb{R}^d$: non-empty, convex, compact sets (default setting).
- $X^2 = X^{\otimes 2} = X \times X$.
- $[1{:}n] = \{1, \ldots, n\}$.
- $r, p, q \in [1, \infty)$: real exponents used in costs/metrics.

## Vector, Norms and Basic Operators

- $\|\cdot\|$: Euclidean norm
- $|\mu|$: total mass of a measure $\mu$. That is $|\mu| = \mu(\Omega)$, where $\mu$ is defined on $\Omega$.
- $\|\cdot\|_{\text{TV}}, \|\cdot\|_{\text{TV}}$: total variation norm. In particular, given a signed measure $\mu = \mu_+ - \nu_-$, where positive measures $\mu_+, \mu_-$ are the unique Jordan measure decomposition of $\mu$. Then

$$\|\mu\|_{TV} = |\mu_+ + \mu_v|$$

- $\langle A, B \rangle = \text{tr}(A^\top B)$: Frobenius inner product.
- $1_n, 1_{n \times m}, 1_{n \times m \times n \times m}$: all-ones vector, matrix, and tensor.
- $\mathbb{1}_E$: indicator of a measurable set $E$, $\mathbb{1}_E(z) = 1$ if $z \in E$, else 0.
- $\nabla$: gradient.

## Measures and Pushforwards

- $\mathcal{M}_+(X)$: finite nonnegative Radon measures on $X$;
- $\mathcal{P}_2(X)$: probability measures on $X$ with finite second moment.
- $|\mu| = \mu(X)$: total mass (TV-norm) of $\mu$.
- $\mu \leq \sigma$: measure domination, i.e., $\mu(B) \leq \sigma(B)$ for all Borel $B \subseteq X$.
- $\mu^{\otimes 2} = \mu \otimes \mu$: product measure.
- $\mu(\phi) := \langle \phi, \mu \rangle := \int \phi(x)\, d\mu(x)$.
- $T_\# \sigma$: pushforward of $\sigma$ by measurable $T : X \to Y$, i.e., $T_\# \sigma(A) = \sigma(T^{-1}(A))$.

**Metric-Measure (mm) Spaces and Distance Matrices**

- $\mathbb{X} = (X, d_X, \mu)$, $\mathbb{Y} = (Y, d_Y, \nu)$: mm-spaces.
- For discrete $\mathbb{X}$ with $X = \{x_i\}_{i=1}^n$, define $C^X \in \mathbb{R}^{n \times n}$ by $C_{i,i'}^X = d_X^q(x_i, x_{i'})$; similarly $C^Y$.
- $\mathbb{X} \sim \mathbb{Y}$: mm-space equivalence if they have equal total mass and $GW_q^p(\mathbb{X}, \mathbb{Y}) = 0$.

**Couplings and Partial Couplings**

- $\Gamma(\mu, \nu) := \{\gamma \in \mathcal{P}_2(X \times Y) : \gamma_1 = \mu, \gamma_2 = \nu\}$.
- Discrete weights: $\mathrm{p} = [p_1^X, \ldots, p_n^X]^\top$, $\mathrm{q} = [q_1^Y, \ldots, q_m^Y]^\top$, $|\mathrm{p}| = \sum_i p_i$, $\mathrm{p} \le \mathrm{p}'$ if $p_j \le p_j' \; \forall j$.
- $\Gamma(\mathrm{p}, \mathrm{q}) = \{\gamma \in \mathbb{R}_+^{n \times m} : \gamma 1_m = \mathrm{p}, \gamma^\top 1_n = \mathrm{q}\}$.
- $\Gamma_{\le}(\mu, \nu) := \{\gamma \in \mathcal{M}_+(X \times Y) : \gamma_1 \le \mu, \gamma_2 \le \nu\}$.
- $\Gamma_{\le}(\mathrm{p}, \mathrm{q}) := \{\gamma \in \mathbb{R}_+^{n \times m} : \gamma 1_m \le \mathrm{p}, \gamma^\top 1_n \le \mathrm{q}\}$.
- $\gamma, \gamma_1, \gamma_2$: joint and marginal measures; in discrete form $\gamma \in \mathbb{R}_+^{n \times m}$, $\gamma_1 = \gamma 1_m$, $\gamma_2 = \gamma^\top 1_n$.
- $\pi_1 : X \times Y \to X$, $\pi_2 : X \times Y \to Y$.
- $\pi_{1,2} : S \times X \times Y \to X \times Y$, $(s, x, y) \mapsto (x, y)$; similarly $\pi_{0,1}, \pi_{0,2}$ for other coordinate pairs.

**Optimal Transport Problems**

- $c : X \times Y \to \mathbb{R}_+$: lower semicontinuous cost for (partial) OT.
- $OT(\mu, \nu)$: classical OT; $W_p(\mu, \nu)$: $p$-Wasserstein distance; $POT_\lambda(\mu, \nu)$: partial OT with parameter $\lambda > 0$; see (20), (21), (22).
- $L : \mathbb{R} \times \mathbb{R} \to \mathbb{R}$, $D : \mathbb{R} \times \mathbb{R} \to \mathbb{R}$: GW loss and scalar distance.
- $GW^L(\cdot, \cdot)$: GW objective with loss $L$; $d_{GW,r}^q$: GW with $L(a, b) = |a - b|^q$; see (1).
- $PGW_\lambda(\cdot, \cdot)$: partial GW objective; see (3).
- $C(\gamma; \lambda, \mu, \nu) := \gamma^{\otimes 2}\big(L(d_X^q, d_Y^q)\big) + \lambda\big(|\mu|^2 + |\nu|^2 - 2|\gamma|^2\big)$: PGW transport cost for $\gamma \in \Gamma_{\le}(\mu, \nu)$.
- $UGW_\lambda(\mathbb{X}, \mathbb{Y})$: unbalanced GW; $FUGW_\lambda(\mathbb{X}, \mathbb{Y})$: fused unbalanced GW.
- $d_{FGW}(\mathbb{X}, \mathbb{Y})$: fused GW distance; $d_{FPGW,\lambda}^p(\mathbb{X}, \mathbb{Y})$: fused partial GW distance.

**Discrete Tensorized Forms**

- Discrete measure setting: $\mu = \sum_{i=1}^n \mathrm{p}_i \delta_{x_i}$, $\nu = \sum_{j=1}^m \mathrm{q}_j \delta_{y_j}$.
- $M \in \mathbb{R}^{n \times m \times n \times m}$ with $M_{i,j,i',j'} = L(C_{i,i'}^X, C_{j,j'}^Y)$; $M^\top$ swaps index pairs: $M_{i,j,i',j'}^\top = M_{i',j',i,j}$.
  In default, $L$ is L2 norm square, and we denote $M = \|C_X - C_Y\|^2$
- $(M - 2\lambda)_{i,j,i',j'} := M_{i,j,i',j'} - 2\lambda$.
- $M \circ \gamma \in \mathbb{R}^{n \times m}$ with $[M \circ \gamma]_{i,j} = \sum_{i',j'} M_{i,j,i',j'} \gamma_{i',j'}$.
- $\langle \cdot, \cdot \rangle_F$: Frobenius inner product on $\mathbb{R}^{n \times m}$.

**Frank-Wolfe Optimization and Algorithmic Symbols**

- $\mathcal{L}$: objective functional for $PGW_\lambda(\cdot, \cdot)$.
- $\alpha \in [0, 1]$: line-search step size.
- $\gamma^{(1)}$: initialization; $\gamma^{(k)}, \gamma^{(k)'}$: transport plans before/after Step 1 in the $k$-th FW iteration.

- $G_{C,M} = \omega_1 C + \omega_2 \tilde{M} + M^\top \circ \gamma$, $\quad G = 2\hat{M} \circ \hat{\gamma}$: gradients in two FW variants.

- $a, b, c \in \mathbb{R}$: coefficients defined in (65) (For FPGW, replace $M$ by $M - 2\lambda$).

- $MPGW_\rho$ and $\Gamma^\rho_\leq(\mu, \nu)$: mass-constrained partial GW and its feasible set; see (4), (6).

**Entropic / Sinkhorn Notation**

- $\epsilon > 0$: entropic regularization parameter.

- $D_\phi(\cdot \| \cdot)$: $\phi$-divergence; special cases below.

- $H(\mu \| \nu) = \bar{H}(\mu \| \nu) + |\nu| - |\mu|$ with $\bar{H}(\mu \| \nu) = \int \log\left(\frac{d\mu}{d\nu}\right) d\mu$ when $\mu \ll \nu$, $+\infty$ otherwise.

- $D_{PTV}(\mu \| \nu) = \begin{cases} \|\mu - \nu\|_{\mathrm{TV}} = |\nu - \mu| & \text{if } \mu \leq \nu, \\ +\infty & \text{otherwise.} \end{cases}$

- $\Gamma_\leq(\mu, \nu)$, $\Gamma^\rho_\leq(\mu, \nu)$: partial feasible sets (mass-unconstrained and mass-constrained with $|\gamma| = \rho$).

- $|\gamma| = \gamma(X \times Y)$, $|\mu| = \mu(X)$, $|\nu| = \nu(Y)$: total mass.

- $\mathcal{L}(\gamma)$: fused GW functional (feature + structure + penalties) used in EFUGW/EFPGW.

- $c_\pi(x, y) = \frac{1}{2}\omega_1 d(x, y) + \omega_2 \left[L(d^r_X, d^r_Y) \circ \pi\right](x, y) + \epsilon \bar{H}(\pi \| \mu \otimes \nu)$: $\pi$-conditioned cost (cf. Prop. I.1).

- $[L(d_X, d_Y) \circ \pi](x, y) = \int_{X \times Y} L(d^r_X(x, x'), d^r_Y(y, y')) \, d\pi(x', y')$.

- $K = \exp(-c/\epsilon)$: Gibbs kernel for feature cost; elementwise exponential.

- $u \in \mathbb{R}^n_+$, $v \in \mathbb{R}^m_+$: Sinkhorn scaling vectors.

- $\odot, \oslash$: elementwise (Hadamard) product and division.

- $\mathrm{diag}(a)$: diagonal matrix with vector $a$ on the diagonal.

- $\mathrm{Proj}^{KL}_{\mathcal{C}_i}$: Bregman (KL) projection onto constraint set $\mathcal{C}_i$ (cf. Alg. 6).

- $\mathcal{C}_1 = \{\gamma \geq 0 : \gamma_2 \leq q\}$, $\mathcal{C}_2 = \{\gamma \geq 0 : \gamma_1 \leq p\}$, $\mathcal{C}_3 = \{\gamma \geq 0 : |\gamma| = \rho\}$: partial-marginal and mass constraints.

- $\gamma^{\otimes 2}$, $(\mu \otimes \nu)^{\otimes 2}$: product measures on $(X \times Y)^2$.

- $\pi, \gamma \in \mathcal{M}_+(X \times Y)$: current and updated couplings in alternating minimization; $\rho \in [0, \min(|p|, |q|)]$ target mass.

- Stopping criteria: norms/duality gap on $(u, v)$ or fixed-point tolerance on $\gamma$.

**Graph-Specific Notation**

- $G = (V, E)$: graph with nodes $V = \{v_1, \ldots, v_N\}$ and edges $E \subset V^2$.

- $d_V : V^2 \to \mathbb{R}$: structural distance (default: shortest-path distance).

- $f : V \to \mathcal{F}$: node feature map; $\mathcal{F}$ is the feature space.

- $wl : \mathcal{F} \to S^H$: Weisfeiler–Lehman feature map Vishwanathan et al. (2010) for discrete $\mathcal{F}$; $S$ finite alphabet, $H \in \mathbb{N}$.

- $d_\mathcal{F}$: feature-space metric. Default: Euclidean if $\mathcal{F} = \mathbb{R}^d$; if $\mathcal{F}$ is discrete, Hamming on $wl(\mathcal{F})$:

$$d_\mathcal{F}\big(f(v_1), f(v_2)\big) := \sum_{h=1}^{H} \delta\big(wl(f(v_1))_h \neq wl(f(v_2))_h\big).$$

It is a lower semi-continuous function.

## B  BACKGROUND OF OPTIMAL TRANSPORT PROBLEMS

**The Optimal Transport (OT) problems** for $\mu, \nu \in \mathcal{P}(\Omega)$, with transportation cost $c(x, y) : \Omega \times \Omega \to \mathbb{R}_+$ being a lower-semi continuous function is defined as:

$$OT(\mu, \nu) := \min_{\gamma \in \Gamma(\mu, \nu)} \gamma(c), \tag{20}$$

$$\text{where} \quad \gamma(c) := \int_{\Omega^2} c(x, y) \, d\gamma(x, y)$$

and where $\Gamma(\mu, \nu)$ denotes the set of all joint probability measures on $\Omega^2 := \Omega \times \Omega$ with marginals $\mu, \nu$, i.e., $\gamma_1 := \pi_{1\#}\gamma = \mu, \gamma_2 := \pi_{2\#}\gamma = \nu$, where $\pi_1, \pi_2 : \Omega^2 \to \Omega$ are the canonical projections $\pi_1(x, y) := x, \pi_2(x, y) := y$. A minimizer for (20) always exists Villani (2009; 2021) and when $c(x, y) = \|x - y\|^p$, for $p \geq 1$, it defines a metric on $\mathcal{P}(\Omega)$, which is referred to as the "$p$-Wasserstein distance":

$$W_p^p(\mu, \nu) := \min_{\gamma \in \Gamma(\mu, \nu)} \int_{\Omega^2} \|x - y\|^p d\gamma(x, y). \tag{21}$$

**The Partial Optimal Transport (POT) problem** Chizat et al. (2018b); Figalli & Gigli (2010); Piccoli & Rossi (2014) extends the OT problem to the set of Radon measures $\mathcal{M}_+(\Omega)$, i.e., non-negative and finite measures. For $\lambda > 0$ and $\mu, \nu \in \mathcal{M}_+(\Omega)$, the POT problem is defined as:

$$POT(\mu, \nu; \lambda) := \inf_{\gamma \in \mathcal{M}_+(\Omega^2)} \gamma(c) + \lambda(|\mu - \gamma_1| + |\nu - \gamma_2|), \tag{22}$$

where, in general, $|\sigma|$ denotes the total variation norm of a measure $\sigma$, i.e., $|\sigma| := \sigma(\Omega)$. The constraint $\gamma \in \mathcal{M}_+(\Omega^2)$ in (22) can be further restricted to $\gamma \in \Gamma_\leq(\mu, \nu)$:

$$\Gamma_\leq(\mu, \nu) := \{\gamma \in \mathcal{M}_+(\Omega^2) : \gamma_1 \leq \mu, \gamma_2 \leq \nu\},$$

denoting $\gamma_1 \leq \mu$ if for any Borel set $B \subseteq \Omega$, $\gamma_1(B) \leq \mu(B)$ (respectively, for $\gamma_2 \leq \nu$) Figalli (2010). Roughly speaking, the linear penalization indicates that if the classical transportation cost exceeds $2\lambda$, it is better to create/destroy' mass (see Bai et al. (2023b) for further details). In addition, the above formulation has an equivalent form, to distinguish them, we call it "mass-constraint partial optimal transport problem" (MPOT):

$$MOPT(\mu, \nu; \rho) := \inf_{\gamma \in \Gamma_\leq^\rho(\mu, \nu)} \gamma(c). \tag{23}$$

## C  (FUSED)-UNBALANCED GROMOV WASSERSTEIN AND (FUSED) PARTIAL GROMOV WASSERSTEIN.

The unbalanced Gromov Wasserstein problem is firstly proposed by Séjourné et al. (2021); Chapel et al. (2020); Bai et al. (2024). Given two $mm-$spaces, $\mathbb{X} = (X, d_X, \mu), \mathbb{Y} = (Y, d_Y, \nu)$, the UGW problem is defined by

$$UGW_\lambda(\mathbb{X}, \mathbb{Y}) := \inf_{\gamma \in \mathcal{M}_+(X \times Y)} \langle L(d_X^r, d_Y^r), \gamma^{\otimes 2} \rangle + \lambda(D_{\phi_1}(\gamma_1^{\otimes 2} \| \mu^{\otimes 2}) + D_{\phi_2}(\gamma_2^{\otimes 2} \| \nu^{\otimes 2})), \tag{24}$$

where $D_{\phi_1}, D_{\phi_2}$ are f-divergence and can be distinct in the general setting.

Similar to this work, the fused-Unbalanced Gromov Wasserstein, proposed by Thual et al. (2022) is defined as:

$$FUGW_\lambda(\mathbb{X}, \mathbb{Y}) := \inf_{\gamma \in \mathcal{M}_+(X \times Y)} \omega_1 \langle C, \gamma \rangle + \omega_2 \langle L(d_X^r, d_Y^r), \gamma^{\otimes 2} \rangle + \lambda(D_{\phi_1}(\gamma_1^{\otimes 2} \| \mu^{\otimes 2}) + D_{\phi_2}(\gamma_2^{\otimes 2} \| \nu^{\otimes 2})).$$

And when the $f$-divergence terms $D_{\phi_1}, D_{\phi_2}$ are KL divergences, the authors in Thual et al. (2022) propose a Sinkhorn computational solver.

At the end of this section, we introduce the following relation between FUGW and FPGW, which is equivalent to the statement (1) in Theorem 3.3.

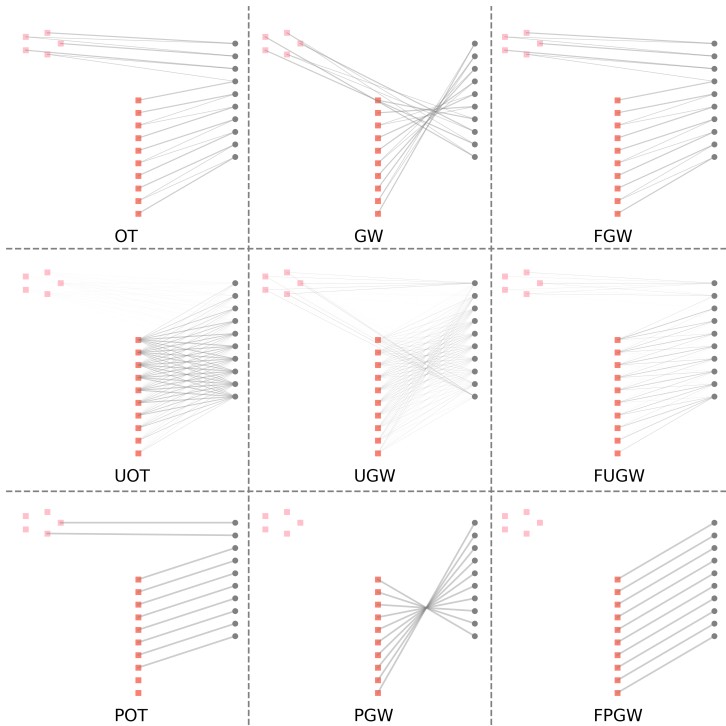

Figure 4: In this toy example, we illustrate the relationships and differences between OT, Unbalanced OT, Partial OT, GW, Unbalanced GW, Partial GW, fused GW, fused unbalanced GW, and Fused Partial GW. The source shape consists of the union of the pink and red points, and the target shape is the grey shape. The objective is to establish a reasonable match between the two shapes.

**Proposition C.1.** *When $D_{\phi_1}, D_{\phi_2}$ are total variation, then we have:*

$$FUGW_\lambda(\mathbb{X}, \mathbb{Y}) = FPGW_\lambda(\mathbb{X}, \mathbb{Y})$$
$$= \inf_{\gamma \in \Gamma_{\le}(\mu,\nu)} \omega_1 \langle C, \gamma \rangle + \omega_2 \langle L(d_X^r, d_Y^r), \gamma^{\otimes 2} \rangle + \lambda(|\mu|^2 + |\nu|^2 - 2|\gamma|^2).$$

*Equivalently speaking, one can restrict the searching space from $\mathcal{M}_+(X \times Y)$ to $\Gamma_{\le}(\mu, \nu)$.*

*Proof.* Choose $\gamma \in \mathcal{M}_+(X \times Y)$ and let $\gamma_{Y|X}(\cdot|x)$ denote the conditional measure of $\gamma$ given $x \in X$. Let $\gamma' = \gamma_{Y|X}(\cdot|X) \cdot (\gamma_1 \wedge \mu)$. From the definition, we have $\gamma' \le \gamma, \gamma_1' \le \mu$. From Inequality (22) in Bai et al. (2023c), we have

$$\omega_2 \langle \|d_X - d_Y\|^p, \gamma'^{\otimes 2} \rangle + \lambda(\|\mu - \gamma_1'\|_{TV} + \|\nu - \gamma_2'\|_{TV}) \le \omega_2 \langle \|d_X - d_Y\|^p, \gamma^{\otimes 2} \rangle + \lambda(\|\mu - \gamma_1\|_{TV} + \|\nu - \gamma_2\|_{TV}).$$

Since $C \ge 0, \gamma' \le \gamma$, we have $\langle C, \gamma' \rangle \le \langle C, \gamma \rangle$. Therefore, we have

$$\omega_1 \langle C, \gamma' \rangle + \omega_2 \langle \|d_X - d_Y\|^p, \gamma'^{\otimes 2} \rangle + \lambda(\|\mu - \gamma_1'\|_{TV} + \|\nu - \gamma_2'\|_{TV})$$
$$\le \omega_1 \langle C, \gamma \rangle + \omega_2 \langle \|d_X - d_Y\|^p, \gamma^{\otimes 2} \rangle + \lambda(\|\mu - \gamma_1\|_{TV} + \|\nu - \gamma_2\|_{TV}).$$

That is, based on $\gamma$, we can construct $\gamma'$ such that $\gamma_1' \le \mu \wedge \gamma_1, \gamma' \le \gamma$ and the $\gamma'$ admits smaller cost. Based on the same process, based on $\gamma'$, we can construct $\gamma''$ such that $\gamma'' \le \gamma', \gamma'' \le \nu$. That is $\gamma'' \in \Gamma_{\le}(\mu, \nu)$ and admits smaller cost than $\gamma'$. Therefore, we can restrict the searching space for (8) in this case from $\mathcal{M}_+(X \times Y)$ to $\Gamma_{\le}(X \times Y)$. $\square$

**Remark C.2.** *In Figure 4, we illustrate the transportation plans of various methods, including optimal transport (OT), unbalanced optimal transport (UOT), partial optimal transport (POT), . . . and fused-Partial Gromov-Wasserstein (FPGW).*

*Due to the balanced mass setting, OT, GW, and fused GW match all points from the source shape to the target shape. In contrast, the Sinkhorn entropic regularization in UOT, UGW, and FUGW allows*

*for mass splitting, resulting in a small amount of mass between the grey shape and the pink points. PGW and fused PGW achieve the desired one-to-one matching, where the straight-line segment in the source shape (red points) corresponds to the straight-line segment in the target shape (grey points). However, since PGW does not account for the spatial location of points, there is a 50% chance of obtaining "anti-identity" matching as demonstrated in Figure 4.*

### C.1   EXISTENCE OF MINIMIZER OF FUSED PGW

In this section, we discuss the basic properties of the fused PGW.

**Proposition C.3.** *Given mm-spaces* $\mathbb{X} = (X, d_X, \mu), \mathbb{Y} = (Y, d_Y, \nu)$, *where* $X, Y$ *are compact sets, the fused-PGW problems (11) and (10) admit minimizers. That is, we can replace* inf *by* min *in the formulations.*

Note, this proposition is equivalent to the statement (2) in Theorem 3.3.

We first introduce the following statements:

**Proposition C.4.** *The set* $\Gamma_{\leq}(\mu, \nu)$ *and* $\Gamma_{\leq}^{\rho}(\mu, \nu)$ *are weakly compact, convex sets, where* $\rho \in [0, \min(|\mu|, |\nu|)]$.

*Proof.* By Liu et al. (2023b) Lemma B.1, we have $\Gamma_{\leq}(\mu, \nu)$ is a weakly compact set. By e.g. Caffarelli & McCann (2010); Bai et al. (2023b), we have $\Gamma_{\leq}(\mu, \nu)$ is convex.

It remains to show the compactness and convexity of the set $\Gamma_{\leq}^{\rho}(\mu, \nu)$.

Pick $\gamma^1, \gamma^2 \in \Gamma_{\leq}^{\rho}(\mu, \nu)$ and $\omega \in [0, 1]$ and let $\gamma = (1 - \omega)\gamma^1 + \omega\gamma^2$. We have $\gamma \in \Gamma_{\leq}(\mu, \nu)$ by the convexity of $\Gamma_{\leq}(\mu, \nu)$. In addition,

$$|(1 - \omega)\gamma^1 + \omega\gamma^2| = (1 - \omega)|\gamma^1| + \omega|\gamma^2| \geq (1 - \omega)\rho + \omega\rho = \rho,$$

thus we have: $\gamma \in \Gamma_{\leq}^{\rho}(\mu, \nu)$.

Pick sequence $(\gamma^n)_{n=1}^{\infty} \subset \Gamma_{\leq}^{\rho}(\mu, \nu) \subset \Gamma_{\leq}(\mu, \nu)$. By the compactness of $\Gamma_{\leq}(\mu, \nu)$, there exists a subsequence $(\gamma^{n_k})_{k=1}^{\infty}$ that converges to some $\gamma^* \in \Gamma_{\leq}(\mu, \nu)$ weakly. It remains to show $\gamma^* \in \Gamma_{\leq}^{\rho}(\mu, \nu)$. Indeed,

$$|\gamma^*| = \langle 1_{X \times Y} \times \gamma \rangle = \lim_{k \to \infty} \langle 1_{X \times Y}, \gamma^{n_k} \rangle = \lim_{k \to \infty} |\gamma^{n_k}|.$$

Since for each $k \geq 1$, we have $|\gamma^{n_k}| \geq \rho$, thus $|\gamma^*| \geq \rho$. That is $\gamma^* \in \Gamma_{\leq}^{\rho}(\mu, \nu)$, and we complete the proof. $\square$

**Lemma C.5.** *Suppose* $X$ *and* $Y$ *are compact sets. Let* $Z = (X \times Y)$ *and* $d_Z((x^1, y^1), (x^2, y^2)) = d_X(x^1, x^2) + d_Y(y^1, y^2)$. *Suppose* $\phi : Z^2 \to \mathbb{R}$ *is Lipschitz continuous and* $C : Z \to \mathbb{R}$ *is bounded, then the following problem admits a minimizer:*

$$\inf_{\gamma \in \mathcal{M}} \langle \phi, \gamma^{\otimes 2} \rangle + \langle C, \gamma \rangle \tag{25}$$

*where* $\mathcal{M}$ *is a non-empty weakly compact set.*

*Proof.* Pick weakly convergent $(\gamma^n)_{n=1}^{\infty} \subset \mathcal{M}$, we have $\gamma^n \xrightarrow{w} \gamma^* \in \mathcal{M}$.

From lemma C.3 in Bai et al. (2024), we have

$$\langle \phi, (\gamma^n)^{\otimes 2} \rangle \to \langle \phi, \gamma^* \rangle.$$

By definition of weak convergence and the fact that the sets $X, Y$ are compact, we have

$$\langle C, \gamma^n \rangle \to \langle C, \gamma \rangle. \tag{26}$$

Thus, we have

$$\langle \phi, (\gamma^n)^{\otimes 2} \rangle + \langle C, \gamma^n \rangle \to \langle \phi, \gamma^{\otimes 2} \rangle + \langle C, \gamma \rangle.$$

Choose a sequence $\gamma^n \in \mathcal{M}$ such that $\langle \phi, (\gamma^n)^{\otimes 2} \rangle + \langle C, \gamma^n \rangle$ achieves the infimum of the problem (25). By compactness of $\mathcal{M}$, there exists a convergent subsequence $\gamma^{(n_k)} \xrightarrow{w} \gamma^* \in \mathcal{M}$. By (26), we have $\gamma^*$ is a minimizer of (25). Thus, we complete the proof. $\square$

*Proof of Proposition C.3.* From Lemma C.1. in Bai et al. (2024), we have

$$(X \times Y)^2 \ni ((x^1, y^1), (x^2, y^2)) \rightarrow |d_X^r(x^1, x^2) - d_Y^r(y^1, y^2)|^q$$

is Lipschitz continuous. In addition, $C : X \times Y$ is a bounded mapping. From lemma C.4, we have $\Gamma_{\leq}(\mu, \nu), \Gamma_{\leq}^{\rho}(\mu, \nu)$ are compact. From lemma C.5, we complete the proof. $\square$

# D  METRIC PROPERTY OF FUSED PGW

In this section, we discuss the proof of Theorem 3.3 (3). We will discuss the details in the following subsections. The related conclusion can be treated as an extension of Theorem 6.3 in Titouan et al. (2019b) and Theorem 3.1 in Vayer et al. (2020).

## D.1  BACKGROUND: ISOMETRY, FUSED-GW SEMI-METRIC.

Given $\mathbb{X} = (X, d_X, \mu), \mathbb{Y} = (Y, d_Y, \nu)$, we say $X, Y$ are equivalent, noted $X \sim Y$ if and only if:

There exists a mapping $\phi : X \rightarrow Y$ such that

- $\phi_{\#}\mu = \nu$
- $d_X(x, x') = d_Y(\phi(x), \phi(x')), \forall x, x' \in X.$

Such a function $\phi$ is called **measure preserving isometry**.

In addition, in the formulation of fused-GW (11), we set $d(x, y) = \|x - y\|^q$ and $L(\mathrm{r}_1, \mathrm{r}_2) = |\mathrm{r}_1 - \mathrm{r}_2|^q$. The reduced formulation is called "fused-Gromov Wasserstein distance" Vayer et al. (2020):

$$d_{FGW}(\mathbb{X}, \mathbb{Y}) := \inf_{\gamma \in \Gamma(\mu, \nu)} \int_{(X \times Y)^2} \omega_1 \frac{\|x - y\|^q}{|\gamma|} + \omega_2 |d_X^r(x, x') - d_Y^r(y, y')|^q d\gamma(x, y) d\gamma(x', y'). \tag{27}$$

and it defines a metric where the above equivalence relation induces the identity.

Inspired by this formulation, we introduce the **Fused Partial GW metric**:

$$d_{FPGW,\lambda}^{\rho}(\mathbb{X}, \mathbb{Y}) := \inf_{\gamma \in \Gamma_{\leq}(\mu, \nu)} \int_{(X \times Y)^2} \omega_1 \frac{\|x - y\|^q}{|\gamma|} + \omega_2 |d_X^r(x, x') - d_Y^r(y, y')|^q + \lambda \left( \frac{|\mu|^2}{|\gamma|^2} + \frac{|\nu|^2}{|\gamma|^2} - 2 \right) d\gamma(x, y) d\gamma(x', y'). \tag{28}$$

**Remark D.1.** *We adapt the convention $\frac{1}{0} \cdot 0 = 1$, thus the above formulation is well-defined. In particular, when $|\gamma| = 0$, i.e., $\gamma$ is zero measure, the above integration is defined by*

$$\int_{(X \times Y)^2} \omega_1 \frac{\|x - y\|^q}{|\gamma|} + \omega_2 |d_X^r(x, x') - d_Y^r(y, y')|^q + \lambda \left( \frac{|\mu|^2}{|\gamma|^2} + \frac{|\nu|^2}{|\gamma|^2} - 2 \right) d\gamma(x, y) d\gamma(x', y')$$

$$= \lambda(|\mu|^2 + |\nu|^2) \tag{29}$$

$$= \lim_{|\gamma| \searrow 0} \int_{(X \times Y)^2} \omega_1 \frac{\|x - y\|^q}{|\gamma|} + \omega_2 |d_X^r(x, x') - d_Y^r(y, y')|^q + \lambda \left( \frac{|\mu|^2}{|\gamma|^2} + \frac{|\nu|^2}{|\gamma|^2} - 2 \right) d\gamma(x, y) d\gamma(x', y').$$

**Remark D.2.** *By Proposition C.3, the above problem admits a minimizer.*

Next, we introduce the formal statement of Theorem 3.3 (2).

**Theorem D.3.** *Define space for mm-spaces,*

$$\mathcal{G} = \{\mathbb{X} = (X, d_X, \mu), X \subset \mathbb{R}^d, X \text{ is compact}; d_X \text{ is a metric}; \mu \in \mathcal{M}_+(X)\}.$$

*Then Fused PGW (28) defines a semi-metric in quotient space $\mathcal{G}/\sim$. In particular:*

   *1. $d_{FPGW,\lambda}(\cdot, \cdot)$ is non-negative and symmetric.*

2. *Suppose $\omega_2, \lambda > 0$, then $d_{FPGW,\lambda}(\mathbb{X}, \mathbb{Y}) = 0$ iff $\mathbb{X} \sim \mathbb{Y}$.*

3. *If $\omega_2 > 0$, for $q \geq 1$, we have*

$$d_{FPGW,\lambda}(\mathbb{X}, \mathbb{Y}) \leq 2^{q-1}(d_{FPGW,\lambda}(\mathbb{X}, \mathbb{Z}) + d_{FPGW,\lambda}(\mathbb{Y}, \mathbb{Z})). \tag{30}$$

*In particular, when $q = 1$, fused-PGW satisfies the triangle inequality.*

### D.2 Proof of the (1)(2) in Theorem D.3

For statement (1), by definition, for each $\gamma \in \Gamma_{\leq}(\mu, \nu)$, we have

$$\int_{X \times Y} \|x - y\|^q d\gamma, \int_{(X \times Y)^2} \|d_X^r(x, x') - d_Y^r(y, y')\|^q d\gamma^{\otimes 2} \geq 0.$$

Thus, $d_{FPGW,\lambda}(\mathbb{X}, \mathbb{Y}) \geq 0$.

In addition,

$d_{FPGW,\lambda}(\mathbb{X}, \mathbb{Y})$

$= \displaystyle\inf_{\gamma \in \Gamma_{\leq}(\mu, \nu)} \int_{(X \times Y)^2} \omega_1 \|x - y\|^q + \omega_2 |d_X^r(x, x') - d_Y^r(y, y')|^q)^p d\gamma(x, y) d\gamma(x', y') + \lambda(|\mu|^2 + |\nu|^2 - 2|\gamma|^2)$

$= \displaystyle\inf_{\gamma \in \Gamma_{\leq}(\nu, \mu)} \int_{(Y \times X)^2} \omega_1 \|y - x\|^q + \omega_2 |d_Y^r(y, y') - d_X^r(x, x')|^q)^p d\gamma(y, x) d\gamma(y', x') + \lambda(|\mu|^2 + |\nu|^2 - 2|\gamma|^2)$

$= d_{FPGW,\lambda}(\mathbb{Y}, \mathbb{X})$.

And we prove the symmetric.

For statement (2), suppose a measure-preserving isometry $\phi : X \to Y$ exists, then we have $|\mu| = |\nu|$. Let $\gamma = (\text{id} \times \phi)_{\#}\mu$, we have $\gamma \in \Gamma(\mu, \nu) \subset \Gamma_{\leq}(\mu, \nu)$. Thus $|\gamma| = |\mu| = |\nu|$.

Furthermore,

$d_{FPGW,\lambda}^p(\mathbb{X}, \mathbb{Y})$

$\leq \displaystyle\int_{(X \times Y)^2} \omega_1 \|x - y\|^q + \omega_2 |d_X^r(x, x') - d_Y^r(y, y')|^q d\gamma(x, y) d\gamma(x', y') + \lambda(|\mu|^2 + |\nu|^2 - 2|\gamma|^2)$

$= \displaystyle\int_{(X \times Y)^2} \omega_1 \underbrace{\|x - \phi(x)\|^q}_{0} + \omega_2 \underbrace{|d_X^r(x, x') - d_Y^r(\phi(x), \phi(x'))|^q}_{0} d\mu(x) d\mu(x') + \lambda(\underbrace{|\mu|^2 + |\nu|^2 - 2|\gamma|^2}_{0})$

$= 0. \tag{31}$

For the other direction, suppose $d_{FPGW,\lambda}(\mathbb{X}, \mathbb{Y}) = 0$. We have two cases:

Case 1: $|\mu| = |\nu| = 0$. We've done.

Case 2: $|\mu| > 0$ or $|\nu| > 0$.

By the Proposition C.3, a minimizer for problem (28) exists. Choose one minimizer, denoted as $\gamma^*$.

First, we claim $|\mu| = |\nu| = |\gamma^*|$.

Indeed, assume the above equation is not true. For convenience, we suppose $|\mu| < |\nu|$. Then $|\gamma| \leq |\mu| < |\nu|$. We have

$$0 = d_{FPGW,\lambda}(\mathbb{X}, \mathbb{Y}) \geq \lambda(|\mu|^2 + |\nu|^2 - 2|\gamma^*|^2) > \lambda(|\mu|^2 - |\gamma^*|^2) \geq 0, \tag{32}$$

since $\lambda > 0$. Thus we have a contradiction.

Since $\gamma^* \in \Gamma_\leq(\mu, \nu)$, we have $\gamma^* \in \Gamma(\mu, \nu)$. Thus, we have

$$0 \leq d_{FPGW}(\mathbb{X}, \mathbb{Y})$$

$$\leq \int_{(X \times Y)^2} \omega_1 \frac{\|x - y\|^q}{|\gamma|} + \omega_2 |d_X^r(x, x') - d_Y^r(y, y')|^q \, d\gamma^*(x, y) d\gamma^*(x', y')$$

$$= \int_{(X \times Y)^2} \omega_1 \frac{\|x - y\|^q}{|\gamma|} + \omega_2 |d_X^r(x, x') - d_Y^r(y, y')|^q + \lambda \left( \frac{|\mu|^2}{|\gamma|^2} + \frac{|\nu|^2}{|\gamma|^2} - 2 \right) \, d\gamma^*(x, y) d\gamma^*(x', y')$$

$$= d_{FPGW}(\mathbb{X}, \mathbb{Y})$$

$$= 0.$$

That is, $d_{FPGW}(\mathbb{X}, \mathbb{Y}) = 0$. Since $\omega_2 > 0$, by Proposition 5.2 in Vayer et al. (2020) or Theorem 6.4 in Titouan et al. (2019b), there exists measure preserving isometry $\phi : X \to Y$ and thus we have $\mathbb{X} \sim \mathbb{Y}$ and we complete the proof.

## D.3 PROOF OF STATEMENT (3) IN THEOREM D.3.

Choose mm-spaces $\mathbb{S} = (S, d_S, \sigma), \mathbb{X} = (X, d_X, \mu), \mathbb{Y} = (Y, d_Y, \nu)$. In this section, we will prove the triangle inequality

$$d_{FPGW,\lambda}(\mathbb{X}, \mathbb{Y}) \leq d_{FPGW,\lambda}(\mathbb{S}, \mathbb{X}) + d_{FPGW,\lambda}(\mathbb{S}, \mathbb{Y}).$$

First, introduce auxiliary points $\hat{\infty}_0, \hat{\infty}_1, \hat{\infty}_2$ and set

$$\begin{cases} \hat{S} & = S \cup \{\hat{\infty}_0, \hat{\infty}_1, \hat{\infty}_2\}, \\ \hat{X} & = X \cup \{\hat{\infty}_0, \hat{\infty}_1, \hat{\infty}_2\}, \\ \hat{Y} & = Y \cup \{\hat{\infty}_0, \hat{\infty}_1, \hat{\infty}_2\}. \end{cases}$$

Define $\hat{\sigma}, \hat{\mu}, \hat{\nu}$ as follows:

$$\begin{cases} \hat{\sigma} & = \sigma + |\mu|\delta_{\hat{\infty}_1} + |\nu|\delta_{\hat{\infty}_2}, \\ \hat{\mu} & = \mu + |\sigma|\delta_{\hat{\infty}_0} + |\nu|\delta_{\hat{\infty}_2}, \\ \hat{\nu} & = \nu + |\sigma|\delta_{\hat{\infty}_0} + |\mu|\delta_{\hat{\infty}_1}. \end{cases} \tag{33}$$

Next, we define $d_{\hat{S}} : \hat{S}^2 \to \mathbb{R} \cup \{\infty\}$ as follows:

$$d_{\hat{S}}(s, s') = \begin{cases} d_S(s, s') & \text{if } (s, s') \in S^2, \\ \infty & \text{elsewhere.} \end{cases} \tag{34}$$

Note, $d_{\hat{S}}(\cdot, \cdot)$ is not a rigorous metric in $\hat{S}$ since we allow $d_{\hat{S}} = \infty$, $d_{\hat{X}}, d_{\hat{Y}}$ are defined similarly.

Then, we define the following mapping $L_\lambda : (\mathbb{R} \cup \{\infty\}) \times (\mathbb{R} \cup \{\infty\}) \to \mathbb{R}_+$:

$$L_\lambda^q(\text{r}_1, \text{r}_2) = \begin{cases} |\text{r}_1 - \text{r}_2|^q & \text{if } \text{r}_1, \text{r}_2 < \infty, \\ \lambda/\omega_2 & \text{if } \text{r}_1 = \infty, \text{r}_2 < \infty \text{ or vice versa,} \\ 0 & \text{if } \text{r}_1 = \text{r}_2 = \infty; \end{cases} \tag{35}$$

and mapping $\hat{D} : \mathbb{R}^d \cup \{\hat{\infty}_i : i \in [0:2]\} \to \mathbb{R}$:

$$\hat{D}^q(x, y) := \begin{cases} \|x - y\|^q & \text{if } x, y \in \mathbb{R}^d \\ 0 & \text{elsewhere.} \end{cases} \tag{36}$$

Finally, we define the following mappings:

$$\Gamma_\leq(\sigma, \mu) \ni \gamma^{01} \mapsto \hat{\gamma}^{01} \in \Gamma(\hat{\sigma}, \hat{\mu}),$$
$$\hat{\gamma}^{01} := \gamma^{01} + (\sigma - \gamma_1^{01}) \otimes \delta_{\hat{\infty}_0} + \delta_{\hat{\infty}_1} \otimes (\mu - \gamma_2^{01}) + |\gamma|\delta_{\hat{\infty}_1, \hat{\infty}_0} + |\nu|\delta_{\hat{\infty}_2, \hat{\infty}_2};$$
$$\Gamma_\leq(\sigma, \nu) \ni \gamma^{02} \mapsto \hat{\gamma}^{02} \in \Gamma(\hat{\sigma}, \hat{\nu}),$$
$$\hat{\gamma}^{02} := \gamma^{02} + (\sigma - \gamma_1^{02}) \otimes \delta_{\hat{\infty}_0} + \delta_{\hat{\infty}_2} \otimes (\nu - \gamma_2^{02}) + |\gamma|\delta_{\hat{\infty}_2, \hat{\infty}_0} + |\mu|\delta_{\hat{\infty}_1, \hat{\infty}_1};$$
$$\Gamma_\leq(\mu, \nu) \ni \gamma^{12} \mapsto \hat{\gamma}^{12} \in \Gamma(\hat{\mu}, \hat{\nu}),$$
$$\hat{\gamma}^{12} := \gamma^{12} + (\mu - \gamma_1^{12}) \otimes \delta_{\hat{\infty}_1} + \delta_{\hat{\infty}_2} \otimes (\nu - \gamma_2^{12}) + |\gamma|\delta_{\hat{\infty}_2, \hat{\infty}_1} + |\mu|\delta_{\hat{\infty}_0, \hat{\infty}_0}. \tag{37}$$

**Remark D.4.** *It is straightforward to verify that the above mappings are well-defined. In addition, we can observe that, for each $\gamma^{01} \in \Gamma_{\leq}(\sigma, \mu), \gamma^{02} \in \Gamma_{\leq}(\sigma, \nu), \gamma^{12} \in \Gamma_{\leq}(\mu, \nu)$,*

$$\hat{\gamma}^{01}(\{\hat{\infty}_2\} \times X) = \hat{\gamma}^{01}(S \times \{\hat{\infty}_2\}) = 0, \tag{38}$$

$$\hat{\gamma}^{02}(\{\hat{\infty}_1\} \times Y) = \hat{\gamma}^{02}(S \times \{\hat{\infty}_1\}) = 0, \tag{39}$$

$$\hat{\gamma}^{12}(\{\hat{\infty}_0\} \times Y) = \hat{\gamma}^{12}(X \times \{\hat{\infty}_0\}) = 0.$$

Based on these concepts, we define the following fused-GW variant problem:

$$\hat{d}_{FGW,\lambda}(\hat{\mathbb{X}}, \hat{\mathbb{Y}}) := \inf_{\hat{\gamma} \in \Gamma(\hat{\mu}, \hat{\nu})} \int_{(\hat{X} \times \hat{Y})^2} \omega_1 \frac{1}{c} \hat{D}^q(x, y) + \omega_2 L_\lambda^q(d_{\hat{X}}^r(x, x'), d_{\hat{Y}}^r(y, y')) d\hat{\gamma}(x, y) d\hat{\gamma}(x', y'). \tag{40}$$

where constant $c = |\sigma| + |\mu| + |\nu|$.

**Proposition D.5.** *For each $\gamma^{12} \in \Gamma(\mu, \nu)$, construct $\hat{\gamma}^{12} \in \Gamma(\hat{\mu}, \hat{\nu})$, we have:*

$$\int_{(X \times Y)^2} \left( \omega_1 \frac{\|x - y\|^q}{|\gamma^{12}|} + \omega_2 |d_X^r(x, x') - d_Y^r(y, y')|^q + \lambda \left( \frac{|\mu|}{|\gamma^{12}|} + \frac{|\nu|}{|\gamma^{12}|} - 2 \right) \right) d(\gamma^{12})^{\otimes 2}$$

$$= \int_{(\hat{X} \times \hat{Y})^2} \left( \omega_1 \frac{D^q(x, y)}{c} + \omega_2 L_\lambda^q(d_{\hat{X}}(x, x'), d_{\hat{Y}}(y, y')) \right) d(\hat{\gamma}^{12})^{\otimes 2} \tag{41}$$

*Furthermore, we have:*

$$d_{FPGW,\lambda}(\mathbb{X}, \mathbb{Y}) = \hat{d}_{FGW,\lambda}(\hat{\mathbb{X}}, \hat{\mathbb{Y}}). \tag{42}$$

*Proof.* We have:

$$\int_{(X \times Y)^2} \omega_1 \frac{\|x - y\|^q}{|\gamma^{12}|}, d(\gamma^{12})^{\otimes 2}$$

$$= \int_{(X \times Y)} \omega_1 \|x - y\|^q d\gamma^{12}$$

$$= \int_{(\hat{X} \times \hat{Y})} \omega_1 D^q(x, y) d\hat{\gamma}^{1,2}$$

$$= \int_{(\hat{X} \times \hat{Y})^2} \omega_1 \frac{D^p(x, y)}{|\hat{\gamma}^{12}|} d(\hat{\gamma}^{12})^{\otimes 2}$$

$$= \int_{(\hat{X} \times \hat{Y})^2} \omega_1 \frac{1}{c} D^p(x, y), d(\hat{\gamma}^{12})^{\otimes 2}. \tag{43}$$

In addition, by Bai et al. (2024) Proposition D.3. We have

$$\int_{(X \times Y)^2} \omega_2 |d_X^r(x, x') - d_Y^r(y, y')|^q d(\gamma^{12})^{\otimes 2} + \lambda(|\mu|^2 + |\nu|^2 - 2|\gamma^{12}|^2)$$

$$= \int_{(\hat{X} \times \hat{Y})^2} \omega_2 D_\lambda^q(d_{\hat{X}}^r(x, x'), d_{\hat{Y}}^r(y, y')) d(\hat{\gamma}^{12})^{\otimes 2}. \tag{44}$$

Combining the above two equalities, we prove (41).

Now, we prove the second equality. Note, if we merge the three auxiliary points, i.e., $\hat{\infty}_1 = \hat{\infty}_2 = \hat{\infty}_3$. The optimal value for the above problem (40) is unchanged.

As we merged the points $\hat{\infty}_1, \hat{\infty}_2, \hat{\infty}_3$, by Bai et al. Bai et al. (2023b), the mapping $\gamma^{12} \mapsto \hat{\gamma}^{12}$ defined in (37) is a bijection. Thus, by (41), we have $\gamma^{12} \in \Gamma_{\leq}(\mu, \nu)$ is optimal in (28) iff $\hat{\gamma}^{12}$ is optimal in (40) and we complete the proof. $\square$

**Lemma D.6.** *Choose $\gamma^{01} \in \Gamma_{\leq}(\sigma, \mu), \gamma^{02} \in \Gamma_{\leq}(\sigma, \nu), \gamma^{12} \in \Gamma_{\leq}(\mu, \nu)$ and construct $\hat{\gamma}^{01}, \hat{\gamma}^{02}, \hat{\gamma}^{12}$. Then there exists $\hat{\gamma} \in \mathcal{M}_+(\hat{S} \times \hat{X} \times \hat{Y})$ such that:*

$$(\pi_{0,1})_\# \hat{\gamma} = \hat{\gamma}^{01}, \tag{45}$$

$$(\pi_{0,2})_\# \hat{\gamma} = \hat{\gamma}^{02}, \tag{46}$$

$$\gamma(A_i) = 0, \forall i = 0, 1, 2 \text{ where } A_i = \{\hat{\infty}_i\} \times X \times Y. \tag{47}$$

*Proof.* By *gluing lemma* (see Lemma 5.5 Santambrogio (2015)), there exists $\hat{\gamma} \in \mathcal{M}_+(\hat{S} \times \hat{X} \times \hat{Y})$, such that (45),(46) are satisfied. For the third property, we have:

$$\hat{\gamma}(A_0) \leq \hat{\gamma}(\{\infty_0\} \times \hat{X} \times \hat{Y}) = \hat{\sigma}(\{\infty_0\}) = 0 \qquad \text{by definition (33) of } \hat{\sigma},$$

$$\hat{\gamma}(A_1) \leq \hat{\gamma}(\{\infty_1\} \times \hat{X} \times Y) = \hat{\gamma}^{02}(\{\hat{\infty}_1 \times Y\}) = 0 \qquad \text{by (39)},$$

$$\hat{\gamma}(A_2) \leq \hat{\gamma}(\{\infty_2\} \times X \times \hat{Y}) = \hat{\gamma}^{01}(\{\hat{\infty}_2 \times X\}) = 0 \qquad \text{by (38)}.$$

And we complete the proof. $\qquad\square$

Now, we demonstrate the proof of triangle inequality for fused-PGW distance (28).

*Proof of Theorem D.3 (3).* Note, by the Proposition D.5, the triangle inequality for fused-PGW distance (28) is equivalent to show

$$\hat{d}_{FGW,\lambda}(\hat{\mathbb{X}}, \hat{\mathbb{Y}}) \leq 2^{q-1}(\hat{d}_{FGW,\lambda}(\hat{\mathbb{S}}, \hat{\mathbb{X}}) + \hat{d}_{FGW,\lambda}(\hat{\mathbb{S}}, \hat{\mathbb{Y}})). \tag{48}$$

Choose optimal transportation plans $\gamma^{01}, \gamma^{02}, \gamma^{12}$ for fused PGW problems $d_{FPGW,r,\lambda}(\mathbb{S}, \mathbb{X}), d_{FPGW,r,\lambda}(\mathbb{S}, \mathbb{Y})$ and $d_{FPGW,r,\lambda}(\mathbb{X}, \mathbb{Y})$ respectively. We construct the corresponding $\hat{\gamma}^{01}, \hat{\gamma}^{02}, \hat{\gamma}^{12}$. By the Proposition D.5, $\hat{\gamma}^{01}, \hat{\gamma}^{02}, \hat{\gamma}^{12}$ are optimal.

Choose $\hat{\gamma}$ in lemma D.6. We have:

$$\hat{d}_{FGW,\lambda}(\mathbb{X}, \mathbb{Y})$$

$$= \int_{(\hat{X} \times \hat{Y})^2} \omega_1 \frac{D^q(x,y)}{c} + \omega_2 L_\lambda^q(d_{\hat{X}}(x,x'), d_{\hat{Y}}(y,y'))d\hat{\gamma}^{12}(x,y)d\hat{\gamma}^{12}(x',y')$$

$$\leq \int_{(\hat{S} \times \hat{X} \times \hat{Y})^2} \omega_1 \frac{D^q(x,y)}{c} + \omega_2 L_\lambda^q(x,y)d\hat{\gamma}(s,x,y)d\hat{\gamma}(s',x',y')$$

$$= \underbrace{\int_{(\hat{S} \times \hat{X} \times \hat{Y})} \omega_1 D^q(x,y)d\hat{\gamma}(s,x,y)}_{A} + \underbrace{\int_{(\hat{S} \times \hat{X} \times \hat{Y})^2} \omega_2 L_\lambda^q(x,y)d\hat{\gamma}(s,x,y)d\hat{\gamma}(s',x',y')}_{B}.$$

To bound $A$, we consider the case $D(x,y) > D(s,x) + D(s,y)$ for some $(s,x,y) \in \hat{S} \times \hat{X} \times \hat{Y}$. By definition of $D$, we have $s \in \{\hat{\infty}_i : i \in [0:2]\}, x \in X, y \in Y$. That is, $(s,x,y) \in \bigcup_{i=0}^{2} A_i$. By lemma D.6, we have $\hat{\gamma}(\bigcup_{i=0}^{2} A_i) = 0$. That is, this case has a measure of 0.

Thus, we have

$$A \leq \int_{\hat{S} \times \hat{X} \times \hat{Y}} \omega_1(D(s,x) + D(s,y))^q d\hat{\gamma}(s,x,y)$$

$$\leq \int_{\hat{S} \times \hat{X} \times \hat{Y}} \omega_1(2^{q-1}D(s,x) + 2^{q-2}D(s,y))d\hat{\gamma}(s,x,y)$$

$$= \int_{\hat{S} \times \hat{X} \times \hat{Y}} \omega_1 2^{q-1}D(s,x)d\hat{\gamma}(s,x,y) + \int_{\hat{S} \times \hat{X} \times \hat{Y}} \omega_1 2^{q-2}D(s,y)d\hat{\gamma}(s,x,y)$$

$$= 2^{q-1}\int_{\hat{S} \times \hat{X}} \omega_1 \frac{D(s,x)}{c} d\hat{\gamma}^{01}(s,x) + 2^{q-1}\int_{\hat{S} \times \hat{Y}} \omega_1 \frac{D(s,y)}{c} d\hat{\gamma}^{02}(s,y). \tag{49}$$

where the second inequality follows from the fact

$$(a+b)^q \leq 2^{q-1}a^q + 2^{q-2}b^q, \forall a,b \geq 0. \tag{50}$$

Now we bounde the term $B$. Proposition D.4 in Bai et al. (2024), we have

$$
\begin{aligned}
B &\leq \int_{(\hat{S} \times \hat{X} \times \hat{Y})^2} \omega_2 (L_\lambda(s,x) + L_\lambda(s,y))^q d\hat{\gamma}(s,x,y) d\hat{\gamma}(s',x',y') \\
&\leq \int_{(\hat{S} \times \hat{X} \times \hat{Y})^2} \omega_2 \left( 2^{q-1} L_\lambda^q(s,x) + 2^{q-1} L_\lambda^q(s,y) \right) d\hat{\gamma}(s,x,y) d\hat{\gamma}(s',x',y') \\
&= 2^{q-1} \int_{(\hat{S} \times \hat{X})^2} \omega_2 L_\lambda^q(s,x) d\hat{\gamma}^{01}(s,x) d\hat{\gamma}^{01}(s',x') + 2^{q-1} \int_{(\hat{S} \times \hat{Y})^2} \omega_2 L_\lambda^q(s,y) d\hat{\gamma}^{02}(s,y) d\hat{\gamma}^{02}(s',y').
\end{aligned}
\tag{51}
$$

where the second inequality holds from (50). Combining (49) and (51), we prove the inequality (48) and we complete the proof. $\qquad\square$

### D.4 FUTURE'S DIRECTION.

Note, the general version for (the p-th power of) fused-Gromov Wasserstein distance is defined by

$$
d_{FGW,p}^p(\mathbb{X}, \mathbb{Y}) := \inf_{\gamma \in \Gamma(\mu,\nu)} \int \left( \omega_1 \|x - y\|^q + |d_X(x,x') - d_Y(y,y')|^q \right)^p d\gamma^{\otimes 2}.
\tag{52}
$$

Inspired by the above formulation, we propose the following generalized fused-partial Gromov Wasserstein distance:

$$
d_{FGW,p}^p(\mathbb{X}, \mathbb{Y}) := \inf_{\gamma \in \Gamma(\mu,\nu)} \int \left( \omega_1 \frac{\|x - y\|^q}{|\gamma|} + |d_X(x,x') - d_Y(y,y')|^q + \lambda \left( \frac{|\mu|^2}{|\gamma|^2} + \frac{|\nu|^2}{|\gamma|^2} - 2 \right) \right)^p d\gamma^{\otimes 2}.
\tag{53}
$$

The fused-PGW distance defined in (28) can be treated as a special case of this general formulation by setting $p = 1$. In our conjecture, a similar (semi-)metric property proposed in Theorem D.3 holds for the above general form. We leave the theoretical study of the metric property for our future work.

## E  FRANK WOLF ALGORITHM FOR THE FUSED MASS-CONSTRAINT PARTIAL GROMOV WASSERSTEIN PROBLEM

In discrete setting, the FMPGW problem (10) becomes the following:

$$
FMPGW_\rho(\mathbb{X}, \mathbb{Y}) = \min_{\gamma \in \gamma_\leq^\rho(p,q)} \underbrace{\omega_1 \langle C, \gamma \rangle + \omega_2 \langle M \circ \gamma, \gamma \rangle}_{\mathcal{L}_{C,M}}
\tag{54}
$$

where $\Gamma_\leq^\rho(\mathrm{p},\mathrm{q}) := \{ \gamma \in \mathbb{R}_+^{n \times m} : \gamma 1_m \leq \mathrm{p}, \gamma^\top 1_n \leq \mathrm{q}, |\gamma| \geq \rho \}$, $C = [C(x_i, y_j)]_{i \in [1:n], j \in [1:m]}$, $M_{i,j,i',j'} := L(C^X, C^Y) := [L(C_{i,i'}^X, C_{j,j'}^Y)]_{i,i' \in [1:n], j,j' \in [1:n]}$, $M \circ \gamma = [\langle M[i,j,\cdot,\cdot], \gamma \rangle]_{i,j}$.

Similar to the Fused Gromov-Wasserstein problem, we propose the following Frank-Wolfe algorithm as a solver: The above problem will be solved iteratively. In every iteration, say $k$, we will adapt the following steps:

**Step 1. Gradient computation.**

Suppose $\gamma^{(k-1)}$ is the transportation plan in the previous iteration; it is straightforward to verify:

$$
\nabla \mathcal{L}_{C,M}(\gamma) = \omega_1 C + \omega_2 ((M + M^\top) \circ \gamma),
$$

where $M^\top = [M_{i',j',i,j}]_{i,i' \in [1:n], j,j' \in [1:m]} \in \mathbb{R}^{n \times m \times n \times m}$. Next, we aim to find $\gamma \in \Gamma_\leq^\rho(\mu, \nu)$ for the following problem:

$$
\gamma^{(k)\prime} := \arg \min_{\gamma \in \Gamma_\leq^\rho(\mu,\nu)} \langle \nabla \mathcal{L}_{C,M}(\gamma^{(k-1)}), \gamma \rangle.
\tag{55}
$$

which is a mass-constraint partial OT problem.

---

**Algorithm 3:** Frank-Wolfe Algorithm for FMPGW

**Input:** $C \in \mathbb{R}^{n \times m}, C^X \in \mathbb{R}^{n \times n}, C^Y \in \mathbb{R}^{m \times m}, \mathrm{p} \in \mathbb{R}^n_+, \mathrm{q} \in \mathbb{R}^m_+,$
$\omega_1 \in [0, 1], \rho \in [0, \min(|\mathrm{p}|, |\mathrm{q}|)].$
**Output:** $\gamma^{(final)}$
**for** $k = 1, 2, \ldots$ **do**
   $G^{(k)} \leftarrow \omega_1 C + \omega_2 (M + M^\top) \circ \gamma^{(k)}$ // Compute gradient
   $\gamma^{(k)\prime} \leftarrow \arg \min_{\gamma \in \Gamma^\rho_{\leq}(\mathrm{p},\mathrm{q})} \langle G^{(k)}, \gamma \rangle_F$ // Solve the POT problem.
   Compute $\alpha^{(k)} \in [0, 1]$ via (14) // Line Search
   $\gamma^{(k+1)} \leftarrow (1 - \alpha^{(k)})\gamma^{(k)} + \alpha^{(k)}\gamma^{(k)\prime}$ // Update $\gamma$
   if convergence, break
**end for**
$\gamma^{(final)} \leftarrow \gamma^{(k)}$

---

**Step 2. linear search algorithm.** In this step, we aim to find the optimal step size $\alpha^{(k)} \in [0, 1]$. In particular,

$$\alpha^{(k)} := \arg \min_{\alpha \in [0,1]} \mathcal{L}_{C,M}((1 - \alpha)\gamma^{(k-1)} + \alpha\gamma^{(k)\prime}).$$

Let $\delta\gamma = \gamma^{(k)\prime} - \gamma^{(k-1)}$, the above problem is essentially quadratic problem:

$$\mathcal{L}((1 - \alpha)\gamma^{(k-1)} + \alpha\gamma^{(k)\prime}) = \alpha^2 \underbrace{\langle \omega_2 M \circ \delta\gamma, \delta\gamma \rangle}_{a}$$

$$+ \alpha \underbrace{\langle \omega_2 (M + M^\top) \circ \gamma^{(k-1)} + \omega_1 C, \delta\gamma \rangle}_{b}$$

$$+ \underbrace{\langle \omega_2 M \circ \gamma^{(k-1)} + \omega_1 C, \gamma \rangle}_{c} \tag{56}$$

and $\alpha^*$ is given by

$$\alpha^* = \begin{cases} 1 & \text{if } a \leq 0, a + b \leq 0, \\ 0 & \text{if } a \leq 0, a + b > 0 \\ \mathrm{clip}(\frac{-b}{2a}, [0, 1]), & \text{if } a > 0 \end{cases} \tag{57}$$

Then $\gamma^{(k)} = (1 - \alpha^*)\gamma^{(k-1)} + \alpha^*\gamma^{(k)\prime}$.

In the next section, we provide a detailed introduction to the derivation of the FW algorithms.

## F    GRADIENT COMPUTATION OF FW ALGORITHMS

In this section, we provide a detailed discussion of the gradient computation in fused-MPGW and fused-PGW.

### F.1    BASICS IN TENSOR PRODUCT

In this section, we introduce some fundamental results for tensor computation. Suppose $M \in \mathbb{R}^{n \times m \times n \times m}$. We define the **Transportation of tensor**, denoted as $M^\top$, as:

$$M^\top_{i,j,i',j'} = M_{i',j',i,j}, \tag{58}$$

and we say $M$ is symmetric if $M = M^\top$.

It is straightforward to verify the following:

**Proposition F.1.** *Given tensor $M \in \mathbb{R}^{n \times m \times n \times m}$, then we have:*

   $- (M^\top)^\top = M.$

$$- M^\top \circ \gamma = [\sum_{i',j'} M_{i',j',i,j}\gamma_{i,j}]_{i,j \in [1:n] \times [1:m]}.$$

*Proof.* The first item follows from the definition of $M^\top$. For the second statement, pick $(i,j) \in [1 : n] \times [1 : m]$, we have

$$\sum_{i',j'} M_{i',j',i,j}\gamma_{i',j'}$$
$$= \sum_{i',j'} M^\top_{i,j,i',j'}\gamma_{i',j'}$$
$$= \langle M^\top \circ \gamma \rangle.$$

$\square$

Therefore, we have:

**Proposition F.2.** *The gradient for $\mathcal{L}_{C,M}(\gamma)$ in Fused-PGW (10) is given by:*

$$\nabla \mathcal{L}_{C,M}(\gamma) = \omega_1 C + \omega_2(M \circ \gamma + M^\top \circ \gamma). \tag{59}$$

.

*Similarly, the gradient for $\mathcal{L}_{C,M-2\lambda}$ in fused-PGW (11) is given by:*

$$\nabla \mathcal{L}_{C,M-2\lambda}(\gamma) = \omega_1 C + \omega_2((M - 2\lambda) \circ \gamma + (M - 2\lambda)^\top \circ \gamma). \tag{60}$$

.

*Proof.* Pick $(i,j) \in [1 : n] \times [1 : m]$, we have:

$$\frac{\partial}{\partial \gamma_{ij}} L_{C,M}(\gamma)$$
$$= \frac{\partial}{\partial \gamma_{ij}} \sum_{i,j,i',j'} \omega_1 C_{i,j}\gamma_{i,j} + \omega_2 \sum_{i,j,i',j'} M_{i,j,i',j'}\gamma_{i,j}\gamma_{i',j'}$$
$$= \omega_1 C_{i,j} + \omega_2(\sum_{i',j'} M_{i,j,i',j'}\gamma_{i,j}\gamma_{i',j'} + \sum_{i',j'} M_{i',j',i,j}\gamma_{i,j}\gamma_{i',j'})$$

Therefore, $\nabla L_{C,M}(\gamma) = \omega_1 C + \omega_2(M \circ \gamma + M^\top \circ \gamma)$ and we complete the proof. The gradient for Fused-PGW (11) can be derived similarly. $\square$

At the end of this subsection, we discuss the computation of $M \circ \gamma$ and $M^\top \gamma$.

In general, the computation cost for $M \circ \gamma$ is $n^2 m^2$. However, if the cost function $L$ satisfies:

$$L(r_1, r_2) = f_1(r_1) + f_2(r_2) - h_1(r_1)h_2(r_2), \tag{61}$$

$$M \circ \gamma = f_1(C^X)\gamma 1_m^\top + 1_n \gamma_2^\top f_2(C^Y) - h_1(C^X)\gamma h_2(C^Y), \tag{62}$$

and the corresponding complexity is $\mathcal{O}(n^2 + m^2)$ (see e.g. Peyré et al. (2016); Chapel et al. (2020); Bai et al. (2024) for details.)

Therefore, we have the following:

**Lemma F.3.** *Suppose $M = [L(d_X(x_i, x_{i'}), d_Y(y_j, y_{j'}))]$, then we have:*

$$(M^\top)_{i,j,i',j'} = f_1((C^X)^\top_{i,i'}) + f_2((C^Y)^\top_{j,j'}) - h_1((C^X)_{i,i'})h_2((C^Y)_{j,j'}).$$

*It directly implies:*

$$M^\top \circ \gamma = f_1((C^X)^\top)\gamma 1_m^\top + 1_n \gamma_2^\top f_2((C^Y)^\top) - h_1((C^X)^\top)\gamma h_2((C^Y)^\top).$$

*Proof.* We have:

$$M_{i,j,i',j'}^\top = M_{i',j',i,j}$$
$$= f_1(C_{i',i}^X) + f_2(C_{j',j}^Y) - h_1(C_{i',i}^X)h_2(C_{j',j}^Y))$$
$$= f_1((C^X)_{i',i}^\top) + f_2((C^Y)_{j',j}^\top) - h_1((C^X)_{i',i}^\top)h_2((C^Y)_{j',j}^\top).)$$

And we complete the proof. $\square$

### F.2 GRADIENT IN FUSED-MPGW.

As we discussed in previous section, after iteration $k-1$, the gradient is given by $\nabla\mathcal{L}(\gamma) = \omega_1 C + \omega_2(M \circ \gamma + M^\top \circ \gamma)$, where $M_{i,j,i',j'} = (L(C_{i,i'}^X, C_{j,j'}^Y)) \in \mathbb{R}_+^{n\times m\times n\times m}$. Furthermore, the computational complexity for $M \circ \gamma := [\langle M[i,j,:,:], \gamma\rangle]$ can be improved to be $\mathcal{O}(n^2 + m^2)$.

Based on the gradient, we aim to solve the following to update the transportation plan in iteration $k$:

$$\gamma^{(k)\prime} = \arg\min_{\gamma\in\Gamma_{\le}^\rho(\mu,\nu)} \langle \underbrace{\omega_1 C + \omega_2(M\circ\gamma^{(k-1)} + M^\top\gamma^{(k-1)})}_{\mathcal{C}}, \gamma\rangle. \tag{63}$$

The above problem is essentially the partial OT problem, where the cost function is defined by $\mathcal{C}$.

### F.3 GRADIENT OF FUSED-PGW.

Similarly, after iteration $k-1$, the gradient of cost $\mathcal{L}_{C,M-2\lambda}$ in FPGW with respect to $\gamma$ is given by

$$\nabla\mathcal{L}(\gamma) = \omega_1 C + \omega_2((M-2\lambda)\circ\gamma^{(k-1)} + (M^\top - 2\lambda)\circ\gamma^{(k-1)}).$$

We aim to solve the following problem:

$$\min_{\gamma\in\Gamma_\le(\mu,\nu)} \langle\omega_1 C + \omega_2((M-2\lambda)\circ\gamma^{(k-1)} + (M^\top - 2\lambda)\circ\gamma^{(k-1)}), \gamma\rangle$$
$$= \min_{\gamma\in\Gamma_\le(\mu,\nu)} \langle\underbrace{\omega_1 C + \omega_2(M\circ\gamma^{(k-1)} + M^\top\circ\gamma^{(k-1)})}_{\mathcal{C}}, \gamma\rangle + \lambda|\gamma^{(k-1)}|(|\mu|+|\nu|-2|\gamma|) - \underbrace{\lambda|\gamma^{(k-1)}|(|\mu|+|\nu|)}_{\text{constant}}.$$

Note, if we ignore the constant term, the above problem is the partial OT problem and can be solved by Bonneel et al. (2011) or Bai et al. (2023b).

## G LINE SEARCH ALGORITHM

The line search for Fused-MPGW is defined as

$$\min_{\alpha\in[0,1]} \mathcal{L}((1-\alpha)\gamma^{(k-1)} + \alpha\gamma^{(k)\prime}), \tag{64}$$

where $\mathcal{L}(\gamma) = \omega_1\langle C, \gamma\rangle + \omega_2\langle M, \gamma^{\otimes 2}\rangle$. Let $\delta\gamma = \gamma^{(k)\prime} - \gamma^{(k-1)}$. The above problem is essentially a quadratic problem with respect to $\alpha$:

$$\mathcal{L}((1-\alpha)\gamma^{(k-1)} + \alpha\gamma^{(k)\prime})$$
$$= \mathcal{L}(\gamma^{(k-1)} + \alpha\delta\gamma)$$
$$= \omega_1\langle C, (\gamma^{(k-1)} + \alpha\delta\gamma)\rangle + \omega_2\langle M\circ(\gamma^{(k-1)} + \alpha\delta\gamma), (\gamma^{(k-1)} + \alpha\delta\gamma)\rangle$$
$$= \alpha^2\underbrace{\omega_2\langle M\circ\delta\gamma, \delta\gamma\rangle}_{a} + \alpha\underbrace{\langle\omega_2 M\circ\gamma^{(k-1)} + \omega_1 C, \delta\gamma\rangle + \langle\omega_2 M\circ\delta\gamma, \gamma^{(k-1)}\rangle}_{b} + \underbrace{\langle\omega_2 M\circ\gamma^{(k-1)} + \omega_1 C, \gamma^{(k-1)}\rangle}_{c}, \tag{65}$$

and $\alpha^*$ is given by

$$\alpha^* = \begin{cases} 1 & \text{if } a \le 0, a+b \le 0, \\ 0 & \text{if } a \le 0, a+b > 0 \\ \text{clip}(\frac{-b}{2a}, [0,1]), & \text{if } a > 0 \end{cases}. \tag{66}$$

Next, we simplify the term $b$ in the above formula. We first introduce the following lemma:

**Lemma G.1.** *Choose $\gamma^1, \gamma^2 \in \mathbb{R}^{n \times m}, M \in \mathbb{R}^{n \times m \times n \times m}$, we have:*

$$\langle M \circ \gamma^1, \gamma^2 \rangle = \langle M^\top \circ \gamma^2, \gamma^1 \rangle. \tag{67}$$

*Proof.* We have

$$
\begin{aligned}
\langle M \circ \gamma^1, \gamma^2 \rangle &= \sum_{i,j} \sum_{i',j'} M_{i,j,i',j'} \gamma^1_{i',j'} \gamma^2_{i,j} \\
&= \sum_{i',j'} \sum_{i,j} M^\top_{i',j',i,j} \gamma^2_{i,j} \gamma^1_{i,j} \\
&= \langle M^\top \circ \gamma^2, \gamma^1 \rangle. 
\end{aligned} \tag{68}
$$

$\square$

Therefore, the term $b$ can be further simplified as

$$
\begin{aligned}
b &= \omega_1 \langle C, \delta\gamma \rangle + \omega_2 (\langle M \circ \gamma^{(k-1)}, \delta\gamma \rangle + \langle M^\top \circ \gamma^{(k-1)}, \delta\gamma \rangle) \\
&= \langle \underbrace{\omega_1 C + M \circ \gamma^{(k-1)} + M^\top \circ \gamma^{(k-1)}}_{\nabla L_{C,M}(\gamma)}, \delta\gamma \rangle.
\end{aligned} \tag{69}
$$

That is, we can directly adapt the gradient obtained before to compute $b$ and improve the computation efficiency.

**Remark G.2.** *When we select quadratic cost, i.e., $L(\mathrm{r}_1, \mathrm{r}_2) := |\mathrm{r}_1 - \mathrm{r}_2|^2$, we can set $f_1(\mathrm{r}_1) = \mathrm{r}_1^2, f_2(\mathrm{r}_2) = \mathrm{r}_2^2, h_1(\mathrm{r}_1) = 2\mathrm{r}_1, h_2(\mathrm{r}_2) = \mathrm{r}_2$, then $L(\mathrm{r}_1, \mathrm{r}_2)$ satisfies (61). Furthermore, suppose the problem is in the balanced fused-GW setting, i.e., $\rho = |\mu| = |\nu| = 1$. We have:*

$$
\begin{aligned}
a &= \omega_2 \langle M \circ \delta\gamma, \delta\gamma \rangle \\
&= \omega_2 \langle f_1(C^X) \underbrace{\delta\gamma 1}_{0_n} 1_m^\top + \underbrace{1_n \delta\gamma_2^\top}_{0_m} f_2(C^Y) - h_1(C^X)\gamma h_2(C^Y), \delta\gamma \rangle \\
&= -2\omega_2 \langle C^X \delta\gamma C^Y, \delta\gamma \rangle.
\end{aligned} \tag{70}
$$

*Similarly,*

$$
\begin{aligned}
b &= \langle \omega_2 (M + M^\top) \circ \gamma^{(k-1)} + \omega_1 C, \delta\gamma \rangle \\
&= \omega_1 \langle C, \delta\gamma \rangle + \omega_2 \langle (M + M^\top) \circ \gamma^{(k-1)}, \delta\gamma \rangle \\
&= \omega_1 \langle C, \delta\gamma \rangle + \omega_2 \langle (M + M^\top) \circ \delta\gamma, \gamma^{(k-1)} \rangle \\
&= \omega_1 \langle C, \delta\gamma \rangle - 2\omega_2 \langle C^X \delta\gamma C^Y, \gamma^{(k-1)} \rangle - 2\omega_2 \langle (C^X)^\top \delta\gamma (C^Y)^\top, \gamma^{(k-1)} \rangle + \omega_2 \langle c_{C^X, C^Y}, \gamma^{(k-1)} \rangle \\
&\quad + \omega_2 \langle c_{(C^X)^\top, (C^Y)^\top}, \gamma^{(k-1)} \rangle.
\end{aligned} \tag{71}
$$

*where $c_{C^X, C^Y} = f_1(C^X)\mathrm{p}1_m^\top + 1_m \mathrm{q}^\top f_2(C^Y)$. Thus, the above formulations recover the line search algorithm (see algorithm 2) in Peyré et al. (2016).*

Next, we discuss the line search step in fused-PGW (7).

Replacing $M$ by $M - 2\lambda$ in (65), the solution is obtained by (14).

# H  CONVERGENCE ANALYSIS

As in Chapel et al. (2020), we will use the results from Lacoste-Julien (2016) on the convergence of the Frank-Wolfe algorithm for non-convex objective functions.

## H.1  FUSED-MPGW

Consider the minimization problems

$$\min_{\gamma \in \Gamma^\rho_\leq (\mathrm{p}, \mathrm{q})} \mathcal{L}_{C,M}(\gamma) := \omega_1 \langle C, \gamma \rangle + \omega_2 \langle M \circ \gamma, \gamma \rangle$$

that corresponds to the discrete fused-PGW problem (11).

The objective functions

$$\gamma \mapsto \mathcal{L}_{\hat{M}}(\gamma) = \omega_1 \langle C, \gamma \rangle + \omega_2 \langle M \circ \gamma, \gamma \rangle,$$

is non-convex in general. However, the constraint set $\Gamma^\rho_\le(p, q)$ are convex and compact on $\mathbb{R}^{n \times m}$ (see Proposition C.4.)

Consider the **Frank-Wolfe gap** of $\mathcal{L}_{C,M}$ at the approximation $\gamma^{(k)}$ of the optimal plan $\gamma$:

$$g_k = \max_{\gamma \in \Gamma_\le(p,q)} \langle \nabla \mathcal{L}_{\tilde{M}}(\gamma^{(k)}), \gamma^{(k)} - \gamma \rangle_F. \tag{72}$$

It provided a good criterion to measure the distance to a stationary point at iteration $k$. Indeed, a plan $\gamma^{(k)}$ is a stationary transportation plan for the corresponding constrained optimization problem in (72) if and only if $g_k = 0$. Moreover, $g_k$ is always non-negative ($g_k \ge 0$).

From Theorem 1 in Lacoste-Julien (2016), after $K$ iterations, we have the following upper bound for the minimal Frank-Wolfe gap:

$$g_K := \min_{1 \le k \le K} g_k \le \frac{\max\{2L_1, \mathrm{Lip} \cdot (\mathrm{diam}(\Gamma^\rho_\le(p,q)))^2\}}{\sqrt{K}}, \tag{73}$$

where

$$L_1 := \mathcal{L}_{\tilde{M}}(\gamma^{(1)}) - \min_{\gamma \in \Gamma_\le(p,q)} \mathcal{L}_{\tilde{M}}(\gamma)$$

is the initial global sub-optimal bound for the initialization $\gamma^{(1)}$ of the algorithm; Lip is the Lipschitz constant of function $\gamma \mapsto \nabla \mathcal{L}_{\tilde{M}}$; and

$$\mathrm{diam}(\Gamma_\le(p,q)) = \sup_{\gamma, \gamma' \in \Gamma_\le(\mu,\nu)} \|\gamma - \gamma'\|_F$$

is the $\| \cdot \|_F$ diameter of $\Gamma_\le(p,q)$ in $\mathbb{R}^{n \times m}$.

The important thing to notice is that the constant $\max\{2L_1, D_L\}$ does not depend on the iteration step $k$. Thus, according to Theorem 1 in Lacoste-Julien (2016), the rate in $\tilde{g}_K$ is $\mathcal{O}(1/\sqrt{K})$. That is, the algorithm takes at most $\mathcal{O}(1/\varepsilon^2)$ iterations to find an approximate stationary point with a gap smaller than $\varepsilon$.

Next, we will continue to simplify the upper bound (73). We first introduce the following fundamental results:

**Lemma H.1.** *In the discrete setting, we have*

$$\mathrm{diam}(\Gamma_\le(p,q)) \le 2\rho. \tag{74}$$

*Proof.* Choose $\gamma, \gamma' \in \Gamma^\rho_\le(p,q)$. We apply the property

$$(a - b)^2 \le 2a^2 + 2b^2, \quad \forall a, b \in \mathbb{R}. \tag{75}$$

and obtain

$$\|\gamma - \gamma'\|_F^2 = \sum_{i,j}^{n,m} |\gamma_{i,j} - \gamma'_{i,j}|^2$$

$$\le \sum_{i,j}^{n,m} 2|\gamma_{i,j}|^2 + 2|\gamma'_{i,j}|^2$$

$$\le 2 \left[ \left( \sum_{i,j}^{n,m} \gamma_{i,j} \right)^2 + \left( \sum_{i,j}^{n,m} \gamma'_{i,j} \right)^2 \right]$$

$$= 2(|\gamma|^2 + |\gamma'|^2) \tag{76}$$

$$= 2(\rho^2 + \rho^2) = 4\rho^2,$$

and thus, we complete the proof. $\square$

**Lemma H.2.** *The Lipschitz constant term in (73) can be bounded as follows:*

$$Lip \leq \omega_2 nm \max(|M|)^2. \tag{77}$$

*In particular, when $L(r_1, r_2) = |r_1 - r_2|^p$ where $p \geq 1$, we have $Lip \leq nm(2^{p-1}C^X + 2^{p-1}C^Y)^2$.*

*Proof.* Pick $\gamma, \gamma' \in \Gamma_{\leq}(p, q)$ we have,

$$\|\nabla \mathcal{L}_M(\gamma) - \nabla \mathcal{L}_{C,M}(\gamma')\|_F^2$$

$$= \|(\omega_1 C + \omega_2(M \circ \gamma + M^\top \circ \gamma)) - (\omega_1 C + \omega_2(M \circ \gamma' + M^\top \circ \gamma'))\|_F^2$$

$$= 2\omega_2^2 \|M \circ (\gamma - \gamma')\|_F^2 + 2\omega_2^2 \|M^\top \circ (\gamma - \gamma')\|_F^2$$

$$= 2\omega_2^2 \sum_{i,j} \left([M \circ (\gamma - \gamma')]_{i,j}\right)^2 + 2\omega_2^2 \sum_{i,j} \left([M^\top \circ (\gamma - \gamma')]_{i,j}\right)^2$$

$$= 2\omega_2^2 \sum_{i,j} \left(\sum_{i',j'} M_{i,j,i',j'}(\gamma_{i',j'} - \gamma'_{i',j'})\right)^2 + 2\omega_2^2 \sum_{i,j} \left(\sum_{i',j'} M^\top_{i,j,i',j'}(\gamma_{i',j'} - \gamma'_{i',j'})\right)^2$$

$$\leq 2\omega_2^2 \sum_{i,j} \left(\sum_{i',j'} |M_{i,j,i',j'}||\gamma_{i',j'} - \gamma'_{i',j'}|\right)^2 + 2\omega_2^2 \sum_{i,j} \left(\sum_{i',j'} |M^\top_{i,j,i',j'}||\gamma_{i',j'} - \gamma'_{i',j'}|\right)^2$$

$$\leq 4\omega_2^2 \underbrace{\max(M)^2}_{A} \cdot \underbrace{\sum_{i,j}^{n,m} \left(\sum_{i',j'}^{n,m} |\gamma_{i',j'} - \gamma'_{i',j'}|\right)^2}_{B}$$

For the second term, we have:

$$\sum_{i,j}^{n,m} \left(\sum_{i',j'}^{n,m} |\gamma_{i',j'} - \gamma'_{i',j'}|\right)^2$$

$$\leq \sum_{i,j}^{n,m} \left(nm \sum_{i',j'}^{n,m} |\gamma_{i',j'} - \gamma'_{i',j'}|^2\right)$$

$$\leq n^2 m^2 \|\gamma - \gamma'\|_F^2 \tag{78}$$

For the first term, when $L(r_1, r_2) = |r_1 - r_2|^p$, from the inequality (50), we have:

$$A = \max M \leq \max\{2^{p-1}(C^X)^p + 2^{p-1}(C^Y)^p\}^2.$$

where

$$\max\{2^{p-1}(C^X)^p + 2^{p-1}(C^Y)^p\} := \max\{\max_{i,i',j',j} 2^{p-1}(C^X_{i,i'})^p + 2^{p-1}(C^Y_{j,j'})^p\}.$$

Thus we obtain

$$Lip \leq \frac{\max(2^{p-1}(C^X)^p + 2^{p-1}(C^Y)^p)nm\|\gamma - \gamma'\|_F}{\|\gamma - \gamma'\|_F} = \omega_2 nm \max(2^{p-1}(C^X)^p + 2^{p-1}(C^Y)^p).$$

and we complete the proof. $\square$

Combined the above two lemmas, we derive the convergence rate of the Frank-Wolfe gap (72):

**Proposition H.3.** *When $L(r_1, r_2) = |r_1 - r_2|^2$ in the PGW problem, the Frank-Wolfe gap of algorithm 3, defined in (72) at iteration $k$ satisfies the following:*

$$g_k \leq \frac{\max\left\{2L_1, 4\omega_2 \rho^2 \cdot nm(\max\{2(C^X)^2 + 2(C^Y)^2\})\right\}}{\sqrt{k}}. \tag{79}$$

*Proof.* The proof directly follows from the upper bounds (74),(77) and the inequality (73). □

**Remark H.4.** *Note, if the cost function in PGW is defined by* $|r_1 - r_2|^p$ *for some* $p \neq 2$, *it is straightforward to verify that the upper bound of* $g_k$ *is obtained by replacing the term* $\max((C^X)^2 + (C^Y)^2)$ *should be replaced by*

$$\max_{i,j,i',j'} \left[ 2^{p-1}((C^X)^p + (C^Y)^p) \right].$$

**Remark H.5.** *From the proposition (H.3), to achieve an* $\epsilon-$*accurate solution, the required number of iterations is*

$$\frac{\max \left\{ 2L_1, 4\rho\omega_2 \cdot n^2 m^2 \max(\{2(C^X)^2 + 2(C^Y)^2\}) \right\}^2}{\epsilon^2}.$$

**Remark H.6.** *Note, when* $\omega_2 = 1$, *this convergence rate is constant to the convergence rate of the FW algorithm for MPGW in Chapel et al. (2020). In addition, the convergence rate is independent of C. The main reason is that the part* $\omega_1 \langle C, \gamma \rangle$ *is linear with respect to* $\gamma$. *Thus, this part does not contribute to the Lipschitz of the mapping* $\gamma \mapsto \nabla_M(\mathcal{L})$.

## H.2 CONVERGENCE OF FUSED-PGW

Similar to the above, in this subsection, we discuss the convergence rate for algorithm 2. Suppose $\gamma$ is a solution for the fused-PGW problem (11). In iteration $k$, we define the gap between $\gamma$ and the approximation $\gamma^{(k)}$ and the Frank-Wolfe gap:

$$g_k := \max_{\gamma \in \Gamma_{\leq}(p,q)} \langle \nabla\mathcal{L}_{C,M-2\lambda}(\gamma^{(k)}), \gamma^{(k)} - \gamma \rangle_F. \tag{80}$$

$$g_K := \min_{1 \leq k \leq K} g_k. \tag{81}$$

Thus, the convergence rate for the FW algorithm for the fused-PGW problem (11) can be bounded by the following proposition:

**Proposition H.7.** *When* $L(r_1, r_2) = |r_1 - r_2|^2$ *in the Fused-PGW problem, the Frank-Wolfe gap of algorithm 1, (72) at iteration* $k$ *satisfies the following:*

$$g_K \leq \frac{\max \left\{ 2L_1, 4\omega_2 \min^2(|p|, |q|) \cdot nm(\max\{2(C^X)^2 + 2(C^Y)^2, 2\lambda\}) \right\}}{\sqrt{k}}, \tag{82}$$

*where* $\max(2(C^X)^2 + 2(C^Y)^2, 2\lambda) = \max\{\max_{i,i',j,j'}(2(C_{i,i'}^X)^2 + 2(C_{j,j'}^Y)^2, 2\lambda\}$.

From Theorem 1 in Lacoste-Julien (2016), we have:

$$g_K \leq \frac{\max\{2L_1, \text{Lip} \cdot (\text{diam}(\Gamma_{\leq}(p,q)))\}}{\sqrt{K}}, \tag{83}$$

where Lip is the Lipschitz constant of function $\gamma \to \nabla_M \mathcal{L}$ (with respect to Fubini norm).

By Bai et al. (2024),

$$\text{diam}(\Gamma_{\leq}(p,q)) \leq 2 \min(|p|, |q|). \tag{84}$$

Next, we bound the Lipschitz constant.

**Lemma H.8.** *The Lipschitz constant in (83) can be bounded as follows:*

$$\text{Lip} \leq \omega_2 nm \max |M - 2\lambda|^2. \tag{85}$$

*When* $L(r_1, r_2) = |r_1 - r_2|^p$, *we have:*

$$\text{Lip} \leq \omega_2 nm(\max(2^{p-1}(C^X)^2 + 2^{p-1}(C^Y)^p, 2\lambda))^2. \tag{86}$$

*Proof.* We have:

$$\|\nabla\mathcal{L}_{C,M-2\lambda}(\gamma) - \nabla\mathcal{L}_{C,M-2\lambda}(\gamma')\|_F^2$$

$$= \|(\omega_1 C + \omega_2((M-2\lambda)\circ\gamma + (M^\top-2\lambda)\circ\gamma)) - (\omega_1 C + \omega_2((M-2\lambda)\circ\gamma' + (M^\top-2\lambda)\circ\gamma'))\|_F^2$$

$$= 2\omega_2^2\|(M-2\lambda)\circ(\gamma-\gamma')\|_F^2 + 2\omega_2^2\|(M^\top-2\lambda)\circ(\gamma-\gamma')\|_F^2$$

$$= 2\omega_2^2\sum_{i,j}\left([(M-2\lambda)\circ(\gamma-\gamma')]_{i,j}\right)^2 + 2\omega_2^2\sum_{i,j}\left([(M^\top-2\lambda)\circ(\gamma-\gamma')]_{i,j}\right)^2$$

$$= 2\omega_2^2\sum_{i,j}\left(\sum_{i',j'}(M_{i,j,i',j'}-2\lambda)(\gamma_{i',j'}-\gamma'_{i',j'})\right)^2 + 2\omega_2^2\sum_{i,j}\left(\sum_{i',j'}(M_{i,j,i',j'}^\top-2\lambda)(\gamma_{i',j'}-\gamma'_{i',j'})\right)^2$$

$$\leq 2\omega_2^2\sum_{i,j}\left(\sum_{i',j'}|M_{i,j,i',j'}-2\lambda||\gamma_{i',j'}-\gamma'_{i',j'}|\right)^2 + 2\omega_2^2\sum_{i,j}\left(\sum_{i',j'}(|M_{i,j,i',j'}^\top-2\lambda||\gamma_{i',j'}-\gamma'_{i',j'}|\right)^2$$

$$\leq 4\omega_2^2\underbrace{\max(|M-2\lambda|)^2}_{A}\cdot\underbrace{\sum_{i,j}^{n,m}\left(\sum_{i',j'}^{n,m}|\gamma_{i',j'}-\gamma'_{i',j'}|\right)^2}_{B}.$$

Term $B$ can be bounded by (78). Thus, we obtain the upper bound (85). Furthermore, when $L(\mathrm{r}_1,\mathrm{r}_2) = |\mathrm{r}_1-\mathrm{r}_2|^p$, from inequality (50), we have:

$$\max(|M-2\lambda|) \leq \max(\max_{i,i'}2^{p-1}(C^X)_{i,i'}^p + \max_{j,j'}2^{p-1}(C^Y)_{j,j'}^p, 2\lambda) := \max(2^{p-1}(C^X)^{p-1}, 2^{p-1}(C^Y)^p, 2\lambda)$$

and we obtain the bound (86). □

*Proof of Proposition H.7.* Combining Lemma H.2, (84) and (83), we obtain the upper bound (79) and complete the proof. □

# I  SINKHON ALGORITHM FOR FUSED PARTIAL GROMOV WASSERSTEIN PROBLEM

For convenience, in this section we default to setting

$$L(d_X^r, d_Y^R) = \|d_X - d_Y\|^2,$$

while all propositions, algorithms and proofs extend without loss of generality to a generic loss function $L(d_X^r, d_Y^r)$.

## I.1  SINKHORN ALGORITHM FOR FPGW METRIC

Given mm-spaces $\mathbb{X} = (X, d_X, \mu), \mathbb{Y} = (Y, d_Y, \nu)$, the general unbalanced Fused-Gromov Wasserstein setting, the entropic problem is defined as the following:

$$\min_{\gamma\in\mathcal{M}_+(\mathcal{X}\times\mathcal{Y})}\mathcal{L}(\gamma) + \epsilon H(\gamma^{\otimes 2} \| (\mu\otimes\nu)^{\otimes 2}) \tag{87}$$

$$\mathcal{L}(\gamma) := \omega_1\int_{X\times Y}d(x,y)d\gamma + \omega_2\int_{X^2\times Y^2}|d_X-d_Y|^2 d\gamma^{\otimes 2}$$

$$+ \lambda(D_\phi(\gamma_1^{\otimes 2} \| \mu^{\otimes 2}) + D_\phi(\gamma_2^{\otimes 2} \| \nu^{\otimes 2}))$$

where $D_\phi$ is $\phi-$divergence, $H$ is the (negative) relative entropy

$$H(\mu \| \nu) = \begin{cases} \int\ln(\frac{d\mu}{d\nu})d\mu & \text{if } \mu\ll\nu \\ +\infty & \text{elsewhere} \end{cases}.$$

In the Fused Partial GW setting, $D_\phi$ becomes

$$D_{PTV}(\mu \parallel \nu) := \begin{cases} |\mu - \nu|_{TV} = |\nu - \mu| & \text{if } \mu \leq \nu \\ +\infty & \text{elsewhere} \end{cases} . \tag{88}$$

Note, from the definition of (88), we can restrict the searching space for $\gamma$ to $\Gamma_\leq(\mu, \nu)$:

$$EFPGW(\mathbb{X}, \mathbb{Y}) := \min_{\gamma \in \Gamma_\leq(\mu,\nu)} \mathcal{L}(\gamma) + \epsilon H(\gamma^{\otimes 2} \parallel (\mu \otimes \nu)^{\otimes 2}) \tag{89}$$

$$\mathcal{L}(\gamma) = \omega_1 \langle c, \gamma \rangle + \omega_2 \langle |d_X - d_Y|^2 \otimes \gamma^{\otimes 2} \rangle + \lambda(|\mu|^2 + |\nu|^2 - 2|\gamma|^2)$$

Problem (89) can be further relaxed as:

$$\min_{\gamma,\pi \in \Gamma_\leq(\mu,\nu)} \mathcal{F}(\gamma, \pi) + \epsilon H(\pi \otimes \gamma \parallel (\mu \otimes \nu)^{\otimes 2}) \tag{90}$$

$$\mathcal{F}(\gamma, \pi) := \omega_1 \langle d(x, y), \frac{\gamma + \pi}{2} \rangle + \omega_2 \langle |d_X - d_Y|^2, \gamma \otimes \pi \rangle + \lambda(|\mu|^2 + |\nu|^2 - 2|\gamma||\pi|)$$

It is clear $\mathcal{F}(\gamma, \gamma) = \mathcal{F}(\gamma)$. Thus, (90) $\leq$ (89), and we denote (90) as $LB - FPGW_\lambda(\mathbb{X}, \mathbb{Y})$ (lower bound of Fused Partial Gromov Wasserstein). Essentially, the Sinkhorn algorithm aims to solve $LB - FPGW$.

We first introduce the following fundamental proposition. Note, a similar version can be found in (Séjourné et al., 2021, Proposition 4):

**Proposition I.1.** *Given a fixed* $\pi \in \Gamma_\leq(\mu, \nu)$*, considering the problem:*

$$\min_{\gamma \in \Gamma_\leq(\mu,\nu)} \mathcal{F}(\pi, \gamma) + \epsilon H(\pi \otimes \gamma \parallel (\mu \otimes \nu)^{\otimes 2}),$$

*it is equivalent to solve the following entropic optimal partial transport problem:*

$$\min_{\gamma \in \Gamma_\leq(\mu,\nu)} \int_{X \times Y} c_\pi(x, y) d\gamma + \lambda|\pi|(|\mu| + |\nu| - 2|\gamma|) + \epsilon|\pi|H(\gamma \parallel \mu \otimes \nu) \tag{91}$$

*where*

$$c_\pi(x, y) = \frac{1}{2}\omega_1 d(x, y) + \omega_2[|d_X - d_Y|^2 \circ \pi](x, y) + \epsilon H(\pi \parallel \mu \otimes \nu)$$

$$[|d_X - d_Y|^2 \circ \pi](x, y) = \int_{X \times Y} |d_X(x, x') - d_Y(y, y')| d\pi(x', y')$$

*Proof.* Fix $\pi \in \Gamma_\leq(\mu, \nu)$ in (90). Since $\pi, \gamma \leq \mu \otimes \nu$, we have $\pi, \gamma \ll \mu \otimes \nu$, thus we have:

$$H(\pi \otimes \gamma \parallel (\mu \otimes \nu)^{\otimes 2})$$

$$= \int_{(X \times Y)^2} \ln(\frac{d\pi d\gamma}{d\mu \otimes \nu \cdot d\mu \otimes \nu}) d\mu d\gamma + (|\mu||\nu|)^2 - |\pi||\gamma|$$

$$= \int_{X \times Y} \left[ \int_{X \times Y} \ln(\frac{d\pi}{d\mu \otimes \nu}) d\pi \right] d\gamma + \int_{X \times Y} \left[ \int_{X \times Y} \ln(\frac{d\gamma}{d\mu \otimes \nu}) d\gamma \right] d\pi + (|\mu||\nu|)^2 - |\pi||\gamma|$$

$$= \int_{X \times Y} d\pi H(\gamma \parallel \mu \otimes \nu) + \int_{X \times Y} d\gamma H(\pi \parallel \mu \otimes \nu) + (|\mu||\nu|)^2 - |\pi||\gamma|$$

$$= |\pi|H(\gamma \parallel \mu \otimes \nu) + |\gamma|H(\pi \parallel \mu \otimes \nu) + ((|\mu||\nu|)^2 - |\pi||\mu||\nu|). \tag{92}$$

We obtain:

$$\mathcal{F}(\pi, \gamma) + \epsilon H(\gamma \times \pi \parallel (\mu \otimes \nu)^{\otimes 2})$$

$$= \omega_1 \langle d, \frac{\gamma + \pi}{2} \rangle + \omega_2 \langle |d_X - d_Y|^2, \gamma \otimes \pi \rangle + \lambda(D_{PTV}(\gamma_1 \otimes \pi_1 \parallel \mu^{\otimes 2}) + D_{PTV}(\gamma_2 \otimes \mu_2 \parallel \nu^{\otimes 2}))$$

$$+ \epsilon H(\gamma \times \pi \parallel (\mu \otimes \nu)^{\otimes 2})$$

$$= \omega_1 \langle d, \frac{\gamma + \pi}{2} \rangle + \omega_2 \langle |d_X - d_Y|^2, \gamma \otimes \pi \rangle + \lambda(|\mu|^2 + |\nu|^2 - 2|\gamma||\pi|) + \epsilon H(\gamma \times \pi \parallel (\mu \otimes \nu)^{\otimes 2})$$

$$= \underbrace{\frac{1}{2} \omega_1 \langle d, \pi \rangle + \lambda(|\mu|^2 + |\nu|^2) + \epsilon[(|\mu||\nu|)^2 - |\pi||\mu||\nu|]}_{constant}$$

$$+ \langle \frac{1}{2} \omega_1 d + \omega_2 |d_X - d_Y|^2 \circ \pi + \epsilon H(\pi \parallel \mu \otimes \nu), \gamma \rangle - 2\lambda|\pi||\gamma|) + \epsilon|\pi| H(\gamma \parallel \mu \otimes \nu)$$

where we use the fact (92) and

$$|\gamma|\epsilon H(\pi \parallel \mu \otimes \nu) = \langle \epsilon H(\pi \parallel \mu \otimes \nu), \gamma \rangle.$$

If we ignore the constant part, the remaining part is exactly the entropic partial OT (up to a constant $\lambda|\pi|(|\mu| + |\nu|)$), and we complete the proof. $\qquad\square$

Note, the partial OT problem (91) can be solved by the classical Sinkhorn algorithm. See e.g. Chizat et al. (2018a); Bai (2024).

---

**Algorithm 4:** Sinkhorn Partial OT

---

**Input:** $c, \epsilon, \lambda, p, q$
**Output:** $\gamma$
1 Initialize $u = 1_n, v = 1_m, K = e^{-c/\epsilon} pq^\top$
2 **for** $l = 1, 2, \ldots$ **do**
3 $\quad u = \min(\frac{p}{Kv}, e^{\lambda/\epsilon})$
4 $\quad v = \min(\frac{q}{K^\top u}, e^{\lambda/\epsilon})$
5 $\quad$ If $(u, v)$ converge, break
6 $\gamma \leftarrow (u_i K_{ij} v_j)_{ij}$

---

### I.2 SINKHORN FOR THE FUSED MASS-CONSTRAINT GROMOV WASSERSTEIN PROBLEM.

Similar to the preious section, in the entropic regularziation setting, the fused mass-constraint Gromov Wasserstein problem is defined as:

$$EFMPGW(\mathbb{X}, \mathbb{Y}) := \inf_{\gamma \in \Gamma^\rho_{\leq}(\mu,\nu)} \omega_1 \langle c, \gamma \rangle + \omega_2 \langle |d_X - d_Y|, \gamma^{\otimes 2} \rangle + \epsilon H(\gamma^{\otimes 2} \parallel (\mu \otimes \nu)^{\otimes 2}) \quad (93)$$

and it can be relaxed as

$$LB - EFMPGW(\mathbb{X}, \mathbb{Y}) := \inf_{\gamma, \pi \in \Gamma^\rho_{\leq}(\mu,\nu)} \underbrace{\omega_1 \langle c, \frac{1}{2}(\gamma + \pi) \rangle + \omega_2 \langle |d_X - d_Y|, \gamma^{\otimes 2} \rangle}_{\mathcal{F}(\gamma,\pi)}$$

$$+ \epsilon H(\gamma \otimes \pi \parallel (\mu \otimes \nu)^{\otimes 2}) \quad (94)$$

Note, since $\frac{1}{2}(\gamma + \pi) \in \Gamma^\rho_{\leq}(\mu,\nu)$, and $\mathcal{F}(\gamma, \gamma) = \mathcal{F}(\gamma)$, thus, we have

$$LB - EFMPGW(\mathbb{X}, \mathbb{Y}) \leq EFMPGW(\mathbb{X}, \mathbb{Y}).$$

Similar to the previous section, we have the following property:

**Proposition I.2.** *Given a fixed $\pi \in \Gamma^\rho_{\leq}(\mu, \nu)$, considering the problem:*

$$\min_{\gamma \in \mathcal{M}_+(X \times Y)} \mathcal{F}(\pi, \gamma) + \epsilon H(\pi \otimes \gamma \parallel (\mu \otimes \nu)^{\otimes 2}),$$

---

**Algorithm 5:** Sinkhorn Algorithm for FMPGW

**Input:** $C \in \mathbb{R}^{n \times m}, C^X \in \mathbb{R}^{n \times n}, C^Y \in \mathbb{R}^{m \times m}, p \in \mathbb{R}_+^n, q \in \mathbb{R}_+^m,$
$\omega_2 \in [0,1], \rho \in [0, \min(|p|,|q|)].$
**Output:** $\gamma$
**for** $k = 1, 2, \ldots$ **do**
   $\pi \leftarrow \gamma$
   Solve the Sinkhorn partial OT problem (95) via algorithm (6):

$$\gamma \leftarrow \min_{\gamma \in \Gamma_{\leq}^{\rho}(\mu,\nu)} \int_{X \times Y} c_\pi(x,y) d\gamma + \epsilon \rho H(\gamma \parallel \mu \otimes \nu)$$

   Fix $\gamma$ and solve the similar Sinkhorn partial OT problem (95) via algorithm (6):

$$\pi \leftarrow \min_{\gamma \in \Gamma_{\leq}^{\rho}(\mu,\nu)} \int_{X \times Y} c_\gamma(x,y) d\pi + \epsilon \rho H(\pi \parallel \mu \otimes \nu)$$

   Break if $\pi \approx \gamma$
**end for**

---

*it is equivalent to solving the following entropic optimal partial transport problem:*

$$\min_{\gamma \in \Gamma_{\leq}^{\rho}(\mu,\nu)} \int_{X \times Y} c_\pi(x,y) d\gamma + \epsilon H(\gamma \parallel \mu \otimes \nu) \tag{95}$$

*where $c_\pi$ is defined in Proposition I.1.*

*Proof.* Similar to the proof in Proposition I.1, given $\pi \in \Gamma_{\leq}^{\rho}(\mu,\nu)$, we have

$$\begin{aligned}
&\mathcal{F}(\pi,\gamma) + \epsilon H(\gamma \otimes \pi \parallel (\mu \otimes \nu)^{\otimes 2}) \\
&= \omega_1 \langle d, \frac{\gamma + \pi}{2} \rangle + \omega_2 \langle |d_X - d_Y|^2, \gamma \otimes \pi \rangle + \epsilon H(\gamma \times \pi \parallel (\mu \otimes \nu)^{\otimes 2}) \\
&= \underbrace{\frac{1}{2}\omega_1 \langle d, \pi \rangle + \epsilon[(|\mu||\nu|)^2 - |\pi||\mu||\nu|]}_{constant} \\
&\quad + \langle \underbrace{\frac{1}{2}\omega_1 d + \omega_2 |d_X - d_Y|^2 \circ \pi + \epsilon H(\pi \parallel \mu \otimes \nu)}_{c_\pi}, \gamma \rangle + \epsilon \underbrace{|\pi|}_{=\rho} H(\gamma \parallel \mu \otimes \nu)
\end{aligned}$$

and we complete the proof. $\square$

The Entropic partial OT problem (95) can be solved by the corresponded Sinkhorn algorithm Benamou et al. (2015); Bai (2024), thus we can derive the Sinkhorn algorithm problem for the relaxed fused-mass constraint Gromov Wasserstein problem:

where

$$\begin{aligned}
\mathcal{C}_1 &= \{\gamma \in \mathbb{R}_+^{n \times m} : \gamma_2 \leq q\} \\
\mathcal{C}_2 &= \{\gamma \in \mathbb{R}_+^{n \times m} : \gamma_1 \leq p\} \\
\mathcal{C}_3 &= \{\gamma \in \mathbb{R}_+^{n \times m} : |\gamma| = \rho\}
\end{aligned}$$

$$\text{Proj}_{\mathcal{C}_i}^{KL}(\gamma) := \min_{\gamma^i \in \mathcal{C}_i} KL(\gamma^i \parallel \gamma) = \begin{cases} \text{diag}(\min(\frac{p}{\gamma_2}, 1_n))\gamma \\ \gamma \text{diag}(\min(\frac{q}{\gamma^\top 1_n}, 1_m)) \\ \gamma \frac{\eta}{\|\gamma\|} \end{cases}$$

---

**Algorithm 6:** Sinkhorn Mass-constraint Partial OT

---

**Input:** $p, q, \rho, c$
**Output:** $\gamma$

1  Initialization
2  $K = e^{-c/\epsilon} pq^\top$
3  **for** $i=1,2,3$ **do**
4    $\xi^i \leftarrow 1_{n \times m}$
5  $\gamma^{(0)} \leftarrow K \frac{\rho}{\|K\|}$
6  Main loop
7  $k = 0$ **for** $l = 0, 1, 2, \ldots$ **do**
8    **for** $i=1,2,3$ **do**
9      $k \leftarrow k + 1$
10     $\gamma^{(k)} \leftarrow \texttt{Proj}_{\mathcal{C}_i}^{KL}(\gamma^{(k-1)} \odot \xi^i)$
11     $\xi^i \leftarrow \xi^i \odot \frac{\gamma^{(k-1)}}{\gamma^{(k)}}$
12    Break if $\gamma^{(k)}$ converges

---

## J   Fused Partial Gromov Wasserstein Barycenter

In discrete setting, suppose we have $K$ mm-spaces $\mathbb{X}^k = (X^k \subset \mathbb{R}^d, d_{X^k}, \mu^k := \sum_{i=1}^{n^k} p_i^k \delta_{x_i^k})$ for $k = 1, \ldots, K$ and a fixed pmf function $p \in \mathbb{R}_+^n$ where $n \in \mathbb{N}$. Note for each $k \in [1 : K]$, let $C^k = [d_{X^k}^r(x_i^k, x_{i'}^k)]_{i,i' \in [1:n^k]} \in \mathbb{R}^{n^k \times n^k}$ and let $X^k = [x_1^k, \ldots x_{n_k}^k]^\top \in \mathbb{R}^{n_k \times d}$, we have $(X^k, \mathrm{p}^k, C^k)$ can represent the space $\mathbb{X}^k$. Thus, for convenience, we use the convention $\mathbb{X}^k = (X^k, p^k, C^k)$ to denote the corresponsded mm-space.

Then, given pmf function $\mathrm{p} \in \mathbb{R}_+^n$ where $n \in \mathbb{N}$, and $\beta_1, \ldots \beta_K \geq 0$ with $\sum_{k=1}^K \beta_k = 1$, the fused-Gromov Wasserstein barycenter problem Titouan et al. (2019b) is defined as:

$$\min_{C \in \mathbb{R}^{n \times n}, X \in \mathbb{R}^{n \times d}} \beta_i FGW_L(\mathbb{X}, \mathbb{X}^k), \text{ where } \mathbb{X} = (X, \mathrm{p}, C), \tag{96}$$

where we adapt notation $FGW_L(\cdot, \cdot)$ to simplify the notation $FGW(\cdot, \cdot)$ since $r$ has been incorporated into matrix $C^k$.

Inspired by this work, we present the following fused Partial GW barycenter problems.

### J.1   Fused-MPGW barycenter.

Choose values $\rho_1, \rho_2, \ldots, \rho_K$ where $\rho_k \in [0, \min(|p|, |p^k|)]$ for each $k$. In addition, choose $(\omega_1^k, \omega_2^k) \in [0, 1]$ for $k = 1, 2, \ldots K$ such that $\omega_1^k + \omega_2^k = 1, \forall k$. The fused mass-constrained Partial GW barycenter problem is defined as:

$$\min_{C \in \mathbb{R}^{n \times n}, X \in \mathbb{R}^{n \times d}} \sum_{k=1}^K \beta_k FMPGW_{L, \rho_k}(\mathbb{X}, \mathbb{X}^k),$$

$$= \min_{\substack{C \in \mathbb{R}^{n \times n} \\ X \in \mathbb{R}^{n \times d}}} \min_{\substack{\gamma^k \in \Gamma_{\leq}^{\rho_k}(p, p^k) \\ k \in [1:K]}} \sum_{k=1}^K \beta_k \left( \omega_1^k \langle D(X, X^k), \gamma^k \rangle + \omega_2^k \langle L(C, C^k)(\gamma^k)^{\otimes 2} \rangle \right), \tag{97}$$

where $\mathbb{X} = (X, p, C), \mathbb{X}^k = (X^k, p^k, C^k)$, and $D(X, X^K) = [\|x_i - x_j^k\|^2]_{i \in [1:n], j \in [1:n^k]} \in \mathbb{R}^{n \times n_k}$.

Note that the above problem is convex with respect to $(C, X)$ when $\gamma^k$ is fixed for each $k$. However, it is not convex with respect to $\gamma^k$ for each $k$. Similar to classical fused-GW, it can be solved iteratively by updating $(C, X)$ and $(\gamma^k)_{k=1}^K$ alternatively in each iteration.

**Step 1.** Given $(C, X)$, we update $(\gamma^k)_{k=1}^K$. Note, when $(C, X)$ is fixed, each optimal $(\gamma^k)^*$ is given by

$$(\gamma^k)^* = \arg \min_{\gamma^k \in \Gamma_{\le}^{\rho^k}(p, p^k)} \omega_1^k \langle D(X, X^k), \gamma^k \rangle + \omega_2^k \langle L(C, C^k), (\gamma^k)^{\otimes 2} \rangle,$$

which is a solution for the fused partial GW problem $F - MPGW_{\rho^k}(\mathbb{X}, \mathbb{X}^k)$.

**Step 2.** Given $\gamma^k$, update $(C, X)$.

Suppose $\gamma^k$ is given for each $k$, the objective function in (96) becomes:

$$\min_{C \in \mathbb{R}^{n \times n}, X \in \mathbb{R}^{n \times d}} \sum_{k=1}^K \beta_k \left( \omega_1 \langle D(X, X^k), \gamma^k \rangle + \omega_2 \langle L(C, C^k), (\gamma^k)^{\otimes 2} \rangle \right)$$

$$= \omega_1 \underbrace{\min_{X \in \mathbb{R}^{n \times d}} \sum_{k=1}^K \beta_k \langle D(X, X^k), \gamma^k \rangle}_{A} + \omega_2 \underbrace{\min_{C \in \mathbb{R}^{n \times n}} \sum_{k=1}^K \beta_k \langle L(C, C^k), (\gamma^k)^{\otimes 2} \rangle}_{B}.$$

Problem $B$ admits the solution (we refer Bai et al. (2024) Section M for details). In particular, if $L$ satisfies (61), $f_1, h_1$ are differentiable, then

$$C = \left( \frac{f_1'}{h_1'} \right)^{-1} \left( \frac{\sum_k \xi_k \gamma^k h_2(C^k)(\gamma^k)^\top}{\sum_k \xi_k \gamma_1^k (\gamma_1^k)^\top} \right). \tag{98}$$

In particular, when $L(r_1, r_2) = |r_1 - r_2|^2$, the above formula becomes:

$$C = \frac{\sum_k \xi_k \gamma^k C^k (\gamma^k)^\top}{\sum_k \xi_k \gamma_1^k (\gamma_1^k)^\top}.$$

### J.2 SOLVING THE SUBPROBLEM A.

We first introduce the barycentric projection:

For each $(X^k, p^k, \gamma^k)$, the barycentric projection Bai et al. (2023a) is defined by

$$\hat{x}_i^k = \begin{cases} \frac{1}{\gamma_1^k[i]} \sum_{j=1}^{n_k} \gamma_{i,j}^k x_j^k & \text{if } \gamma_1^k[i] = \sum_{j=1}^{n_k} \gamma_{i,j}^k > 0, \\ x_i & \text{elsewhere.} \end{cases} \tag{99}$$

Note, if $|\gamma^k| = |p|, \forall k$, by Cuturi & Doucet (2014) ( Eq 8), the optimal $X$ is given by

$$X = [x_1, \ldots x_n]^\top, x_i = \sum_{k=1}^K \beta_k \hat{x}_i^k, \forall i \in [1 : n].$$

In this subsection, we will extend the above result to the general case.

**Proposition J.1.** *Any matric $X$ satisfies the following is a solution for problem $A$.*

$$X = [x_1, \ldots, x_n]^\top, x_i = \frac{\sum_{k=1}^K \beta_k \hat{x}_i^k}{\sum_{k=1}^K \beta_i \gamma_1^k[i]} \qquad \text{if } \sum_{k=1}^K \beta_i \gamma_1^k[i] > 0. \tag{100}$$

Note, for each $i$, we use convention $\frac{0}{0} = 0$ if $\sum_{k=1}^K \beta_i \gamma_1^k[i] = 0$.

*Proof.* Problem A can be written in terms of barycentric projection (99). In particular, let $\mathcal{D}_k :=$ $\{i : \sum_{i,j} \gamma_{i,j}^k > 0\}$ and $\mathcal{D} = \bigcup_{k=1}^K \mathcal{D}_k$. We have:

$$A = \min_{X \in \mathbb{R}^{n \times d}} \sum_{k=1}^K \beta_k \sum_{i=1}^n \gamma_1^k[i](\|x_i - \hat{x}_i^k\|^2)$$

$$= \min_{X \in \mathbb{R}^{n \times d}} \sum_{i=1}^n \sum_{k=1}^K \beta_k \gamma_1^k[i](\|x_i - \hat{x}_i^k\|^2)$$

$$= \sum_{i=1}^n \min_{x_i \in \mathbb{R}^d} \sum_{k=1}^K \beta_k \gamma_1^k[i](\|x_i - \hat{x}_i^k\|^2).$$

For each $i$, we have two cases:

Case 1: $\sum_{k=1}^K \beta_k \gamma_1^k[i] > 0$. Then the optimal $x_i$ is given by the weighted average vector:

$$\frac{\sum_{k \in D_i} \beta_k \gamma_1^k[i] \hat{x}_i^k}{\sum_{i \in D_k} \beta_k \gamma_1^k[i]}.$$

Case 2: $\sum_{k=1}^K \beta_k \gamma_1^k[i] = 0$. The problem becomes

$$\min_{x_i \in \mathbb{R}^d} 0,$$

and there is no requirement for $x_i$. And we complete the proof.

$\square$

**Remark J.2.** *In practice, the above formulation can be described by matrices. In particular, let* $\hat{X}^k = [\hat{x}_1^k, \dots \hat{x}_n^k]^\top$, *we have*

$$\hat{X}^k = \frac{\gamma^k X^k}{\gamma_1^k}, \gamma_1^k = \gamma^k 1_m.$$

*Then we have (100) becomes*

$$X = \frac{\sum_{k=1}^K \beta_k \gamma_1^k 1_d^\top \odot \hat{X}}{\sum_{k=1}^K \beta_k \gamma_1^k},$$

*where $\odot$ denotes the element-wise multiplication; the notation $\frac{A}{B}$ where $A \in \mathbb{R}^{n \times d}, B \in \mathbb{R}^n$ denotes the following*

$$\frac{A}{B} = [A[:, 1]/B_1, \dots, A[:, n]/B_n]^\top,$$

*and we use $\frac{0}{0} = 0$ if any element in the denominator is 0.*

## J.3 FUSED-PGW BARYCENTER

Similar to the previous subsection, we define and derive the solution for the fused-Partial GW problem.

Given $\lambda_1, \dots \lambda_K \geq 0, p \in \mathbb{R}_+^n$, the fused partial GW barycenter problem is defined as:

$$\min_{C \in \mathbb{R}^{n \times n}, X \in \mathbb{R}^{n \times d}} \sum_{k=1}^K \beta_k FPGW_{L,\lambda_i}(\mathbb{X}, \mathbb{X}^k), \text{ where } \mathbb{X} = (X, p, C)$$

$$= \min_{C \in \mathbb{R}^{n \times n}, X \in \mathbb{R}^{n \times d}} \min_{\gamma^k \in \Gamma_{\leq}(p, p^k): k \in [1:K]} \sum_{k=1}^K \omega_1^k \langle D(X, X^k), \gamma^k \rangle$$

$$+ \omega_2^k \langle L(C, C^k), (\gamma^k)^{\otimes 2} \rangle + \lambda_k(|p^k|^2 + |p|^2 - 2|\gamma^k|^2). \tag{101}$$

Similar to the fused MPGW barycenter problem, the above problem can be solved iteratively. In each iteration, we have two steps:

Step 1. Given $X, D$, update $\gamma^k$. Note finding each $\gamma^k$ is essentially solving the fused-PGW problem:

$$\gamma^k = \arg\min_{\gamma \in \Gamma_{\leq}(p, p^k)} \omega_1^k \langle D(X, X^k), \gamma \rangle + \omega_2^k \langle L(C, C^k), (\gamma^{\otimes 2}) \rangle + \lambda_k (|p^k|^2 + |p|^2 - 2|\gamma^k|^2)$$

And it can be solved by algorithm 1.

Step 2. Given $\gamma^k : k \in [1 : K]$, update $C, X$.

In this case, (101) becomes

$$\omega_2^k \underbrace{\min_{C \times \mathbb{R}^{n \times n}} \langle L(C, C^k) - 2\lambda_k, (\gamma^k)^{\otimes 2} \rangle}_{A} + \omega_1^k \underbrace{\min_{X \in \mathbb{R}^{n \times d}} \langle D(X, X^k), \gamma^k \rangle}_{B}. \tag{102}$$

Optimal $C$ for subproblem A is (98) by [Proposition M.2 Bai et al. (2024)]. Optimal $X$ for subproblem B is given by (100) by the Proposition J.1.

## K  RELATION BETWEEN FGW, FPGW AND FMPGW.

In this section, we briefly discuss the relation between FGW, Fused-PGW problem (11), and the fused-MPGW (10).

First, we introduce the following "equivalent relation" between FPGW and FMPGW. It can be treated as the generalization of Proposition L.1. in Bai et al. (2024) in the fused-PGW formulation.

**Proposition K.1.** *Given mm-spaces $\mathbb{X} = (X, d_X, \mu), \mathbb{Y} = (Y, d_Y, \nu)$, $r \geq 1, \lambda > 0$. Suppose $\gamma^*$ is a minimizer for FPGW problem $FPGW_\lambda(\mathbb{X}, \mathbb{Y})$, then $\gamma^*$ is also a minimizer for Fused-MPGW problem $FPGW_\rho(\mathbb{X}, \mathbb{Y})$ where $\rho = |\gamma^*|$.*

*Proof.* Pick $\gamma \in \Gamma_{\leq}^\rho(\mu, \nu) \subset \Gamma_{\leq}(\mu, \nu)$. Since $\gamma^*$ is optimal for $FPGW_\lambda(\mathbb{X}, \mathbb{Y})$, we have:

$$\omega_1 \langle C, \gamma^* \rangle + \omega_2 \langle L, (\gamma^*)^{\otimes 2} \rangle + \lambda(|\mu|^2 + |\nu|^2 - 2|\gamma^*|^2)$$
$$\leq \omega_1 \langle C, \gamma \rangle + \omega_2 \langle L, \gamma^{\otimes 2} \rangle + \lambda(|\mu|^2 + |\nu|^2 - 2|\gamma|^2).$$

Combine it with the fact $|\gamma| = |\gamma^*| = \rho$, we complete the proof. □

Next, we discuss the relation between FGW and FMPGW problems.

**Proposition K.2.** *Under the setting of Proposition K.1, suppose $|\mu| = |\nu|$, then*

$$FMPGW_\rho(\mathbb{X}, \mathbb{Y}) = FGW_\rho(\mathbb{X}, \mathbb{Y}).$$

*Proof.* In this case, $\Gamma_{\leq}^\rho(\mu, \nu) = \Gamma(\mu, \nu)$ and we complete the proof. □

Similarly, in the extreme case, the FPGW problem can also recover the FGW problem. We first introduce the following lemma:

**Lemma K.3.** *Suppose $\mu, \nu$ are supported in compact set and $|\mu|, |\nu| > 0$. Let $X = supp(\mu), Y = supp(\nu)$, and $\max L(d_X^r, d_Y^r) := \max_{x, x' \in X, y, y' \in Y} L(d_X^r(x, x'), d_Y^r(y, y')), \max C := \max_{x \in X, y \in Y} C(x, y)$.*

*Suppose*

$$2\lambda \geq \omega_1 \frac{1}{\min(|\mu|, |\nu|)} \max(C) + \omega_2 \max L(d_X^r, d_Y^r) \tag{103}$$

*there exists a solution, denoted as $\gamma^*$, for $FPGW_\lambda(\mathbb{X}, \mathbb{Y})$ such that*

$$|\gamma^*| = \min(|\mu|, |\nu|). \tag{104}$$

*If we replace "$\geq$" by "$>$" in the inequality (103), then every solution of $FPGW_\lambda(\mathbb{X}, \mathbb{T})$ satisfies (104).*

*Proof.* For convenience, we suppose $|\mu| \leq |\nu|$. Suppose (103) holds. Choose an optimal $\gamma \in \Gamma_{\leq}(\mu, \nu)$. By lemma E.1. in Bai et al. (2024), there exists $\gamma' \in \Gamma_{\leq}(\mu, \nu)$ such that $\gamma \leq \gamma'$ with $|\gamma'| = |\mu|$, i.e. $\gamma'_1 = \mu$.

We have:

$$
\int_{X \times Y} \omega_1 C(x, y) d\gamma' + \int_{(X \times Y)^2} \omega_2 L(d_X^r, d_Y^r) d\gamma'^{\otimes 2} + \lambda(|\mu|^2 + |\nu|^2 - 2|\gamma'|^2)
$$

$$
- \int_{X \times Y} \omega_1 C(x, y) d\gamma + \int_{(X \times Y)^2} \omega_2 L(d_X^r, d_Y^r) d\gamma^{\otimes 2} + \lambda(|\mu|^2 + |\nu|^2 - 2|\gamma|^2)
$$

$$
= \int_{X \times Y} \omega_1 C(x, y) d(\gamma' - \gamma) + \int_{(X \times Y)^2} \omega_2 L(d_X^r, d_Y^r) - 2\lambda d(\gamma'^{\otimes 2} - \gamma^{\otimes 2})
$$

$$
\leq \omega_1 \max(C)|\gamma' - \gamma| + \omega_2 (\max(L(d_X^r, d_Y^r)) - 2\lambda)|\gamma'^{\otimes 2} - \gamma^{\otimes 2}|
$$

$$
= \underbrace{\left( \omega_1 \frac{\max(C)}{|\gamma + \gamma'|} + \omega_2 \max(L(d_X^r, d_Y^r)) - 2\lambda \right)}_{A} \underbrace{|\gamma'^{\otimes 2} - \gamma^{\otimes 2}|}_{B}
$$

$$
\leq 0, \tag{105}
$$

where (105) follows by the fact $A \leq 0, B \geq 0$. Thus we have $\gamma'$ is optimal.

Now we suppose

$$
2\lambda > \omega_1 \frac{1}{\min(|\mu|, |\nu|)} \max(C) + \omega_2 \max L(d_X^r, d_Y^r).
$$

We assume there exists an optimal $\gamma$ such that $|\gamma| < |\mu|$.

If we can find optimal $\gamma$ with $|\gamma| < |\mu|$, then $A < 0$ and $B > 0$ and we have (105) $< 0$. That is, $\gamma'$ admits a smaller cost. It is a contradiction since $\gamma$ is optimal, and we complete the proof. $\square$

**Remark K.4.** *If $\min(|\mu|, |\nu|) = 0$, $\Gamma_{\leq}(\mu, \nu) = \{\mathbf{0}\}$ where $\{\mathbf{0}\}$ is the zero measure. In this case, (104) is automatically satisfied, and there is no requirement for $\lambda$.*

Based on this lemma, the FPGW can recover FGW in the extreme case:

**Proposition K.5.** *Under the setting of Proposition K.2, suppose $\lambda$ satisfies (103), then*

$$
FPGW_\lambda(\mathbb{X}, \mathbb{Y}) = FGW(\mathbb{X}, \mathbb{Y}).
$$

*Proof.* By Lemma K.3, there exists optimal $\gamma$ for $FPGW_\lambda(\mathbb{X}, \mathbb{Y})$ with $|\gamma| = |\mu| = |\nu|$. By Proposition K.1, we have $\gamma$ is optimal for $FMPGW_\rho(\mathbb{X}, \mathbb{Y})$ where $\rho = |\mu| = |\nu|$. By Proposition K.2, we have $\gamma$ is optimal for $FGW(\mathbb{X}, \mathbb{Y})$. Thus, $\gamma$ and we complete the proof. $\square$

**Remark K.6** (Summary of relations between FPGW, FMPGW, FGW, and OT). *Due to the inherent non-convexity of Gromov–Wasserstein objectives, it is not rigorous to claim that FGW and FPGW are fully equivalent. Nevertheless, the following informal relations help clarify their conceptual connections:*

**Relation between $\lambda$ in FPGW and $\rho$ in FMPGW.** *Both parameters control the amount of transported mass. In the FPGW formulation, increasing $\lambda$ encourages more mass to be transported, while in FMPGW, $\rho$ directly specifies the transported mass fraction. Hence, $\lambda$ in FPGW can be viewed as the Lagrange multiplier associated with the transported-mass constraint in FMPGW.*

**Relation between FPGW and FGW.** *When $|\mu| = |\nu|$ and $\lambda$ is sufficiently large, FPGW reduces to FGW. In this case, the partial-mass penalty vanishes, and the formulation coincides with the standard fused Gromov–Wasserstein distance.*

**Relation between FPGW and OT.** *In the same setting where FPGW recovers FGW, by choosing $\omega_1 = 1$ and $\omega_2 = 0$, FPGW simplifies to the classical Optimal Transport problem.*

*These relationships summarize how the proposed FPGW framework unifies and generalizes several existing OT-based formulations.*

### K.1 Comparison for FPGW, FUGW, Sinkhorn and Frank-Wolfes

**Remark K.7** (FPGW vs. FUGW: Total Variation vs. KL Divergence)**.**

*(1) Theory. We show that FPGW admits a (semi-)metric, while the metric property of FUGW remains unclear. In the graph classification experiments, we also do not observe an advantage of FUGW over FPGW (see Appendix N).*

*(2) Algorithms. FPGW can be optimized by both a Frank–Wolfe (FW) solver and a Sinkhorn-based solver, whereas FUGW can only be solved via the Sinkhorn algorithm.*

*(3) Computational aspects. The KL divergence used in FUGW is nonlinear, while the total variation (TV) penalty used in FPGW is linear. We hypothesize that this difference contributes to faster convergence for FPGW even under entropic regularization. For example, in graph classification, **FUGW (Sinkhorn)** requires 100–400 seconds, whereas **FPGW (Sinkhorn)** completes in 7–40 seconds (see the updated PDF).*

**Remark K.8** (FW vs. Sinkhorn: Transportation Cost vs. Partial Matching Plan)**.**

*(1) Transportation cost. When the downstream task requires the* transportation cost itself *(e.g., as a kernel matrix in graph classification), the Frank-Wolfe (FW) algorithm provides more accurate estimates. The Sinkhorn solver, on the other hand, requires a sufficiently large entropic regularization weight to prevent numerical instability (NaN errors), which in turn biases the objective and blurs the transport cost. Therefore, we use **FPGW (FW)** in the graph classification experiments, where it yields higher accuracy than **FUGW (Sinkhorn)** (see Table 4). For example, on the Protein dataset, FPGW (FW) achieves approximately 69–72% accuracy, while FUGW (Sinkhorn) gives 68–70%. On the Synthetic dataset, FPGW (FW) attains 94–97% versus 45–60% for FUGW (Sinkhorn).*

*(2) Partial matching plan. When the task requires identifying a* partial matching plan *(e.g., in graph matching problems), the objectives of GW, PGW, FGW, and FPGW are non-convex, and the FW algorithm can converge to suboptimal local minima. In this case, introducing an entropic regularization term improves the optimization landscape and enhances numerical stability. Therefore, for partial matching tasks, we employ the **Sinkhorn-based** solver, which provides more stable convergence and reliable partial correspondences.*

### K.2 Wall-clock Comparison Experiment

In this section, we present a wall-clock time comparison among **FPGW**, **Sinkhorn-FPGW**, **FMPGW** (from PythonOT Flamary et al. (2021)), and **Sinkhorn-FMPGW** (from PythonOT).

**Dataset.** Given two empirical measures $X = \{x_1, \ldots, x_n\} \subset \mathbb{R}^2$ and $Y = \{y_1, \ldots, y_m\} \subset \mathbb{R}^2$ with $x_i \sim$ i.i.d. $\mathcal{N}(0, I_2)$ and $y_j \sim$ i.i.d. $\mathcal{N}(0, I_2)$, we construct the distance matrices

$$C_X = [\|x_i - x_{i'}\|^2]_{i,i'=1}^n, \tag{106}$$

$$C_Y = [\|y_j - y_{j'}\|^2]_{j,j'=1}^m. \tag{107}$$

We test problem sizes $n \in \{10, 100, 200, 400, 800, 1000, 2000, 5000, 10000\}$, and for each $n$, we choose $m$ uniformly at random from the interval $[1.1n, 1.5n]$.

**Experiment and parameter settings.** For FPGW in Eq. 11, we test the cases $\lambda \in \{0.1, 1, 10\}$. For FMPGW and Sinkhorn-FMPGW, the mass parameter is set to $|\gamma|$, where $\gamma$ is obtained from FPGW (see Proposition K.1).

Each experiment is repeated $k = 3$ times and we report the average wall-clock time. We set $\omega_2 = 0.5$ and the Sinkhorn regularization parameter $\epsilon = 0.1$.

For the FW-based methods, the linear subproblem is solved by the C++ solver "emd" from PythonOT, with a maximum number of iterations

$$\max\{1000, \; 100(n + m)\}. \tag{108}$$

This prevents non-convergence inside an FW iteration.

The Sinkhorn solvers for FPGW-Sinkhorn and FMPGW-Sinkhorn are implemented in NumPy, with maximum iterations

$$\max\{200, \; 10(n + m)\}. \tag{109}$$

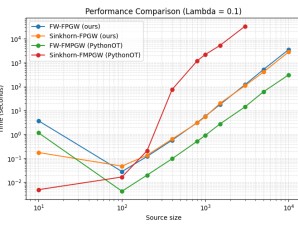 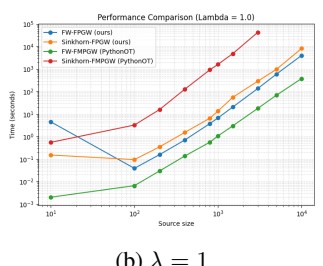 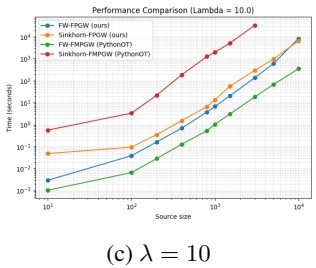

$(a)\ \lambda = 0.1$ $\qquad\qquad$ $(b)\ \lambda = 1$ $\qquad\qquad$ $(c)\ \lambda = 10$

Figure 5: We evaluate problem sizes $n = 10, \ldots, 10^4$ for **FMPGW–FW (PythonOT)**, **FPGW–FW (ours)**, and **FPGW–Sinkhorn (ours)**. The **FMPGW–Sinkhorn (PythonOT)** solver becomes extremely slow when $n \geq 2000$; for example, it requires *approximately two days* to solve a single pair, whereas the other methods finish within *2–3 hours*. Therefore, for FMPGW–Sinkhorn we report results only for $n = 10, \ldots, 2000$.

The convergence tolerance for all methods is set to $10^{-8}$.

**Performance Analysis.** The log-log wall-clock comparison is shown in Figure 5. FMPGW-FW, FPGW-FW, and FPGW-Sinkhorn have similar computational cost, with FPGW-FW being slightly faster than FMPGW-FW. Both FW algorithms are faster than FPGW-Sinkhorn because they use a C++ linear programming solver.

Sinkhorn-FMPGW is extremely slow. When $n \geq 2000$, it requires about two days to solve a single pair, while the other methods take only two to three hours. This is because the partial OT Sinkhorn update in FMPGW-Sinkhorn follows the algorithm of Benamou et al. Benamou et al. (2015), which needs six matrix operations per iteration, while FPGW-Sinkhorn requires only two (see Chizat et al. Chizat et al. (2018a)).

In summary, although FPGW and FMPGW are theoretically related, in practice both the FW and Sinkhorn solvers for FPGW are faster than those for FMPGW, with the difference being especially large for the Sinkhorn variants.

## L  MEASURE GRAPH SIMILARITY VIA FUSED GW/PGW.

In the following, we discuss how to model graphs as mm-spaces. Based on this modeling, fused-GW and fused-PGW can be adapted to measure graph similarity.

Given graphs $G_1 = (V_1, E_1)$ and $G_2 = (V_2, E_2)$, where $V_1, V_2$ are sets of vertices (nodes) and $E_1, E_2$ are sets of edges.

First, consider $G_1$. Suppose $V_1 = \{v_1^1, \ldots, v_{N_1}^1\}$. We construct the mm-space $\mathbb{X}_1 = (V_1, d_{V_1}, \mu^1 = \sum_{i=1}^{N_1} \mathrm{p}_i^1 \delta_{v_i})$, where $\mathrm{p}_i^1 > 0$ for all $i$. Let $\mathrm{p}^1 = [p_1^1, \ldots, p_{N_1}^1]$. Note that when $\sum(\mathrm{p}^1) = 1$, $\mathrm{p}^1$ represents the probability mass function (pmf) of the measure $\mu^1$. For general cases, we refer to $\mathrm{p}^1$ as the mass function (mf) of $\mu^1$.

In this formulation, the function $d_{V_1} : V_1^2 \to \mathbb{R}$ can be defined as follows:

- **Adjacency indicator function:** $d_{V_1}(v_1, v_2) = 1$ if $(v_1, v_2) \in E_1$, and $d_{V_1}(v_1, v_2) = 0$ otherwise.

- **Shortest distance function:** $d_{V_1}(v_1, v_2)$ is the length of the shortest path connecting $v_1$ and $v_2$ if such a path exists, or $\infty$ if $v_1$ and $v_2$ are not connected.

Let $\mathcal{F}$ denote the space of feature assignments for all nodes. We define a feature function $f : V \to \mathcal{F}$ such that $x_i = f(v_i)$ represents the feature of the vertex $v_i$. For convenience, when the graph has discrete features, $\mathcal{F}$ is a discrete set; when the graph has continuous features, $\mathcal{F} = \mathbb{R}^d$, where $d$ is the dimension of each feature. In both cases, we define a metric $d_{\mathcal{F}}$ in the feature space.

The feature similarity function $d_{\mathcal{F}}$ is set as follows:

– **Continuous feature graph:** Since $\mathcal{F} = \mathbb{R}^d$, we define $d_{\mathcal{F}}$ as the (squared) Euclidean distance in $\mathbb{R}^d$.

– **Discrete feature graph:** We first apply the Weisfeiler-Lehman kernel Vishwanathan et al. (2010), which encode elements in feature space as $\mathrm{wl} : \mathcal{F} \to S^H$, where $S$ is finite discrete set, $H \in \mathbb{N}$, typically set to 2 or 4. We then use the Hamming distance in $S^{\mathrm{h}}$ as $d_{\mathcal{F}}$. For details, refer to Section 4.2 of Vayer et al. (2020).

**Remark L.1** (mm-space to gm-space). *Classical Gromov–Wasserstein distances are defined on metric measure spaces (mm-spaces), where the ground cost is a metric satisfying the triangle inequality. In graph settings, adjacency or structural relations are typically not metrics. Therefore, GW-type formulations on graphs are built on* gauge measure spaces (gm-spaces), *where the structure function is a symmetric measurable gauge function Beier et al. (2022). This relaxation is standard in graph optimal transport and allows non-metric structure costs (e.g. adjacent matrix).*

## M  PARAMETER SETTING IN GRAPH MATCHING AND GEOMETRY MATCHING EXPERIMENTS

We present the detailed parameter settings for the graph and geometry matching experiments in this section. First, consider the graph node distribution settings:

$$\mu = \sum_{i=1}^{n} p_i \delta_{v_i^q}, \quad \nu = \sum_{j=1}^{m} q_j \delta_{v_j^o},$$

where $\mu$ denotes the node distribution of the query graph and $\nu$ denotes the node distribution of the original graph.

For unbalanced GW methods, **SpecGW** (Spectral Gromov–Wasserstein Chowdhury & Needham (2021)), **eBPG** (Entropic Bregman Projected Gradient Solomon et al. (2016)), **BPG** (Bregman Projected Gradient Xu et al. (2019)), **BAPG** (Bregman Alternating Projected Gradient Li et al. (2023)), and **srGW** (Semi-relaxed Gromov–Wasserstein Vincent-Cuaz et al. (2022)), we follow their default settings

$$p_i = \tfrac{1}{n}, \quad q_j = \tfrac{1}{m}. \tag{110}$$

For **PGW** (Partial Gromov–Wasserstein Chapel et al. (2020); Bai et al. (2024)), **UGW** (Unbalanced Gromov–Wasserstein Séjourné et al. (2021)), **RGW** (Outlier Robust Gromov–Wasserstein Kong et al. (2024)), **FUGW** (Fused Unbalanced Gromov–Wasserstein Thual et al. (2022)), and **Sink-FUPGW** (Sinkhorn Fused Partial Gromov–Wasserstein, ours), in addition to the above balanced setting, we alternatively consider the unbalanced setting:

$$p_i = q_j = \tfrac{1}{\min(n,m)} = \tfrac{1}{n}. \tag{111}$$

And select the best option for each method.

Other parameter settings are summarized in Tables 2 and 3.

In particular, in Spectral GW, $time$ is the time limit of the heat kernel. In methods: eBPG, BPG, BAPG, srGW, UGW, FUGW, RGW, and sink-FPGW, $\epsilon$ is the weight of the entropic regularization term. In BAPG, $\rho$ is the weight of the Bregman divergence penalty. In PGW, "mass" is the mass constraint. In UGW, FUGW, sink-FPGW $\rho$ is the weight of marginal regularization terms. In FPGW, $\lambda$ is the weight of marginal regularization terms. In the fused methods (FGW, FUGW, sink-FPGW), $\alpha$ denotes the balance between graph structure and node features. In RGW, $\rho$ is the hard marginal constraint, $\eta$ is the weight of the marginal constraint (soft constraint), $t$ are the stepsizes in Bregman proximal alternating linearized minimization (BPALM).

**Other settings in geometry matching.** In geometric matching, we only consider unbalanced GW methods (RGW, FUGW, FPGW), since UGW and PGW can be regarded as special cases of FUGW and FPGW and are therefore not included separately. We adopt the unbalanced mass function setting in (111).

In this experiment, we further assume that a fixed pair of points from the ground-truth correspondence is known, denoted by $(x_0^q, x_0^o)$, where the first point belongs to the query shape and the second to the original shape. For each point $x$ in the query shape, its feature is defined as $d(x, x_0^q)$, where $d$ is the Euclidean distance on the shape. Features of points in the original shape are defined analogously.

Table 2: The parameter settings in all the methods. Datasets with node attributes include Synthetic, Enzymes, Cuneiform, COX2, BZR, Protein and AIDS.

| | parameter | Dataset with node attributes | Douban |
|---|---|---|---|
| SpecGW | $time$ | 10 | 10 |
| eBPG | $\epsilon$ | 0.1 | 0.01 |
| BPG | $\epsilon$ | 0.2 | 0.01 |
| BAPG | $\rho$ | 0.1 | 0.01 |
| | $\epsilon$ | 1e-6 | 1e-6 |
| srGW | $\epsilon$ | 2.0 | 10 |
| PGW | mass | $\min(|p|, |q|)$ | $\min(|p|, |q|)$ |
| UGW | $\epsilon$ | 0.01 | 1e-3 |
| | $\rho$ | 0.05 | 1.0 |
| RGW | $\rho$ | 0.05/0.1 | 0.1 |
| | $\epsilon$ | 0.05 | 1e-3 |
| | $t$ | 0.1 | 0.1 |
| | $\tau$ | 0.1 | 0.1 |
| FGW | $\alpha$ | 0.5 | 0.5 |
| FUGW | $\alpha$ | 0.5 | 0.5 |
| | $\epsilon$ | 0.01 | 2e-4 |
| | $\rho$ | 1.0 | 1.0 |
| sink-FPGW(ours) | $\alpha$ | 0.33 | 0.5 |
| | $\epsilon$ | 0.02 | 1e-4 |
| | $\lambda$ | 1.0 | 1.0 |

Table 3: Parameter setting of all the methods in geometry matching.

| | parameter | Upper Body | Lower Body |
|---|---|---|---|
| RGW | $\rho$ | 0.1 | 0.1 |
| | $\epsilon$ | 1e-3 | 1e-3 |
| | $t$ | 0.1 | 0.1 |
| | $\tau$ | 0.1 | 0.1 |
| FUGW | $\alpha$ | 0.5 | 0.5 |
| | $\epsilon$ | 0.005 | 0.1 |
| | $\rho$ | 1.0 | 1.0 |
| sink-FPGW(ours) | $\alpha$ | 0.5 | 0.5 |
| | $\epsilon$ | 1e-3 | 1e-4 |
| | $\lambda$ | 1.0 | 1.0 |

## N  GRAPH CLASSIFICATION

**Dataset Setup.** We consider four widely used benchmark datasets, divided into two groups. The first group includes *Mutag* Debnath et al. (1991) and *MSRC-9* Rossi & Ahmed (2015), which consist of graphs with discrete attributes. The second group consists of vector-attributed graphs, including *Synthetic* Feragen et al. (2013), *Cuneiform* Kriege et al. (2016), and *Proteins* Borgwardt & Kriegel (2005).

For each dataset, we randomly select 50% of the graphs and add outlier nodes. Specifically, for each selected graph, suppose $N$ is the number of vertices. We manually add $\eta N$ extra nodes, where $\eta \in \{0, 10\%, 20\%, 30\%\}$ represents the "level of outliers/noise". For clarity, nodes that are not outliers are referred to as "regular nodes". The outlier nodes are randomly connected to both the regular nodes and each other. The features of the outlier nodes are defined as follows:

- For graphs with discrete features, each outlier node is assigned a label that does not occur among the regular nodes in the graph.

- For graphs with continuous features, suppose all features lie within a compact set $[x_1, y_1] \times \ldots \times [x_d, y_d] \subset \mathbb{R}^d$, where $d \in \mathbb{N}$ is the dimension of the feature space. We assign features

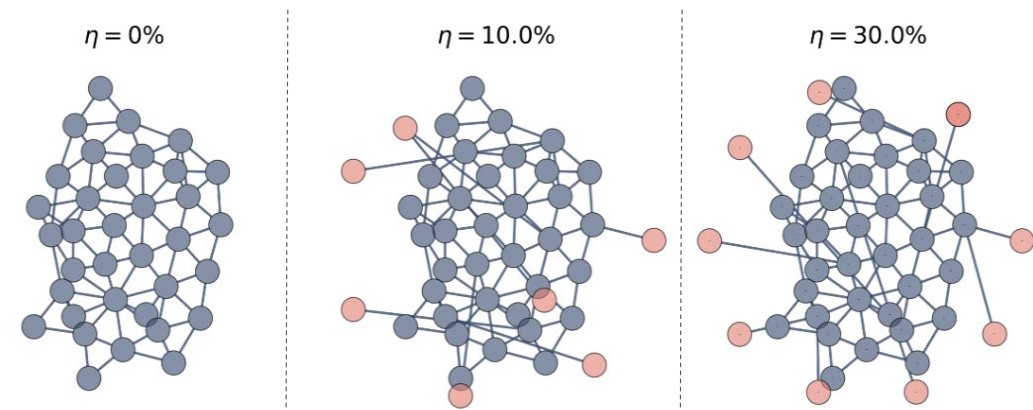

Figure 6: We visualize the graph classification data in this plot. The parameter $\eta \in \{0, 10\%, 20\%, 30\%\}$ represents the proportion of outlier nodes. The blue nodes are the regular nodes in the graph, and the pink nodes are the outliers.

to the outlier nodes by sampling vectors from $[y_1, y_1 + 2\mathrm{sd}_1] \times \ldots \times [y_d, y_d + 2\mathrm{sd}_d]$, where $\mathrm{sd}_i$ is the standard deviation of all node features in dimension $i$.

Furthermore, for each graph $G$, let $N_G$ denote the number of regular nodes (i.e., nodes that are not outliers). We assume $N_G$ is known.

**Fused Gromov/Unbalanced Gromov/Partial Gromov Setting.**

We adapt the process in section L to model each graph $G$ as an mm-space. In particular, in the FGW setting, we define the (probability) mass function p as $\mathrm{p}_i = \frac{1}{N}$, where $N$ is the number of nodes in graph $G$. In the FUGW/FPGW setting, we define the mass function p as $\mathrm{p}_i = \frac{1}{N_G}$, where $N_G$ is the number of regular nodes in $G$.

For the distance functions $d_{V_1}$ and $d_{V_2}$ in graphs $G_1$ and $G_2$, we use the "shortest path" metric in this experiment. We select FMPGW (10) and set the parameter $\rho = 1$.

**Baseline methods.** We consider the fused Gromov Wasserstein Vayer et al. (2020); Titouan et al. (2019b) and the Unbalanced fused Gromov Wasserstein Thual et al. (2022) as baselines for both discrete and continuous feature graph datasets. In addition, for discrete feature graph datasets, we add the Weisfeler Lehman kernel (WLK) Vishwanathan et al. (2010),Graphlet count kernel (GK) Shervashidze et al. (2009), random walk kernel (RWK) Kriege et al. (2018), ODD-STh Kernel (ODD) Da San Martino et al. (2012), vertex histogram (VH) Sugiyama & Borgwardt (2015), Lovasz Theta (LT) Lovász (1979), SVM theta (ST) Jethava et al. (2013) as baselines methods. For the continuous feature graphs dataset, we also consider the HOPPER kernel Feragen et al. (2013) and the propagation kernel Neumann Neumann et al. (2016).

**Classifier setup.** We adapt the SVM classification model in this experiment. In particular, given a graph dataset, we first compute the pairwise distance via FGW/(other baselines)/FPGW. By using the approach given by Beier et al. (2022); Titouan et al. (2019b), we combine each distance with a support vector machine (SVM), applying stratified 10-fold cross-validation. In each iteration of cross-validation, we train an SVM using $\exp(-\sigma D)$ as the kernel, where $D$ is the matrix of pairwise distances (w.r.t. one of the considered distances) restricted to 9 folds, and compute the accuracy of the model on the remaining fold. We report the accuracy averaged over all 10 folds for each model.

**Performance Analysis.** The accuracy comparison is summarized in Table 4. For discrete feature graph datasets, VH, FGW, and FPGW achieve the highest overall accuracy. However, as the noise level $\eta$ increases, FGW's performance noticeably deteriorates, while the other three methods remain robust against outlier corruption.

In contrast, for continuous datasets such as *Proteins* and *Synthetic*, the baseline methods, including "Propagation," "GraphHopper," and "FGW," experience a significant drop in performance. Notably, FPGW maintains strong performance across these datasets.

Table 4: Kernel accuracy (%) with standard deviation under different noise levels (0%, 10%, 20%, 30%). Top: results for discrete node features. Bottom: results for continuous node features.

| | MSRC-9 | | | | MUTAG | | | |
|---|---|---|---|---|---|---|---|---|
| | 0% | 10% | 20% | 30% | 0% | 10% | 20% | 30% |
| WLK | $85.5^{\pm5.9}$ | $86.5^{\pm5.2}$ | $86.9^{\pm4.7}$ | $86.0^{\pm3.7}$ | $75.0^{\pm6.3}$ | $73.4^{\pm6.7}$ | $72.4^{\pm7.0}$ | $73.5^{\pm7.8}$ |
| GK | $15.8^{\pm4.5}$ | $15.4^{\pm3.7}$ | $14.9^{\pm6.2}$ | $17.7^{\pm8.0}$ | $77.1^{\pm6.3}$ | $70.2^{\pm2.6}$ | $70.2^{\pm4.4}$ | $67.0^{\pm6.6}$ |
| RWK | $74.7^{\pm5.8}$ | $73.8^{\pm6.6}$ | $74.7^{\pm7.4}$ | $74.2^{\pm4.1}$ | $66.5^{\pm2.3}$ | $66.5^{\pm2.3}$ | $66.5^{\pm2.3}$ | $66.5^{\pm2.3}$ |
| ODD | $67.9^{\pm6.5}$ | $69.2^{\pm7.6}$ | $64.7^{\pm8.2}$ | $63.8^{\pm9.0}$ | $64.9^{\pm4.0}$ | $64.9^{\pm4.0}$ | $64.9^{\pm4.0}$ | $64.9^{\pm4.0}$ |
| VH | $86.4^{\pm4.0}$ | $\mathbf{86.9}^{\pm4.2}$ | $\mathbf{87.8}^{\pm4.5}$ | $\mathbf{87.8}^{\pm4.0}$ | $66.5^{\pm2.3}$ | $66.5^{\pm2.3}$ | $66.5^{\pm2.3}$ | $66.5^{\pm2.3}$ |
| LT | $15.8^{\pm5.0}$ | $13.1^{\pm4.3}$ | $13.1^{\pm5.1}$ | $14.5^{\pm4.5}$ | $69.7^{\pm3.9}$ | $68.1^{\pm2.5}$ | $66.5^{\pm2.3}$ | $68.6^{\pm8.5}$ |
| ST | $13.6^{\pm0.2}$ | $13.6^{\pm0.2}$ | $13.6^{\pm0.2}$ | $13.6^{\pm0.2}$ | $75.0^{\pm2.7}$ | $72.9^{\pm3.7}$ | $72.3^{\pm3.9}$ | $72.3^{\pm3.9}$ |
| RGW | $79.7^{\pm0.1}$ | $76.1^{\pm0.1}$ | $59.8^{\pm0.1}$ | $60.2^{\pm0.1}$ | $81.8^{\pm0.1}$ | $80.3^{\pm0.1}$ | $77.7^{\pm0.1}$ | $76.1^{\pm0.0}$ |
| FGW | $\mathbf{87.4}^{\pm4.4}$ | $86.9^{\pm4.7}$ | $63.8^{\pm5.8}$ | $62.0^{\pm5.0}$ | $\mathbf{85.6}^{\pm5.6}$ | $83.5^{\pm5.7}$ | $79.8^{\pm6.5}$ | $76.6^{\pm6.8}$ |
| FUGW | $73.8^{\pm6.1}$ | $68.3^{\pm4.9}$ | $5.0^{\pm3.7}$ | $5.4^{\pm4.4}$ | $82.4^{\pm5.5}$ | $81.9^{\pm5.6}$ | $\mathbf{81.9}^{\pm6.1}$ | $80.3^{\pm6.0}$ |
| FPGW (ours) | $\mathbf{87.0}^{\pm4.7}$ | $\mathbf{88.3}^{\pm4.1}$ | $\mathbf{87.3}^{\pm4.4}$ | $86.9^{\pm4.7}$ | $\mathbf{85.6}^{\pm5.6}$ | $\mathbf{85.1}^{\pm5.9}$ | $\mathbf{84.6}^{\pm5.6}$ | $\mathbf{82.5}^{\pm6.3}$ |

| | PROTEINS | | | | SYNTHETIC | | | |
|---|---|---|---|---|---|---|---|---|
| | 0% | 10% | 20% | 30% | 0% | 10% | 20% | 30% |
| Propagation | $59.6^{\pm0.2}$ | $59.6^{\pm0.2}$ | $59.6^{\pm0.2}$ | $59.6^{\pm0.2}$ | $48.0^{\pm7.0}$ | $51.3^{\pm6.5}$ | $47.3^{\pm5.5}$ | $53.0^{\pm6.1}$ |
| GraphHopper | $69.7^{\pm3.8}$ | $68.2^{\pm4.8}$ | $67.3^{\pm3.5}$ | $67.2^{\pm4.7}$ | $86.0^{\pm5.3}$ | $78.7^{\pm9.5}$ | $\mathbf{70.3}^{\pm7.1}$ | $\mathbf{70.7}^{\pm10.7}$ |
| RGW | $59.6^{\pm0.0}$ | $47.3^{\pm0.1}$ | $44.7^{\pm0.1}$ | $51.4^{\pm0.1}$ | $46.0^{\pm0.1}$ | $50.0^{\pm0.0}$ | $50.0^{\pm0.0}$ | $49.7^{\pm0.0}$ |
| FGW | $\mathbf{72.0}^{\pm3.7}$ | $68.5^{\pm3.1}$ | $65.3^{\pm4.4}$ | $65.8^{\pm4.4}$ | $\mathbf{97.7}^{\pm4.0}$ | $\mathbf{95.7}^{\pm2.9}$ | $48.7^{\pm1.6}$ | $49.3^{\pm0.7}$ |
| FUGW | $70.6^{\pm3.7}$ | $\mathbf{69.7}^{\pm2.7}$ | $69.0^{\pm3.3}$ | $68.7^{\pm2.4}$ | $60.0^{\pm6.1}$ | $48.3^{\pm6.7}$ | $45.3^{\pm4.5}$ | $45.3^{\pm4.5}$ |
| FPGW (ours) | $\mathbf{72.0}^{\pm3.7}$ | $\mathbf{71.6}^{\pm4.0}$ | $\mathbf{71.2}^{\pm4.9}$ | $\mathbf{69.2}^{\pm5.5}$ | $\mathbf{97.7}^{\pm4.0}$ | $\mathbf{96.3}^{\pm3.1}$ | $\mathbf{97.7}^{\pm4.0}$ | $\mathbf{94.3}^{\pm4.5}$ |

Regarding FUGW, its performance on the Mutag/Proteins datasets is comparable to that of FPGW. However, on the SCRC/Synthetic datasets, FUGW's accuracy is significantly lower than that of FGW and FPGW.

We refer to Section N.1 for the parameter settings and wall-clock time comparison. In summary, focusing on the comparison between FGW, FUGW, and FPGW, FGW is slightly faster than FPGW, while both FGW and FPGW are significantly faster than FUGW.

## N.1 NUMERICAL DETAILS IN GRAPH CLASSIFICATION

**Parameter and Numerical Settings.**

The parameter settings are provided in Table (6). For methods not explicitly listed, we use the default values from the GraKeL library.

For the FUGW method, we adapt the solver from Flamary et al. (2021). We test different marginal penalty parameters $\rho$ to ensure that the transported mass in each sampled pair is approximately 1. Additionally, we choose the smallest entropy regularization term $\epsilon$ that prevents NaN errors.

**Wall-clock time analysis.**

The wall-clock times are reported in Table 5. All graph data are formatted as NetworkX graphs using the NetworkX library. Continuous features are represented as 64-bit float NumPy vectors, while discrete features are stored as 64-bit integers.

For discrete feature graphs, WLK, GK, and Vertex Histogram are the fastest methods, while FGW and FPGW have similar wall-clock times. In contrast, Lovász Theta and Random Walk Kernel are the slowest. For continuous feature graphs, "Propagation" and "GraphHopper" are significantly faster than FGW and FPGW.

Table 5: Kernel computation times (minutes) across different datasets and noise levels

| Method | 0% | 10% | 20% | 30% |
|---|---|---|---|---|
| **SYNTHETIC** | | | | |
| Propagation | 0.01 | 0.02 | 0.01 | 0.02 |
| GraphHopper | 5.69 | 5.11 | 5.32 | 8.31 |
| RGW | 4.3 | 4.5 | 4.7 | 5.1 |
| FGW | 7.20 | 13.23 | 16.80 | 22.81 |
| FUGW | 41.00 | 40.53 | 35.32 | 37.88 |
| FPGW | 13.23 | 14.66 | 17.03 | 19.44 |
| **PROTEINS** | | | | |
| Propagation | 0.02 | 0.02 | 0.02 | 0.03 |
| GraphHopper | 7.06 | 7.43 | 7.79 | 8.16 |
| RGW | 844.7 | 804.1 | 766.8 | 713.2 |
| FGW | 26.05 | 28.71 | 29.83 | 30.17 |
| FUGW | 96.78 | 186.93 | 204.20 | 220.96 |
| FPGW | 62.03 | 65.99 | 71.54 | 73.19 |
| **MSRC** | | | | |
| WLK (auto) | 0.00 | 0.00 | 0.00 | 0.00 |
| GK (k=3) | 0.04 | 0.04 | 0.04 | 0.04 |
| RWK | 70.79 | 71.01 | 100.29 | 90.90 |
| Odd Sth (k=3) | 0.05 | 0.05 | 0.05 | 0.05 |
| Vertex Histogram | 0.00 | 0.00 | 0.00 | 0.00 |
| Lovasz Theta | 151.14 | 313.54 | 298.87 | 228.57 |
| SVM Theta | 0.00 | 0.00 | 0.00 | 0.00 |
| RGW | 3.0 | 3.2 | 3.4 | 3.6 |
| FGW | 4.03 | 4.31 | 4.63 | 5.05 |
| FUGW | 44.09 | 44.16 | 41.10 | 42.30 |
| FPGW | 4.54 | 4.95 | 5.17 | 5.77 |
| **MUTAG** | | | | |
| WLK (auto) | 0.00 | 0.00 | 0.00 | 0.00 |
| GK (k=3) | 0.03 | 0.03 | 0.03 | 0.03 |
| RWK | 211.48 | 61.07 | 66.10 | 305.90 |
| Odd Sth (k=3) | 0.00 | 0.00 | 0.00 | 0.00 |
| Vertex Histogram | 0.00 | 0.00 | 0.00 | 0.00 |
| Lovasz Theta | 22.48 | 19.14 | 18.67 | 13.82 |
| SVM Theta | 0.00 | 0.00 | 0.00 | 0.00 |
| RGW | 32.6 | 31.2 | 30.2 | 27.5 |
| FGW | 0.79 | 0.83 | 0.87 | 0.91 |
| FUGW | 19.70 | 21.76 | 20.98 | 17.69 |
| FPGW | 1.09 | 1.54 | 3.62 | 9.21 |

Table 6: Parameter setting of all methods in graph classification. We present the parameter settings in all the methods. $\sigma$ is the weight parameter in the SVM classifier. In FGW and FPGW, $\alpha$ parameter is the $\omega_2$ in formulations of FGW (7) and FPGW (10). For WLK/FGW/FUGW/FPGW, the value $H$ is the Weisfeiler-Lehman labeling parameter. $\rho, \epsilon$ in $FUGW$ is the weight parameter for the marginal penalty and entropy regularization. Parameter $\kappa$ in the GK/ODD method is the graphlet size. "n-samples" in GK is the random draw sample size.

| DATA SET | PARAMETER | SYNTHETIC | PROTEINS | MUTAG | MSRC-9 |
|---|---|---|---|---|---|
| SVM (COMMON) | $\sigma$ | 1 | 15 | 2 | 1 |
| RGW | $\rho$ | 0.1 | 0.1 | 0.1 | 0.1 |
| | $\epsilon$ | 0.05 | 0.05 | 0.05 | 0.05 |
| | T | 0.1 | 0.1 | 0.1 | 0.1 |
| | $\tau$ | 0.1 | 0.1 | 0.1 | 0.1 |
| FGW | $\alpha$ | 0.5 | 0.5 | 0.5 | 0.5 |
| | H | – | – | 2 | 4 |
| FPGW | $\alpha$ | 0.5 | 0.5 | 0.5 | 0.5 |
| | H | – | – | 2 | 4 |
| FUGW | $\alpha$ | 0.5 | 0.5 | 0.5 | 0.5 |
| | H | – | – | 2 | 4 |
| | $\rho$ | 1 | 1 | 0.4 | 0.4 |
| | $\epsilon$ | 0.05 | 0.1 | 0.02 | 0.02 |
| WLK | H | – | – | 5 | 5 |
| GK | $\kappa$ | – | – | 3 | 3 |
| | N-SAMPLES | – | – | 100 | 100 |
| ODD STH | $\kappa$ | – | – | 3 | 3 |

Among the three fused-GW-based methods, FGW is the fastest overall, while FUGW is the slowest. A potential reason for this difference is that FGW and FPGW leverage a C++ linear programming solver for the OT/POT solving step. In addition, FUGW adapts the Sinkhorn solver, leading to an accuracy-efficiency trade-off. Specifically, a smaller $\epsilon$ can reduce the accuracy gap introduced by the entropic term; however, it increases the number of iterations required for convergence.

All experiments presented in this paper are conducted on a computational machine with an AMD EPYC 7713 64-Core Processor, $8 \times 32$GB DIMM DDR4, 3200 MHz, and an NVIDIA RTX A6000 GPU.

## O  PARAMETER SENSITIVITY AND SELECTION STRATEGY

This section summarizes the strategy used for parameter selection across all experiments.

**FMPGW.**  In the FMPGW formulation, the parameter $\rho$ determines the amount of mass being transported. When prior knowledge is available (e.g., in graph matching, where the smaller graph should be fully transported), we set $\rho$ equal to the total mass of the smaller graph. Otherwise, $\rho$ is treated as a tunable parameter, and its value is selected via a grid search, following the same procedure used in FUGW and UGW.

**FPGW.**  In the FPGW formulation, the parameter $\lambda$ acts as a *soft upper bound* on the transport cost. Intuitively speaking, a higher $\lambda$ induces a plan transferring more mass. When $\lambda$ exceeds a certain threshold (see Eq. (103)), the solution transports $\min(|\mu|, |\nu|)$, corresponding to full-mass transport. Therefore, in experiments where full transportation of the smaller graph is required (e.g., graph matching), we select a sufficiently large $\lambda$. In all other settings, $\lambda$ is tuned via grid search, similar to UGW and FUGW.

**Other baseline methods.**  For baselines such as FUGW and UGW, we perform a grid search over their respective parameters (e.g., $\rho_1, \rho_2$) and report the configuration achieving the best performance. Other control parameters, including the maximum number of iterations and stopping thresholds, are kept at their default values. For the entropic regularization weight, we consistently use the smallest

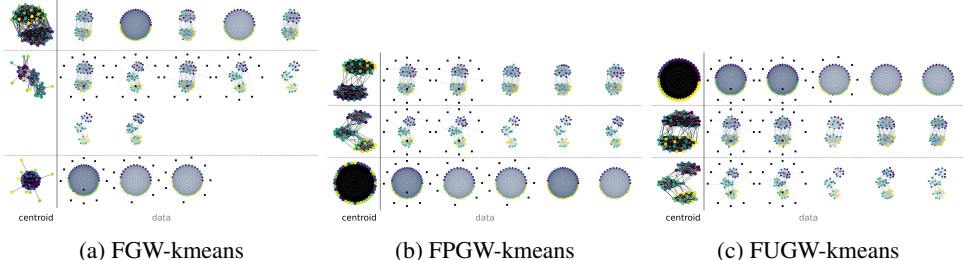

(a) FGW-kmeans      (b) FPGW-kmeans      (c) FUGW-kmeans

Figure 7: We present the clustering results of the FGW, FPGW, and FUGW k-means methods.
In the first column, we visualize the centroids obtained using three methods. The centroids are represented by their features and structure (distance matrix). The edges of these graphs are either reconstructed or approximated based on the returned distance matrix. Additionally, the color of each node corresponds to the feature it represents.
The clustering results are shown in the remaining columns. For each graph, the color of regular nodes represents their features, while all outlier nodes are depicted as black squares.

value that avoids numerical instability (e.g., NaN errors); Excessively large regularization leads to inaccurate solutions, whereas overly small values can cause divergence.

## P EXTRA RESULT IN GRAPH CLUSTERING.

In graph clustering expriment, we set $\omega = 0.999$, following the setting from Vayer et al. (2020), and present the result.

The dataset and parameter setting is same to the clustering experiment in the main text. The only different is, we set $\omega = 0.999$.

Regarding the wall-clock time, FGW requires 41.9 seconds, FPGW requires 82.7 seconds, while FUGW requires 1456.4 seconds.

## IMPACT STATEMENT

This paper aims to advance the theoretical foundations and potential applications of Optimal Transport in the field of Machine Learning. There are many potential societal consequences of our work, none of which we feel must be specifically highlighted here.

