# OpenReview forum: "FUSED PARTIAL GROMOV-WASSERSTEIN FOR STRUCTURED OBJECTS"
_ICLR.cc/2026/Conference — Submitted to ICLR 2026_

### Official Review · Reviewer_WxFv · 2025-10-26

**Soundness:** 3
**Presentation:** 3
**Contribution:** 2
**Rating:** 4
**Confidence:** 3

**Summary:**

This paper introduces the Fused Partial Gromov-Wasserstein (FPGW) formulation, which combines Partial Optimal Transport (POT) and Partial Gromov-Wasserstein (PGW). The FPGW model allows the comparison of structured objects with unequal total mass. To solve the FPGW problem efficiently, the authors develop optimization methods based on the Frank–Wolfe and Sinkhorn algorithms. Experimental results on graph matching, clustering, and classification tasks demonstrate that FPGW achieves more robust and consistent performance over existing baselines.

**Strengths:**

- The proposed formulation is inspired by the Fused Gromov–Wasserstein model and effectively incorporates both node features and structural information for graph comparison.
- By introducing partial matching into the fused formulation, the method can handle structured objects with unequal mass, which is meaningful.
- The paper is generally well written and well organized.
- Experimental results show the robust and consistent performance of the proposed approach over baselines across multiple tasks.

**Weaknesses:**

- The fusion of a linear term (from OT or partial OT) and a quadratic term (from GW or partial GW) has already been explored in prior literature. The proposed FPGW appears to build mainly upon the Partial Gromov–Wasserstein (PGW) model [1], and the theoretical results (e.g., Theorem 3.1) seem to be direct corollaries of those in PGW.
- From the fusion perspective, FPGQ of this paper and PGW of [1] replaces the KL divergence used in fused unbalanced GW [2] with total variation for the regularization function D_{\Phi_1}. Please elaborate more on the advantage of using TV over KL.
- The proposed FW and Sinkhorn algorithms for FPGW appear to be straightforward adaptations of those from [1]. In addition, in the graph matching experiments, only the Sinkhorn-based FPGW (first mentioned in line 379, presumably Algorithm 2) was used for comparisons.
Please clarify why the FW variant was not evaluated.
- In the toy graph clustering experiment, the setting is not convincing. The weight \omega_2 was set as 0.999, which means that the contribution of linear term (OT cost) was negligible. Then it becomes PGW, which is not original contribution of this paper. Hence, this example fails to demonstrate the distinct effectiveness of the fused formulation.
- In other experiments (e.g., graph matching), the selection of \omega parameters was not explained.

Below are some minor things:
- In Eq (12), the constraint should be \gamma \in T_{\leq} (p,q).
- In Eq (16), the left hand side should be F(\gamma, \pi).
- In Proposition 3.2 (line 252), the constraint should read \gamma \in \in T_{\leq} (\muy,\nuy).
- In line 1001, where is the proposition L.2.
- In line 1375, should be |\gamma| = \rho.
- In line 1679, should use \leq instead of "="
- In line 1858, left-hand side should be F(\gamma, \pi).
- In line 1892, use \gamma \times \pi instead of \gamma \bigotimes \pi
- In line 1990, the objective is missing--only the defintion of the constraint set is given.
- In line 1982, a ">" is missing for the first term.







[1] Bai et al,. Partial Gromov-Wasserstein Metric, ICLR 2025.

[2] Thual et al, Aligning individual brains with Fused Unbalanced Gromov-Wasserstein, NeuRIPS 2022.

**Questions:**

see the weaknesses.

---

> ### Author Response · Authors · 2025-11-17
> **Weakness: 1, 2, 3**
>
> > The fusion of a linear term (from OT or partial OT) and a quadratic term (from GW or partial GW) has already been explored in prior literature. The proposed FPGW appears to build mainly upon the Partial Gromov–Wasserstein (PGW) model [1], and the theoretical results (e.g., Theorem 3.1) seem to be direct corollaries of those in PGW.
>
> Answer: We agree that our formulation is closely related to PGW [1] and that the proofs share some techniques. In particular, for the metric property, we follow a similar strategy: we convert the problem into an FGW-like setting and then establish a (relaxed) triangle inequality.
>
> However, the results are not direct corollaries. In the FPGW setting, the cost structure and the mass-penalty term lead to a **relaxed** triangle inequality (See Theorem D.3) that differs from the PGW case. As a consequence, both the statement and the proof require new arguments to handle the fused term and the partial mass constraint simultaneously. We have clarified these distinctions in the revised version and in the “Novelty / Contributions’’ section of the official response.
>
>
> > From the fusion perspective, FPGW of this paper and PGW of [1] replace the KL divergence used in fused unbalanced GW [2] with total variation for the regularization function $D_{\Phi_1}$. Please elaborate more on the advantages of using TV over KL.
>
> Answer: We refer the the "TV vs. KL" part of the official comment and summarize here:
>
> - Model: TV penalizes the absolute mass difference in a linear way, which is natural for partial/trimmed matching where one wants to exactly discard some mass. KL, in contrast, is asymmetric and tends to encourage smooth mass spreading, which is less suited to strict partial matchings.
>
> - Theory: TV fits naturally in our metric analysis; it leads to a (semi-)metric structure for FPGW. In contrast, metric properties of FUGW (with KL penalties) remain unclear.
> - Numerics: TV-based penalties give rise to simpler updates inside the Sinkhorn iterations (partial OT with TV), whereas KL-based unbalanced GW involves Hellinger–Kantorovich-type updates that are more complex. Empirically, this leads to faster convergence and less mass-splitting for FPGW.
>
> We have added explicit TV vs. KL comparisons (accuracy and runtime) in Appendix O.
>
> > The proposed FW and Sinkhorn algorithms for FPGW appear to be straightforward adaptations...
>
> Answer:
> ## About “Sinkhorn adaptations”.
> While the high-level structure of the algorithms resembles the PGW and UGW solvers, applying FW and Sinkhorn to FPGW is not merely symbolic substitution. The cost, constraint set, and inner subproblems differ:
>
>    - In FUGW [2], the inner loop solves a Hellinger–Kantorovich-type problem:
> $$ \min_{\gamma} \int c_\pi^\epsilon(x,y) d\gamma(x,y)+\rho |\pi| KL(\gamma_1 \parallel \mu) + \rho |\pi|  KL(\gamma_2 \Vert \nu) + \epsilon |\gamma| + H(\gamma \parallel \mu \otimes \nu),
> $$
> where
> $$c_\pi^\epsilon(x,y)=\int |d_X(\cdot,x)-d_Y(\cdot,y)| d\pi+\rho \int \ln \frac{d\pi_1}{d\mu}d\pi_1+\rho \int \ln \frac{d\pi_2}{d\nu}d\pi_2+\epsilon \int \ln \frac{d\pi}{d\mu\otimes \nu}d\pi.
> $$
>
> - In FPGW, the inner loop solves a **partial OT** problem:
> $$ \min_{\gamma \in \Gamma_{\le}(\mu,\nu)} \int_{X \times Y} c_\pi(x,y)d\gamma+ \lambda |\pi| (|\mu|+|\nu|-2|\gamma|) + \epsilon |\pi| D_{KL}(\gamma\parallel \mu \otimes \nu), $$
> where
> $$c_\pi=\frac{1}{2}\omega_1 d(x,y)+\omega_2 \int |d_X(\cdot,x)-d_Y(\cdot,y)|^2d\pi +\epsilon |\pi|D_{KL}(\gamma\parallel \mu\otimes \nu).$$
>
> Thus, the inner Sinkhorn updates differ from both PGW and FUGW. We have made these distinctions more explicit in the revised manuscript.
>
> > Please clarify why the FW variant was not evaluated.
>
> Answer: For graph matching, we aim to recover a **partial matching plan**. In this highly non-convex setting, FW is prone to getting stuck in poor local minima. Entropic regularization improves the optimization landscape and numerical stability, so we use the Sinkhorn-based EFPGW solver in matching experiments. Conversely, for tasks where the transport cost itself is used (e.g., as a kernel for graph classification), FW is preferable and is evaluated there. This “FW vs. Sinkhorn’’ trade-off is now clearly described in the updated version. In the updated manuscript, we've added FMPGW-FW as additional baseline. Its performance "20-60 accuracy" is not as good as FUGW-Sinkhorn or FPGW-Sinkhorn.

---

> ### Author Response · Authors · 2025-11-17
> **Weakness: 4-5**
>
> > In the toy graph clustering experiment, the setting is not convincing...
>
> Answer:
>
> We originally followed the setting in Fused GW [1], where $\omega_2 = 0.999$ is used to emphasize structural information. Our goal was to reproduce the FGW baseline as closely as possible.
>
> In the revised version, **we added experiments with $\omega_2 = 0.5$** (i.e., balanced feature and structure weights). The updated results show that:
> - FUGW and FPGW remain more robust to outliers than purely balanced formulations;
> - FPGW achieves comparable or better accuracy while being significantly faster.
>
> See the following numerical results:
>
>  [result for FGW, $\omega_2=0.5$](https://anonymous.4open.science/r/fused-pgw-041B/clustering/res_0.5/fgw_0.5.png)
>
>  [result for FPGW, $\omega_2=0.5$](https://anonymous.4open.science/r/fused-pgw-041B/clustering/res_0.5/fpgw.png)
>
>  [result for FUGW, $\omega_2=0.5$](https://anonymous.4open.science/r/fused-pgw-041B/clustering/res_0.5/fugw.png)
>
> We refer the section 4.1 in the updated manuscript for the detailed settings. The experiment results for $\omega_2=0.999$ have been moved to the appendix P.
>
>
> These results better highlight the effect of the fused formulation and are reported in the updated manuscript.
>
>
> > In other experiments (e.g., graph matching), the selection of \omega parameters was not explained.
>
> Answer: We clarify all the parameter settings in the appendix. We parameterize $\omega_1, \omega_2$ as $1 - \alpha, \alpha$:
>
> - For graph and geometry clustering, the exact values are listed in Table 2;
> - For graph classification, the values are listed in Table 6.
>
> In most datasets, we use $\alpha = \omega_2 = 0.5$, giving equal importance to features and structure. This is now clearly stated in the revised text.

---

> ### Author Response · Authors · 2025-11-17
> **minor issues and corrections**
>
> >  In Eq. (12), the constraint should be $\gamma \in T_{\leq}(p,q)$.
>
> Answer: FIxed.
>
> > In Eq. (16), the left-hand side should be $F(\gamma,\pi)$.
>
> Answer: Fixed.
>
> > In Proposition 3.2 (line 252), the constraint should read $\gamma \in T_{\leq}(\mu_y,\nu_y)$.
>
> Answer:
> We thank the reviewer for these corrections. All three notational issues have been fixed in the revised manuscript.
>
>
> > In line 1001, where is Proposition L.2?
>
> Answer: There is no Proposition L.2. We have replaced the reference by a direct argument: “From the definition, we have $\gamma' \le \gamma$ and $\gamma'_1 \le \mu$. From Inequality (22) in [1], we obtain the following inequality...”
>
> [1] Bai et al., Partial Gromov–Wasserstein Metric, ICLR 2025.
>
>
> > In line 1375, it should be $|\gamma| = \rho$.
>
> Answer:
> We have rewritten the definition of $\gamma^\rho_\leq(\mu,\nu)$. The correct condition should be $|\gamma|\ge \rho$.
>
> However, whether the constraint is $|\gamma| = \rho$ or $|\gamma| \ge \rho$, the optimizer lies on the boundary $|\gamma| = \rho$. So we do not distinguish them in the initial submission.
>
>
>
> > In line 1679, use $\leq$ instead of “=”.
>
> Answer:
> In line 1679, there is no inequality; if the reviewer meant line 1689, then it is indeed an inequality based on $ab \le |a||b|$. We have double-checked the lines and clarified the relevant inequality in the proof.
>
>
> > In line 1858, the left-hand side should be $F(\gamma,\pi)$.
>
> Answer:
> Thanks. Corrected.
>
>
>
> > In line 1892, use $\gamma \times \pi$ instead of $\gamma \otimes \pi$.
>
> Answer:
> We thank the reviewer. In the revised version, we rewrite Section I in terms of relative entropy rather than KL divergence and adopt consistent notation (using $\gamma \times \pi$) to avoid confusion in the unbalanced setting.
>
>
> > In line 1990, the objective is missing; only the constraint set is given.
>
> Answer:
> We corrected this by explicitly stating the full optimization problem. We also replace “where’’ with “set’’ when introducing $C_1, C_2, C_3$, emphasizing that these are auxiliary sets used to define the mass constraint in partial OT.
>
>
> >  In line 1982, a “$>$’’ is missing for the first term.
>
> Answer: Fixed.
>
>
> ## Clarification about FMPGW solvers and contribution scope.
>
> As stated in our official comment, the FW and Sinkhorn solvers for fused-MPGW were introduced around mid-September 2025, while our submission date was September 30, 2025. In the revised paper, we clearly state that:
>
> - FMPGW and its FW/Sinkhorn solvers are **not** claimed as our contributions; (Although we provided related FW convergence analysis, Barycenter algorithm, which we can neither find a reference nor code implementations.)
> - we include them only for theoretical completeness and comparison;
> - Our contributions focus on the FPGW formulation, its metric analysis, its FW and Sinkhorn solvers, and the barycenter extension.

---

> > ### Comment · Reviewer_WxFv · 2025-11-27
> > **Thank you for the rebuttal**
> >
> > I highly appreciate the authors' efforts to improve the quality of the manuscript. Accordingly, I have raised the score from 4 to 6.

---

> > > ### Author Response · Authors · 2025-11-29
> > >
> > > Thank you for the helpful and constructive discussion. We sincerely appreciate the reviewer’s insightful comments and feedback.

---

### Official Review · Reviewer_nywn · 2025-10-28

**Soundness:** 3
**Presentation:** 3
**Contribution:** 1
**Rating:** 2
**Confidence:** 5

**Summary:**

In this paper the authors propose a partial OT version of the existing Fused Gromov
Wasserstein (FGW) distance, called Fused Partial Gromov Wasserstein. The FGW
distance is a metric to compare structured objects (e.g. graphs) with node
features. The proposed distance allows to compare only a fraction of the total
mass of the two distributions, which is useful when the two objects to compare
have outliers, different proportions or only part of objects to be aligned. The
authors propose two variants of the problem, one with a hard constraint on the
amount of mass to be matched, and one with a Total Variation penalty. The they
discuss the theoretical properties of the proposed distance, and propose a Frank
Wolfe and entropic/Sinkhorn optimization scheme to compute it. Finally, they
demonstrate the interest of the proposed distance on several experiments (graph
clustering, partial graph matching, geometry matching) on both synthetic and
real data.

**Strengths:**

+ The paper is overall well written and the proposed method is clearly
  explained.

+ In applications, one might have access to structured objects with feature on the
  nodes that are only partially comparable. The proposed distance is a useful
  extension of the existing FGW distance to handle such cases.

+ The experiments suggests that the proposed distance indeed work better than FGW
  when only part of the objects are comparable.

**Weaknesses:**

+ The contribution novelty is extremely limited. The proposed distance is a
  straightforward extension of the existing FGW distance (Vayer 2019) to the
  partial case, following the same principles as the Partial GW with only a
  linear term added from (Chapel 2020). While this is obviously useful in
  practice, the adaptation is easy to implement and probably already used in the
  community (see next point).

+ The optimization schemes are also straightforward  adaptations of existing
  schemes for Partial GW (the Alg. 1 Frank Wolfe) and Unbalanced GW of (Sejourne
  2021) for the Alg. 2 sinkhorn/entropic version. As a matter of fact the
  Partial FGW (Frank Wolfe) is already implemented in the  POT library  at the
  following URL:

  https://pythonot.github.io/gen_modules/ot.gromov.html#ot.gromov.partial_fused_gromov_wasserstein

  and the entropic version is available here:

  https://pythonot.github.io/gen_modules/ot.gromov.html#ot.gromov.entropic_partial_fused_gromov_wasserstein

  While those two implementations above might not solve exactly the formulation
  (but something equivalent), it shows that the proposed method and/or
  formulation cannot really be claimed as an original contribution.

  See the example of the POT library for use of the Partial FGW solver:
  https://pythonot.github.io/auto_examples/gromov/plot_partial_fgw.html


+ All theoretical results are interesting and have probably not specifically
  been investigated in the literature but also seem like simple adaptations of
  existing results, metric properties. If really new proving strategies have
  been devised  the authors should have highlighted them

+ Numerical experiments show that FPGW works better than FGW when only part of
  the objects are comparable, which is expected. However they have been
  specifically designed to make FGW fail (e.g. partial alignment). The Graph
  clustering example is just the experiment from FGW paper with outlier nodes
  added. The graph matching setup where part of the graphs are removed seem to
  be far from any realistic application. The geometry matching example is a
  beautiful visualization but again does no correspond to any real application.
  Overall the experiments lack diversity and realism.

**Questions:**

Please discuss weaknesses above.

---

> ### Author Response · Authors · 2025-11-18
> **Weakness 1, 2, 3**
>
> > W1: The contribution novelty is extremely limited.
>
> Answer: We refer to the "novelty and contribution" section in the official comment for details.
>
> >W1: The proposed distance is a straightforward extension of the existing FGW distance ([1] Vayer 2019) to the partial case, following the same principles as the Partial GW with only a linear term added from ([2] Chapel 2020). While this is obviously useful in practice, the adaptation is easy to implement and probably already used in the community (see next point).
>
> Answer:
>
> Regarding the Sinkhorn algorithm, we refer to our answer in “Sinkhorn adaptation” for reviewer WxFv for details.
>
> Regarding the FW algorithm, although FMPGW and FPGW share the general FW structure, their formulations differ. Therefore, we must develop a new gradient computation step and a new line-search step tailored to FPGW. The overall framework is, of course, similar because both are FW-type algorithms, but the technical components are not directly inherited.
>
> > W2: The optimization schemes are also straightforward adaptations of existing schemes for Partial GW (the Alg. 1 Frank Wolfe) and Unbalanced GW of (Sejourne) for the Alg. 2 sinkhorn/entropic version.
>
> Answer: We refer to the answer to the previous question about the adaption of FW/Sinkhorn to the new FPGW setting for details.
>
>
> > W2: As a matter of fact the Partial FGW (Frank Wolfe) is already implemented in the POT library at the following URL: https://pythonot.github.io/gen_modules/ot.gromov.html#ot.gromov.partial_fused_gromov_wasserstein ...
>
>
> Answer:
> The released PythonOT FMPGW solver was introduced on *Sept. 14, 2025*, while the submission of our paper was on *Sept. 30, 2025*. We refer to the **“FMPGW vs. FPGW”** section in the official comment for a detailed explanation.
> In summary:
>
> - In Section 3, we explicitly state that **FMPGW is not our contribution** (although we provide a FW convergence analysis and a barycenter algorithm for FMPGW, for which we found no prior references). A clarifying remark has been added to Section 3 in the updated version.
> - At the time of submission, we were **not aware** that the newest PythonOT release already included FW and Sinkhorn solvers for FMPGW.
> - **Sinkhorn–FMPGW (PythonOT)** is significantly slower than our proposed **Sinkhorn–FPGW**.
>   We refer to Section O in the updated version and the **“FMPGW vs. FPGW”** section for detailed wall-clock comparisons.
>   When the graph size is **≥ 5000**, PythonOT’s Sinkhorn–FMPGW requires **more than 3 days**, whereas our Sinkhorn–FPGW requires only **2–3 hours**.
> - **FW–FMPGW (PythonOT)** and **FW–FPGW (ours)** have similar wall-clock time; however, due to the non-convexity of the FW formulation, **both perform poorly in the graph-matching experiment**.
>
> > W3: All theoretical results are interesting and have probably not specifically been investigated in the literature but also seem like simple adaptations of existing results, metric properties. If really new proving strategies have been devised the authors should have highlighted them
>
> Answer:
> To the best of our knowledge, the **metric property for UGW/FUGW remains unclear** in the literature.
> We provide a new theoretical result establishing the **metric property of FPGW**, thereby completing a missing component in the theoretical development of fused unbalanced/partial GW formulations.
>
> Regarding the proof strategy:
> We extend the argument introduced in **[5] Bai et al., 2024**, which reduces partial (fused) PGW to a special case of fused GW and proves the corresponding (relaxed) triangle inequality.
> However, because the **triangle inequality for FPGW is structurally different** from that of PGW, our conclusion is **not** a direct corollary of prior work. Instead, it requires adapting the proof technique and developing new steps specific to the FPGW formulation.

---

> ### Author Response · Authors · 2025-11-18
> **Weakness 4**
>
> > Numerical experiments show that FPGW works better than FGW when only part of the objects are comparable, which is expected. However they have been specifically designed to make FGW fail (e.g. partial alignment).
>
> Answer: In the Graph/Geometry matching experiment,  we utilize the real data proposed in [5]. This is not designed for PGW. Graph matching has its own impact, e.g., detecting if certain motif areas in a tissue in single-cell analysis.  In these type of experiment, FPGW/FUGW have average better performance and FPGW has faster speed in all the tested datasets.
>
> > The Graph clustering example is just the experiment from FGW paper with outlier nodes added. The graph matching setup, where part of the graphs are removed, seems to be far from any realistic application. The geometry matching example is a beautiful visualization but again does no correspond to any real application. Overall, the experiments lack diversity and realism.
>
> Answer:
> ## Graph clustering.
> We fully agree that the clustering experiment is a toy example. This is explicitly stated in the section title (and follows the FGW paper). Its purpose is to illustrate the behavior of the FPGW barycenter, not to serve as a realistic large-scale benchmark.
>
> ## Graph matching.
> In contrast, the graph matching experiments are conducted on real-world datasets and correspond to practically relevant tasks. Graph matching is a core motivation for our work, with applications in areas such as biological network alignment and social network matching, where the graphs are related but not identical and partial correspondence is natural.
>
> We follow the Robust GW setup (Kong et al., NeurIPS 2023)[4], using both standard TU benchmark graphs (e.g., PROTEINS) and larger real-world datasets (e.g., the Douban network). Applying FPGW to even more diverse real-world matching problems is part of our planned future work.
>
> ## Geometry matching.
> The geometry matching example is intended as a visual illustration of graph/subgraph matching, not as a standalone application. The dataset and experimental design follow Robust GW [4]. In the revised version, we:
>
> - move this experiment to a sub-section, and
> - clearly describe it as a visualization built on the same graph-matching framework.
>
> References:
>  [1] Vayer et al., Fused Gromov–Wasserstein Distance for Structured Objects, Algorithms, 2020.
>
> [2] Chapel et al., Partial Gromov–Wasserstein, ICLR 2021 (or related venue).
>
> [3] S´ejourn´e et al., Unbalanced Gromov–Wasserstein Distance, NeurIPS 2021.
>
> [4] Kong et al., Outlier-Robust Gromov–Wasserstein for Graph Data, NeurIPS 2023.
>
> [5] Bai et al., Partial Gromov Wassertein Metric, ICLR 2024

---

> ### Comment · Reviewer_nywn · 2025-11-27
> **Thank you for your responses**
>
> I appreciate the authors response and updates. I respond to a few of them and give my updated opinion below.
>
> **W1** I appreciate the clarification. The semi-metric proof is interesting but its importance for ML theory of application is limited. The formal relationship between the penalized vs contained version is also new on FGW but very classical in partial/TV OT. I agree that FUGW was also incremental similarly to the current paper but it had more depth in the numerical experiments on large scale brain fMRI alignment with comparison (and large gain) to SOTA alignment methods from neuroscience field which highlighted the potential impact of the method. As I said in my review I think that the experiments are a bit weak because they are designed specifically for PFGW and do not show the potential impact of the method on practical problems.
>
> **W2 and code** I agree that the timing for the code on github is very close to your submission and of course it would not be reasonable to require to follows code updates. It is very interesting that your implementation is faster and I appreciate the comparison, it makes the paper stronger.
>
> **W3** thanks for the clarification, I still believe that this is a minor extension of an existing result.
>
> **Experiments** I understand that some of these experiments have been proposed in other papers but that does not make them realistic or corresponding to real problems. Again the graph matching setup "simulates" missing parts which makes no sens physically and I fail to find real problems where this could happen. The fact that it is done on real graphs in not enough to show its practical utility. I am really interested by your answer about biological networks or single cell data analysis (that already have applications of OT and GW) that could show the practical interest of the method (or even better the barycenter). This would make the paper much stronger.
>
> In conclusion, I really appreciate the response of the authors and their honesty. But I believe that since the theoretical and novelty contributions is limited, the paper needs stronger  numerical experiments.  I'm not inclined to change my score but I encourage the authors to continue working on it because there are probably practical applications where the propose framework works better than existing methods.

---

> > ### Author Response · Authors · 2025-12-03
> >
> > Motivation: We needed an OT-related graph matching algorithm for our experiments, but found no suitable existing method. This led us to conduct a detailed literature search and ultimately develop the fused partial Gromov–Wasserstein (FPGW) formulation and corresponding algorithms.
> >
> > Novelty relative to prior work: While writing our paper, a similar algorithm, fused-FMPGW-Sinkhorn, appeared in PythonOT on Sept 25, which we refer to. We have claimed that the FMPGW is not our contribution in both the initial and updated submissions (although we provide the related Frank-Wolfe convergence analysis and Barycenter algorithm, which we cannot find in any references). However, the released implementation is extremely slow. (We refer to the wall-clock time comparison in the official comments. When the size of the data >=2000, FMPGW-Sinkhorn requires 2-3 days, and FPGW-Sinkhorn requires 2-3 hours.)
> >
> > Theoretical contribution: Our work fills a gap in the optimal transport literature by introducing the FPGW formulation and establishing its metric properties.
> >
> > Practical contribution: Our proposed FPGW algorithms achieve the best performance considering both accuracy and efficiency. Although FMPGW-Sinkhorn can theoretically achieve similar accuracy, its wall-clock time is significantly slower and is not feasible in the large graph setting.

---

### Official Review · Reviewer_kDTw · 2025-11-02

**Soundness:** 2
**Presentation:** 3
**Contribution:** 1
**Rating:** 2
**Confidence:** 4

**Summary:**

The article proposes an extension of the existing method based on the Gromov-Wasserstein (GW) distance (from Optimal Transport theory) to solve ML tasks for attributed graphs (also called structured data). The authors use the Fused GW distance, where the GW distance between the graphs (coded as the sets of distances in a graph) to be compared and the attribute distances between nodes related by the transport plan are added (with some weights to balance them). This Fused GW distance is already well developed in the literature from 2019 and 2020.

The article points to the specific extension where the graphs are unbalanced, so that one can not expect to have the same total distributions for the two graphs. This is called the Fused Partial GW problem  (+ the sub-variant of Mass-constrained PGW problem) and the question has also been well studied between 2020 and 2024.

The novelty here is on two aspects: the authors propose to use a different, specific form for age distance between the attributes, as being the Total variation (instead of KL divergence or L_q distances as in previous works). From that, there derive algorithms from the Franck-Wolfe method and the Sinkhorn algorithm to solve this specific instance of FPGW or FMPGW problems.

The authors have the result that the problem, formulated that way, provides a semi-metric. Then, they show examples on three ML task with graphs: clustering, graph matching, and geometry matching. For that, they use classical machinery for OT methods with graphs, computing barycenters for FPGW distances ; coding graphs as discrete mass measures ; selecting a specific node as anchor for geometry matching;  and so on. The results are good but they are not game changer on these questions.

**Strengths:**

The strengths of the article are :

1. It is well written, with (almost) all required elements defined (e.g., there are 2 pages of notations in the supplements, and background on OT and all the elements up to GW distances).

2. There is an effort to prove some theoretical properties: existence of minimiser for the unbalanced FGW or FPGW; properties related to being a metric or semi-metric;  details on the described algorithms and convergence properties.

3. I have found one point with some novelty: the fact that the proposed method allows to “put aside” nodes which are outliers in a graph, and not transport them to the other graph (as seen on Fig. 2). However, $N_G$ appear to be needed (the number of nodes which are not outliers) and this diminishes the usefulness of the method.

**Weaknesses:**

The article has several weaknesses and I find that they overcome the strengths of the article:

1. First of all, the work is way too **incremental**: the problem of FPGW (and its variants) was already well considered before, with methods to solve it and better discussions than in the present article about why the problem is relevant. Here, the main novelties are technical results and algorithms in a specific choice of distance ; and these technicalities and algorithms come almost straight from the literature.

2. The authors give **no element to justify whether their choices are better, or more sound, or just sound**, as compared to previous works. For instance, why use a total variation ? Is there a reason for that ? When to use the FW algorithm to the Sinkhorn version ? Why is the  Sinkhorn one mostly used in the experiments ? (and then, why propose also the FW version ?)

3. The numerical experiments are not that illuminating. They are on specific questions and situations already well known. The geometry matching one for instance is a niche topic, here on only one 3d mesh.The graph datasets used for clustering or graph matching are small.

4. For graph clustering, did not understand how the parameters were chosen. The number of non-outliers node appear to be known. How can that relate to any practical situation ? How were set the other parameters ? Also, it is written on page 7:
> This robustness is attributed to the partial matching property of FPGW.

Why that ? Any proof of this assertion ?

5. In 4.2, is the fact that only 50% of the nodes are extracted a known fact ? It seems that globally the method is only valid if each node corresponds to a node in the other graph, while an interest of the OT approach for ML on graphs is that it allows a change of scale in some situations (one node might be matched to 5 or 10 in the second graph, if need be). Also, knowing $N_G$ in advance seems to be a weak point of the method.

6. Table 1 has to be re-considered: are the std of the results (in %) precise up to 2 digits  ? How come some methods have a ±0.00 std ? This is not statistically sound. What is the unit of Time ? The results should include the FW algorithm.

7. The example in 4.3 (geometry matching) is of limited impact. Also, the underlying task (“geometry matching task”) is not really detailed.

8. A limitation of methods based on Fused GW is often the computation time. Is it also the case here ? Or not because of the sinkhorn version ? What is the theoretical complexity and its scaling ?
The used datasets for 4.1 and 4.2 are not fully described but they are quite small graphs. Can the method be extended to larger graphs (in terms of nodes) ?

9. In the results, FUGW appear to have often similar performance than sink-FPGW (in Table 1). Is it the case ? Then, the contribution is about acceleration by using a sinkhorn algorithm, no ? Why is this acceleration not obtained on the Douban dataset ? Is is related to  the size of this dataset ?

**Questions:**

I have a small list of suggestions or questions, that will not change the weaknesses that I see in this article :

- it seems that the definition of the total variation is not given (despite the 2 pages of notations). Also, it comes as $ \Vert . \Vert_{TV}$ or $ | . |_{TV}$ depending on the pages.

- Titouan et al. (2019a) should be Vayer et al. (2019a)

- Some misprints (examples: “postivie” on page 5 ; “RAPAMETER” in title M. ; “As shown in Table 2” on page 8 -> should be Table 1)

-  Add some ablative or comparative study to assess whether using the TV distance is sound for this problem.

- Compare the two proposed algorithms in many situations, and compare them better to existing approaches. For instance, it is not useful to compare them to methods assuming balanced sets, as this topic (partial/unbalanced vs balanced problem, has already been discussed in the literature). But the 2 proposed algorithms and ways to estimated the unbalances (+ ways to choose the parameters) should be compared to the literature.

---

> ### Author Response · Authors · 2025-11-18
> **Weakness 1-4**
>
> > W1 First of all, the work is way too incremental...
>
> Answer:
> ### Contribution
> We refer to the **“Novelty/Contributions”** section and the **“FPGW vs. FMPGW”** section in the official comment for a detailed discussion of the novelty of this work.
> To the best of our knowledge, the mathematical properties of FPGW—such as its **metric property** and the **formal convergence analysis** of its FW solvers and the **barycenter algorithm** (for clustering)—have not been explicitly studied in the literature. These constitute the primary contributions of this work.
>
>
> > W1. and these technicalities and algorithms come almost straight from the literature.
>
> Answer: Regarding the Sinkhorn algorithm, we refer to our answer in **“Sinkhorn adaptation”** for reviewer *WxFv* for details.
> Regarding the FW algorithm, although FMPGW and FPGW share the general FW structure, their formulations differ. Therefore, we must develop a **new gradient computation step** and a **new line-search step** tailored to FPGW. The overall framework is of course similar because both are FW-type algorithms, but the technical components are not directly inherited.
>
>
> >W2.The authors give no element...  why use a total variation?
>
> Answer: We refer to the **“TV vs. KL”** section in the official comment for details.
>
> >W2: When to use the FW algorithm to the Sinkhorn version ?
>
> Answer: We refer to the **“FW vs. Sinkhorn”** section in the official comment for details.
>
> >  why propose also the FW version ?)
>
> Answer: As clarified in Section 1, both the FW and Sinkhorn formulations for **FPGW** (**not** FMPGW) are part of our contribution.
>
> -The FW algorithm is tailored to the **exact (non-entropic)** FPGW objective and is preferable when an accurate cost is required (e.g., kernel-based classification), whereas the Sinkhorn variant is useful for obtaining **robust partial matchings** in highly non-convex settings.
>
> - In the graph clustering experiment (Section 4.1) and graph classification experiment (Section N), we adopt **FPGW-FW** because precise transportation cost is more important in these tasks.
>
> > W3 The numerical experiments are not that illuminating....
>
> Answer:
>
> -**Graph clustering**: This is explicitly a **toy** example, following the setup in Vayer et al. [2]. Our goal here is to illustrate our novel FPGW barycenter formulation, not to claim large-scale empirical superiority.
>
> - **Graph classification**:  We use standard TU benchmark datasets, including ENZYMES (600 graphs, 6 classes, average $\sim 32.6$ nodes and $\sim 62.1$ edges) and PROTEINS (1113 graphs, 2 classes, average $\sim 39.1$ nodes and $\sim 72.8$ edges). These are widely used in FGW [2], UGW [3], and related GW-based works.
>
> - **Graph matching / alignment.** We follow the setup of Robust GW [4], using both TU benchmarks and the real-world Douban dataset (approximately 129K users, 58K movies, 1.7M social links). These graphs are far from “toy” and represent realistic scales for GW-based graph matching in the literature.
>
> - **Geometry matching.**  The geometry experiment also follows Robust GW [4] (referenced as [4] in our paper). The underlying graphs have roughly 4000–5000 nodes. This experiment is essentially a visual subgraph matching benchmark, not an isolated “mesh-only” toy example. In the revision, we clarify this and move it to a sub-section to make its illustrative nature explicit.
>
>
> > W4: For graph clustering, did not understand how the parameters were chosen. The number of non-outlier nodes appears to be known. How can that relate to any practical situation? How were set the other parameters? Also, it is written on page 7:
>
> Answer:We refer to the **“parameter selection”** section in the official comment for details on how we select parameters.
> In this experiment, we assume that the proportion of outliers is known. If it is unknown, FPGW can still be applied, but in that case a grid search is required.
>
> > W4: Why that? Any proof of this assertion?
>
> Answer: In this clustering experiment, outlier nodes should not contribute to either the clustering center or the clustering assignments.
> Under the FGW setting, these outliers must still be matched, which distorts both the estimated distances and the barycenter used in the k-means procedure; consequently, FGW does not produce accurate clustering results.
> In contrast, FPGW/FUGW are more likely to **ignore** outlier nodes when computing graph distances and the barycenter (i.e., the representative graph). As a result, they yield significantly more reliable clustering performance in the presence of outliers.

---

> ### Author Response · Authors · 2025-11-18
> **Weakness 5-7**
>
> > W5: In 4.2, is the fact that only 50% of the nodes are extracted a known fact ?
>
> Answer: The “50\%” refers only to the way the benchmark subgraph is constructed (following Robust GW [4]) and is **not** known nor used by any method. We have clarified this point in the revision.
>
> > W1: It seems that globally the method is only valid if each node corresponds to a node in the other graph, while an interest of the OT approach for ML on graphs is that it allows a change of scale in some situations (one node might be matched to 5 or 10 in the second graph, if need be). Also, knowing $N_G$ in advance seems to be a weak point of the method.
>
> Answer: We fully agree that OT can allow many-to-one correspondences (one node matched to several nodes). In fact, under the balanced setting $p_i = 1/\sum_{i'} p_{i'}$ and $q_j = 1/\sum_{j'} q_{j'}$, both our method and FUGW achieve higher matching accuracy, which highlights this flexibility. Equation (103) in the appendix corresponds to an alternative unbalanced configuration, applicable only to methods that can handle mass variation (FUGW, RGW, FPGW).
>
> > W2: Table 1 has to be re-considered: are the std of the results (in %) precise up to 2 digits ? How come some methods have a ±0.00 std ? This is not statistically sound. What is the unit of Time ? The results should include the FW algorithm.
>
> Answer: Thank you for pointing this out.
> The zero standard deviation values occurred mainly in the Synthetic Experiment formethods that rely only on structural information. In our original setup, the syntheticdataset consisted of 300 randomly generated graphs that all shared the same edgestructure, differing only in node attributes. As a consequence, for methods that do not usenode features, evaluating 300 graphs effectively amounted to evaluating the same graph300 times, producing an artificially low (often zero) standard deviation. We acknowledgethis issue and apologize for the unclear experimental design.To address this, we redesigned the synthetic experiment to ensure that each of the 300graphs contains a different subgraph, preventing the repeated evaluation of identicalstructures. After re-running all methods under this corrected setting, the updated results(including non-zero and statistically meaningful standard deviations) are reported in Table1.
>
>
> We also clarify explicitly that:
> - time is measured in seconds;
> - FW-based results are included for completeness where they are relevant.
>
>
> > The example in 4.3 (geometry matching) is of limited impact. Also, the underlying task (“geometry matching task”) is not really detailed.
>
> Answer: The dataset and setup in Section 4.3 follow the Robust GW framework [4]. Although the data are meshes, the underlying structure is still a graph, and the task is subgraph matching with a known ground truth.
>
> We clarify the construction as follows:
> Given a large graph $G$ and a subgraph $G_{\text{sub}}$, we define the ground-truth correspondence matrix $A$ by $A_{ij}=1$ if node $i$ in $G$ corresponds to node $j$ in $G_{\text{sub}}$. In our experiment, $A = I_{N_1 \times N_2}$, where $N_1$ and $N_2$ are the sizes of the subgraph and the larger graph, respectively; $G_{\text{sub}}$ is formed by selecting the first $N_1$ nodes of $G$.
>
> Thus, Section 4.3 is a visual **subgraph-matching** example rather than an independent geometry-only task. In the revised version, we move it to a sub-section and clarify its role accordingly.

---

> ### Author Response · Authors · 2025-11-18
> **Weakness 8-9**
>
> > W8: A limitation of methods based on Fused GW is often the computation time. Is it also the case here ? Or not because of the sinkhorn version ? What is the theoretical complexity and its scaling ? The used datasets for 4.1 and 4.2 are not fully described but they are quite small graphs. Can the method be extended to larger graphs (in terms of nodes) ?
>
> ### Answer:
> Our current work focuses on the standard FPGW solvers and does **not** yet incorporate low-rank or stochastic acceleration techniques.
> Therefore, the theoretical complexity matches that of FGW:
> - **Cost per iteration**
>   - exact OT solver: $ \mathcal{O}(n^3) $,
>   - Sinkhorn-based solver: $\mathcal{O}(n^2)$ per iteration.
>
> > The used datasets for 4.1 and 4.2 are not fully described...
>
> **Answer:**
> The toy example in Section 4.1 follows the same small-scale setting as Vayer et al. [2].
> The graph matching experiments, however, include:
> - standard **TU benchmarks** (e.g., PROTEINS, 1113 graphs, average $\sim 39$ nodes), and
> - the **large-scale Douban graph** with $\sim 10^5$ nodes and $\sim 10^6$ edges.
>
> > Can the method be extended to larger graphs
>
> **Answer:**
> Yes.
> On the Douban dataset, whose graph size is $\sim 10^5$ nodes:
> - **FPGW–Sinkhorn** takes **300–400 seconds**,
> - **FUGW** takes **$\sim 1500$ seconds**.
>
> These results show that our method scales to substantially larger graphs.
>
> > In the results, FUGW appear to have often similar performance than sink-FPGW (in Table 1). Is it the case ? Then, the contribution is about acceleration by using a sinkhorn algorithm, no ? Why is this acceleration not obtained on the Douban dataset ? Is is related to the size of this dataset ?
>
> **Answer:**
> We refer to the **“FUGW vs. FPGW”** section in the official comment for a full comparison.
>
> In summary:
> - Under the Sinkhorn formulation, the **KL penalty** (FUGW) is much slower than the **TV penalty** (FPGW).
>   Hence, **FUGW–Sinkhorn is consistently slower** than **FPGW–Sinkhorn**.
> - On the Douban dataset:
>   - **Sink-FPGW** and **FGW** take **300–400 seconds**,
>   - **FUGW** takes **$\sim 1500$ seconds**.
> - This behavior is consistent across all datasets:
>   **FGW ≈ FPGW**, while **FUGW is significantly slower** in the Sinkhorn setting.
>
> Thus, the performance difference is due to the **choice of divergence (TV vs. KL)** rather than the dataset size.
>
> Regarding running time, the apparent lack of acceleration on the Douban dataset is due to differences in parameter settings. In the original experiment, we used smaller epsilon and tighter tolerance values for our sink-FPGW (epsilon 1e-4, maximum iteration 1e4, and tolerence 1e-9), which increased the runtime. After adjusting to consistent parameter settings across methods (same as FUGW: epsilon 2e-4, maximum iteration 1e3, and tolerence 1e-7), the runtimes are as follows: FPGW = 306.64 s, FGW = 370.52 s, and FUGW = 1674.82 s. This confirms that the Sinkhorn-based FPGW achieves significant acceleration compared to FUGW, while maintaining comparable accuracy.

---

> ### Author Response · Authors · 2025-11-18
> **Questions**
>
> > it seems that the definition of the total variation is not given (despite the 2 pages of notations). Also, it comes as $ \Vert . \Vert_{TV}$ or $ | . |_{TV}$ depending on the pages.
>
> Answer: We have added a precise definition of the total variation norm in the appendix and
> made the notation consistent throughout.
>
>
> > Titouan et al. (2019a) should be Vayer et al. (2019a)
>
> Answer: corrected.
>
> Some misprints (examples: “postivie” on page 5 ; “RAPAMETER” in title M. ; “As shown in Table 2” on page 8 -> should be Table 1)
>
> Answer: We corrected all reported typos and carefully re-checked the manuscript for similar
> issues.
>
> Add some ablative or comparative study to assess whether using the TV distance is sound for this problem.
>
> Answer: We refer "TV vs. KL" section in the official comment for details. This discussion has also been added to the updated version.
>
> > Compare the two proposed algorithms in many situations, and compare them better to existing approaches. For instance, it is not useful to compare them to methods assuming balanced sets, as this topic (partial/unbalanced vs balanced problem, has already been discussed in the literature). But the 2 proposed algorithms and ways to estimated the unbalances (+ ways to choose the parameters) should be compared to the literature.
>
> Answer: We added additional accuracy and wall-clock time tests in Appendix K.2. We refer the official comment, "FPGW vs. FMPGW" section for details. In conclusion, FMPGW-FW (Frank Wolfe) is slower than FPGW-FW.
> FMPGW-Sinkhorn(from PythonOT) is significantly slower than FPGW-Sinkhorn. When n>=2000,  FMPGW-Sinkhorn requires about 2 days and other three methods require 2-3 hours.
>
> The experiment is in an unbalanced setting, and the range of size is from 10 to 10^4.
>
> We thank the reviewer for these detailed comments and believe the revisions substantially improve the clarity and scope of the paper.
>
>
> ## References:
> [1] Thual et al., Aligning Individual Brains with Fused Unbalanced Gromov–Wasserstein, NeurIPS 2022.
> [2] Vayer et al., Fused Gromov–Wasserstein Distance for Structured Objects, Algorithms, 2020.
> [3] Séjourné et al., Unbalanced Gromov–Wasserstein Distance, NeurIPS 2021.
> [4] Kong et al., Outlier-Robust Gromov–Wasserstein for Graph Data, NeurIPS 2023.

---

### Official Review · Reviewer_Rx4S · 2025-11-04

**Soundness:** 3
**Presentation:** 2
**Contribution:** 3
**Rating:** 4
**Confidence:** 4

**Summary:**

Authors introduce a novel partial OT problem for structured data, such as graphs with node features,  named Fused Partial Gromov-Wasserstein (FPGW), which relates to a previous formulation called Fused Mass-constrained Partial Gromov-Wasserstein (FMPGW). They theoretically study the relationships between these two divergences and establish that FPGW define (semi-)metrics while endowed with $L_p$ distances as inner costs. They propose Frank-Wolfe algorithms to solve both problems and study their convergence. They also introduce an entropically regularized variant EPFGW inspired by previous works on unbalanced OT. For all OT problems, they study how to compute barycenters w.r.t these divergences and leverage those in Kmeans like algorithms to learn denoised graph structures.  The authors also showcase the empirical relevance of these metrics on many real-world datasets to perform subgraph matching tasks, matching of 3D meshes and finally, in the supplementary material, noise-robust graph matchings on several graph classification tasks.

**Strengths:**

-	introduce new variants for partial FGW matching, namely FPGW and an entropically regularized variant EFPGW. With solvers to estimate solutions to these OT problems, respectively a conditional gradient solver for FPGW and a sinkhorn-like solver for EFPGW.
-	Provide theoretical results on the convergence of the CG solver for FPGW and FMPGW (supplementary).
-	Provide theoretical results in Theorem 3.1 showing that: (1) FPGW can be formulated as a non-convex QP; (2) FMPGW and FPGW admit minimizers; (3) FPGW defines a (semi-)metric while endowed with $L_p$ losses.
-	Study in the supplementary relations between FPGW, FMPGW and FGW.
-	Introduce barycenter problems for these metrics and corresponding solvers to perform graph clustering, illustrated on a synthetic dataset containing SBM graphs with different number of clusters and varying levels of structural noise.
-	Showcase that EFPGW outperforms many competitors over 8 real-world graph datasets on subgraph matching tasks.
-	Show that EFPGW outperforms competitors on 3d mesh matching.
-	Study in supplementary the relevance of FPGW for noisy graph classification under different level of noise, showcasing that FPGW outperforms competitors when noise is high enough.

**Weaknesses:**

- **W1: overall clarity of the paper**: I believe the paper contains many writing issues which are detrimental to the clarity of the paper including:
   - L51: "similarity between datasets". I suggest authors to be more specific about what they mean here as it does not include datasets of structured data like graphs with or without node features.
   - L57-58: "structured feature data". This terminology seems very strange to me and unique to the paper. I would advice authors to change that for instance by referring to attributed graphs or graphs with node features, which admissible no matter to topology of the graph (metric or not)
   - L61-62: it is not clear in which sense relying on sinkhorn solvers for FUGW is a limitation.
   - I think it would be relevant to broaden the scope of the paper referring to the work of Chowdhury and al (2019) on GW between any networks in Section 2 as adjacency matrices are used in the experiments. The adjacency indicator function mentioned in Appendix L does not seem to be a metric as I believe it does not satisfy the triangle inequality.
   - Section 2.1: For comparison with your own results on FPGW, you should also detail the metric property of FGW when q=1.
   - Most theoretical results are not discussed in the paper for instance those in Theorem 3.1 or those only mentioned in the supplementary material.
    - Mention in L208 that you are referring to Eq. 13. And epsilon is not defined in the complexity analysis.
    - The caption of Figure 1 seems incomplete as the meaning of the red and purple nodes is not given.
    - No need to go to next lines in the caption of Figure 3.

- **W2: sinkhorn solver**:
   - The solver seems to solve for an entropically regularized variant of FPGW and not the exact problem. This should be clarified and referring to EFPGW in the rest of the paper instead of sink-FPGW would be clearer.
   - It is not clear to me why authors use a such regularization instead of a classical entropic regularization e.g the entropy of transport plan, see e.g Peyré and al 2016. Moreover authors should justify their choice of algorithm, as it does not seem to be able to solve the EFPGW problem but a relaxed version.

- **W3: relations between FPGW and FMPGW**:
   - I believe that Proposition K1 and so on should be mentioned in the main paper as these are important to understand the relevance of introducing two different solvers. Moreover the study of the relations between both OT problems seem incomplete as I believe that a user would be interested in how to link $\lambda$ (in FPGW$ and $\rho$ (in FMPGW). Could authors try to complete this proposition in this direction ? Intuitively i think that a relation can be rather simply established using KKT conditions even but maybe i am wrong.

- **W4: benchmark in Table 1**:
   - Overall it seems that the node features play a huge role to get a proper matching between studied graphs. Therefore i think it would be appropriate to increase the number of existing baselines able to leverage those, such as srFGW and FPGW (equivalent to FMPGW).
   - Could authors explain the trick mentioned in eq.106 in the supplementary and its impact in the benchmark ?
   - It is not clear how hyperparameters have been validated for each method include theirs, could authors detail that ? Moreover could they provide a sensivity analysis on the parameters $lambda$ and potentially $epsilon$ ?
   - Could authors explain why there are that significant differences in computational speed between sinkhorn-based solvers for FUGW and EFPGW ?

**Tipos**:

   - Feydy & al (2017) seems to be misused in many ways and i believe that authors refer instead to a paper of Vayer and al.
   - L75-76: "Note" -> to remove ?
   - L114: "structure" -> "structured" ?
   - L168: "In" -> "In the" ?
   - There are many mistakes in math notations within the paper such as the one in L177, $FPGW_{\rho}$ which should be $FPGW_{r, L, \lambda}$, please make a thorough review.
   - L384. graphs

**Questions:**

I invite the authors to directly address the weaknesses mentioned above. Overall I consider that the paper has many interesting contributions, but their presentation is rather poor. If the authors address my concern, I would gladly increase my score.

---

> ### Author Response · Authors · 2025-11-16
> **W1: Overall clarity of the paper.**
>
> We thank the reviewer for the careful reading and helpful comments.
>
> > L51: “similarity between datasets”.
>
> Answer: We have revised the sentence to: “Optimal transport ... due to their capacity to measure the similarity between unstructured datasets, such as point clouds and images.”
>
> > L57–58: “structured feature data”.
>
> Answer: We have replaced this terminology with “attributed graphs.”
>
> > L61–62: It is not clear in which sense relying on Sinkhorn solvers for FUGW is a limitation.
>
> Answer: We refer to the "FUGW vs. FPGW" in the official comment for the limitation of FUGW.
> In summary, there are two limitations:
> (1) Practical: Sinkhorn-based FUGW optimizes a KL-regularized relaxation, which encourages mass splitting. In graph matching, this often leads to fractional and less accurate correspondences.
> (2) Theoretical: Unlike unbalanced OT, metric properties of UGW/FUGW under entropic regularization remain unclear. This motivates exploring formulations that do not require KL-based Sinkhorn relaxations.
>
> > The adjacency indicator in Appendix L.
>
> Answer: We agree. In graph-based GW formulations, the underlying space is not necessarily a metric measure space but a **gauge measure space**, where pairwise relations are given by a symmetric measurable function rather than a metric. This relaxation is standard and discussed in Remark L.1, with citation to Beier et al. (2022).
>
> > ... also detail the metric property of FGW when $q=1$.
>
> Answer: We appreciate the suggestion. The metric property of FGW for $q=1$ is established in our manuscript. Theorem 3.1 shows that the relaxed triangle inequality becomes the standard triangle inequality when $q=1$. The formal statement is presented in Theorem D.3, with a full proof provided in Appendix D. In the updated manuscript, we highlight this connection more clearly in Section 2.1.
>
> > Most theoretical results...
>
> Answer: Due to page limits, we summarized the main conclusions in the main text and provided full proofs in the appendix. The major theoretical contributions include:
> (1) an equivalent formulation simplifying FPGW optimization,
> (2) existence of minimizers,
> (3) metric properties of FPGW,
> (4) convergence guarantees for the FW solver, and
> (5) a detailed relationship between FPGW and FGW (Appendix K).
> In the revised version, we will expand the discussion of these theoretical findings for improved clarity.
>
> > L208: mention explicitly that you are referring to Eq. 13. Also, epsilon is not defined in the complexity analysis.
>
> Answer: Fixed. Line 208 now references Eq. 13 explicitly. We also added that $\epsilon>0$ denotes the weight of the entropic regularization term in the Sinkhorn objective.
>
> > Figure 1 caption....
>
> Answer: There is no "red" note in Figure 1. We guess the reviewer refers to the **orange** node (left graph) and purple node (right graph). These nodes represent unmatched mass, contributing to the penalty term in FPGW. We clarified this in the revised caption. This explanation has been added to the caption.
>
> > No need for line breaks in the caption of Figure 3.
>
> Answer: Fixed.

---

> ### Author Response · Authors · 2025-11-16
> **W2 sinkhorn solver:**
>
> > The solver solves an entropic variant of FPGW, not the exact problem...
>
> Answer:
> We agree. In the original submission, we noted that the Sinkhorn algorithm solves a relaxation of the entropic FPGW problem (see lines 234–237), similar to UGW/FUGW. In the updated version, we further emphasize this point.
>
> > Why use this regularization instead of classical entropic OT (e.g., Peyré et al. 2016)?
>
> Answer: Thank you for the sharp comment.
>
> There are indeed **two legitimate KL regularization choices** in the literature, and both have been used for different variants of entropic (unbalanced) GW.
>
> ### Form (1)  $\mathrm{KL}\bigl(\gamma \parallel \mu \otimes \nu\bigr)$
>
> This formulation appears in
> - Rioux et al. (2023) [1],
> - Le et al. (2021), Eq. (3) [2].
>
> ### Form (2)  $\mathrm{KL}\bigl(\gamma^{\otimes 2} \|\ (\mu \otimes \nu)^{\otimes 2}\bigr)$
>
> This alternative appears in
> - Séjourné et al. (2020) [3],
> - Le et al. (2021), Eq. (4) [2].
>
> Thus, **both formulations are valid**, and both have theoretical support in prior work.
>
> However, since our **FPGW is intrinsically an unbalanced formulation**, we adopt the **second KL term**, which is also the one used in the **PythonOT implementation** of UGW/FUGW. The first formulation is more common in theoretical analysis, whereas the second is widely adopted in practice for unbalanced or partially balanced variants.
>
> In future work, we plan to implement **both** KL regularization variants for FPGW and provide algorithms corresponding to each formulation.
>
> **Algorithm implementation.** To the best of our knowledge, for Case 1, an efficient algorithm is available only when the structures $C_1$ and $C_2$ correspond to quadratic costs. Graph distances or adjacency matrices do not satisfy this requirement. In contrast, most existing Sinkhorn-type algorithms are designed for Case 2.
>
> ---
>
> **References**
>
> [1] Rioux, G., Goldfeld, Z., & Kato, K. (2023). *Entropic Gromov-Wasserstein Distances: Stability and Algorithms*. arXiv:2306.00182.
> Link: https://arxiv.org/abs/2306.00182
>
> [2] Le, K., Le, D., Nguyen, H., Do, D., Pham, T., & Ho, N. (2021). *Entropic Gromov-Wasserstein between Gaussian Distributions*. arXiv:2108.10961.
> Link: https://arxiv.org/abs/2108.10961
>
> [3] Séjourné, T., Vialard, F.-X., & Peyré, G. (2020). *The Unbalanced Gromov-Wasserstein Distance: Conic Formulation and Relaxation*. arXiv:2009.04266.
> Link: https://arxiv.org/abs/2009.04266
>
>
> > It does not seem to be able to solve the EFPGW problem, but a relaxed version.
> Revised Answer:
>
> Answer: In theory, the Sinkhorn algorithm can be developed only when the underlying optimization problem is convex (so that strong duality holds) and an equivalent dual formulation is available. GW-type problems (including GW, FGW, UGW, and FPGW) are inherently non-convex. Their entropic relaxations also remain non-convex in the joint variable $(\gamma,\pi)$. However, conditioning on either variable individually yields a convex subproblem (see our Propositions I.1 and I.2). Therefore, to the best of our knowledge, Sinkhorn-type algorithms for GW problems can currently be applied only to these relaxed formulations, rather than to the original non-convex objectives.

---

> ### Author Response · Authors · 2025-11-16
> **W3: relations between FPGW and FMPGW**
>
> > Proposition K.1 and related results should appear in the main paper.
>
> Answer: We will add an informal summary of these relations to the main text.
> We refer to the section "FPGW vs. FMPGW" for details in the official comment. This discussion has been added to the updated manuscript.
>
>
> > Please clarify the relationship between $\lambda$ (FPGW) and $\rho$ FMPGW.
>
> Answer: Both control the effective transported mass. Increasing $\lambda$ encourages larger transported mass in FPGW, while $\rho$ directly specifies transported mass in FMPGW. We clarify this connection in Appendix K.
>
> We refer "parameter analysis" section for details, and it has been added to the updated manuscript.

---

> ### Author Response · Authors · 2025-11-16
> **W4: Benchmark in Table 1. and additional comments**
>
> > More baselines leveraging node features should be compared (e.g., srFGW, FPGW/FMPGW).
>
> Answer: We found no method named “srFGW’’ in the literature and believe the reviewer refers to entropic FGW.
> Balanced OT approaches such as FGW/entropic FGW are not suitable for partial matching.
> Additionally, FPGW (non-entropic) is highly non-convex; FW often converges to suboptimal minima. Thus it is not added as our method in this experiment.
> We refer to "FW vs. Sinkhorn" in the official comment for a detailed explanation.
> In addition, we've added the "FMPGW-FW" as an extra baseline.
> In the updated Table 1, we can still observe that FPGW-Sinkhorn admits state-of-the-art performance in most of the datasets.
>
>
> > Explain Eq. 106 in the supplement.
>
> Answer:
> Setting $p_i=q_j=\mathrm{constant}$ is standard in PGW/UGW/FUGW/FPGW. It enforces uniform node masses and increases the chance of partial matchings when total masses differ. We clarified this in the supplement.
>
>
> > How are hyperparameters validated?
>
> Answer:
> In graph matching, we set $\lambda$ large enough to ensure full transportation from the smaller graph.
> In graph classification, we set $\rho=1$.
> For other baselines (UGW/FUGW), we perform a grid search and choose the best-performing hyperparameters. We use the smallest stable entropic weight to avoid NaNs.
> We refer "parameter selection" section in the official comments for details.
>
> > Provide sensitivity analysis.
>
> Answer:
> Added as Remark M.1. We explain how $\lambda$ acts as a soft threshold and how selecting sufficiently large $\lambda$ guarantees maximal transported mass.
> We refer to "parameter selection" in the official comments for details.
>
>
> > Why is EFPGW faster than FUGW Sinkhorn?
>
> Answer:
>
> Two reasons:
> (1) TV (linear penalty) leads to simpler updates than KL (nonlinear).
> (2) EFPGW uses an optimized implementation. The FUGW code we use is from PythonOT.
> We discuss this in the revised text.
>
>
> ## Additional corrections.
>
> We have fixed all noted typos, including L75–L76, L114, L168, L177, and L384.
> We also corrected misuse of the Feydy et al. (2017) reference, replacing it with the correct Vayer et al. citation where appropriate.

---

> > ### Comment · Reviewer_Rx4S · 2025-11-27
> > **Answer to authors**
> >
> > Thank you for your detailed answer.
> >
> > By srFGW, I meant the semi-relaxed Fused-Gromov Wasserstein algorithm also introduced in the srGW paper, which includes experiments with node features, with details in the supplementary material. An implementation is available in POT so i would still invite authors to add those results in the benchmark.
> > In the revised manuscript, I suggest referring to FW-FPGW(POT) as FMPGW since it corresponds to a Frank-Wolfe solver for this problem, which is already used in previous works as already mentioned in the first version of the paper.
> >
> > I find surprising that FW-FPGW perform so poorly compared to the other competitors and especially its entropic counterpart. Could authors further discuss this matter ? I wonder whether these results are essentially related to a very high sensitivity of these solvers to their initialization, leading to a poor minimization of the objective, or whether the objective itself is drastically less relevant for the task at hand than the one tackled in the entropic version.
> >
> > I agree with other reviewers that many of the contributions are incremental, but authors still complete those with interesting additional ones on both theoretical or algorithmic sides. I still find unfortunate that authors did not manage to establish more formal links between $\lambda$ and $\rho$  but overall I find the rebuttal convincing. Therefore I will increase my score from 4 to 6.

---

> ### Author Response · Authors · 2025-11-29
>
> We thank the reviewer for the helpful discussion.
>
> > By srFGW, I meant the semi-relaxed Fused-Gromov Wasserstein algorithm.
>
> We have added this baseline and update the results in the updated manuscript. Since srFGW is essentially a balanced FGW formulation, its performance is limited.
>
> > I suggest referring to FW-FPGW(POT) ...
>
> Thanks. We have corrected the naming.
>
> > I find surprising that FW-FPGW perform so poorly
>
> The FW-FPGW, FW-FMPGW, and PGW methods indeed show similar behavior and all perform poorly. The following detailed explanation has been added to the updated version. In theory, FW-FPGW and FW-FMPGW are well-suited for the graph matching experiment, since a perfect matching would yield zero transportation cost. However, in practice, we use adjacency matrices (0–1 matrices) as the feature cost, which creates a highly non-convex optimization landscape with many local minima. The Frank–Wolfe methods are prone to getting trapped in these local minima. In contrast, the Sinkhorn algorithm achieves better performance because entropic regularization improves the objective's convexity. This observation also motivates our development of the Sinkhorn-FPGW variant in this paper.

---

### Author Response · Authors · 2025-11-13
**Answers for common questions**

## Novalty/contributions of this work.

We agree that the formulation of FPGW naturally builds upon two existing components:
(1) Partial OT / Partial GW (Chapel et al., 2020), and
(2) Fused GW (Vayer et al., 2019).

However, to the best of our knowledge, several **key theoretical and algorithmic aspects** of the fused partial setting have **not been studied** before:

- the **metric / (semi-)metric property** of the fused partial formulation,
- the **formal relationship** between FPGW, FGW, and FMPGW,
- the **derivation and analysis** of both the **Frank–Wolfe solver** and the **Sinkhorn-based solver** for the FPGW objective,
- and the corresponding **barycenter formulation and algorithms**, which were not available in the literature.

We note that neither the FW nor the Sinkhorn for the FPGW used in this paper exists in prior work; otherwise, we would have directly used the implementation in libraries such as POT instead of deriving new ones.

The incremental nature of our contribution is similar to that of
**Fused Unbalanced GW** (Thual et al., *NeurIPS 2022*),
which also combines two known OT ideas (FGW + UGW) but contributes new theory, new algorithmic formulations, and new applications.
Our work plays a similar role for the partial fused setting.

## FPGW vs. FUGW ---TV penalty vs. KL penalty.
1. **Theory:** We show that **FPGW admits a (semi-)metric**, while the **metric property of FUGW remains unclear**. In the graph classification experiments, we also do **not observe an advantage** of FUGW over FPGW (see Appendix N).
2. **Algorithms:** **FPGW** can be optimized by both a **Frank–Wolfe (FW)** and a **Sinkhorn** solver, whereas **FUGW** can only be solved with the Sinkhorn algorithm.
3. **Computational aspects:** The KL divergence in FUGW is **nonlinear**, while the TV penalty in FPGW is **linear**; we hypothesize this contributes to faster convergence for FPGW even under entropic regularization.
   *Example:* In graph classification, **FUGW (Sinkhorn)** requires **100–400 s**, while **FPGW (Sinkhorn)** completes in **7–40 s** (see the table 1 in the updated PDF).

## Relation between FPGW vs. FMPGW
Contribution issue:
- As noted in Section 3, FMPGW was already introduced in prior works, and we do claim the associated FW/Sinkhorn algorithms are **NOT** new contributions of this paper (although we provide convergence analysis, barycenter Algorithm, which we can not find a related reference).  We further emphasis it in the updated version.
- The FW/Sinkhorn solvers for FMPGW in PythonOT were implemented on **Sept. 15, 2025**, while this manuscript was submitted on **Sept. 30, 2025**.
During the preparation of the submission, we did not continuously monitor the most recent code updates.
However, in the updated version, we have clearly stated that this part is **not** our contribution.


In theory, the two formulations are related; however, due to the inherent non-convexity, it is difficult to assert any formal equivalence. We refer the reader to Appendix K for a detailed discussion.

In practice, **FMPGW–Frank–Wolfe is slower** than our **FPGW–Frank–Wolfe** method, as the mass-constraint partial OT linear-programming solver used in FMPGW-FW is noticeably slower than the partial OT solver employed in FPGW-FW.

Moreover, **FMPGW–Sinkhorn is significantly slower** than our **FPGW–Sinkhorn**. For FMPGW, each Sinkhorn iteration requires six matrix operations (and up to twelve in practice), whereas FPGW requires only two operations in theory (and three in practice).

We will include the following wall-clock comparison in the revised version.
Here, n denotes the data size (ranging from 10 to 10^4).
Please refer to Appendix K.2 in the updated manuscript for the detailed experimental settings.



[result for $\lambda=0.1$](https://anonymous.4open.science/r/fused-pgw-041B/wall-clock/1_1124.png)

[result $\lambda = 1$](https://anonymous.4open.science/r/fused-pgw-041B/wall-clock/1_1124.png)

[result for $\lambda = 10$](https://anonymous.4open.science/r/fused-pgw-041B/wall-clock/10_1124.png)

[code](https://anonymous.4open.science/r/fused-pgw-041B/wall-clock/wallclock_test.py)

---

### Author Response · Authors · 2025-11-18
**Answer for common questions**

## Parameter selection

### FMPGW and FPGW
In the **FMPGW** formulation, $\rho$ represents the amount of transported mass.
The choice of $\rho$ typically depends on prior knowledge or the requirements of each experiment.
For example, in **graph matching**, where full transportation of the smaller graph is required, we simply set
$\rho = \text{mass of the smaller graph}$.

In the **FPGW** formulation, the parameter $\lambda$ plays a role analogous to $\rho_1$ and $\rho_2$ in UGW/FUGW, serving as a *threshold* that determines how much mass is transported.  Intuitively speaking, a larger $\lambda$ admits more transported mass and vice versa.
When $\lambda$ exceeds a certain threshold, e.g.
$$
2\lambda \ge \max_{i,j,i',j'} \, \omega_1|x_i - y_j| \frac{1}{\min(|p|,|q|)} + \omega_2|d_X(x_i, x_{i'}) - d_Y(y_j, y_{j'})|^2,
$$
the solution transports $\min(|\mu|, |\nu|)$, which corresponds to the largest possible transported mass.
We exploit this property when choosing $\lambda$ in the graph-matching experiments.

In general, larger $\lambda$ values imply higher mass-penalty terms and therefore lead to transporting a larger fraction of the total mass.
Without prior information, $\lambda$ can be tuned via a **grid search**, similar to the practice used for UGW, FUGW, and other unbalanced OT formulations.

## Other baselines
For FUGW and UGW, we perform grid search over key parameters (e.g., $\rho_1, \rho_2$). All other controls (maximum iterations, stopping thresholds) use recommended defaults. For entropic regularization, we use the smallest value that avoids numerical instability (NaNs), since larger values harm accuracy and smaller values cause numerical issues.

---

### Meta-Review · Area_Chair_FzeY · 2026-01-07

**Summary:**

Authors propose a novel method for solving the Fused Partial Gromov-Wasserstein, and study some of its properties. Most of the reviewers find that the paper is well-written, but the contribution is noted to be rather incremental wrt. existing literature or even public repository such as the PythonOT package. While authors discussed the merits of their approach in the rebuttal (notably wall-clock time execution vs existing implementations), I also concur that added value of the contribution might be a little bit insufficient with respect to what is expected for an ICLR paper. I expect that the authors might change their paper so as to better magnify the differences and advantages wrt. existing literature and code.

**Reviewer Concerns:**

Novelty (all), partially addressed

**Reviewer Scores:**

Apart from reviewer WxFv that mentionned raising its note to 6, I do not believe that the other reviewers would have changed their scores.

---

### Decision · Program_Chairs · 2026-01-26

Reject